# Physiological DNA damage promotes functional endoreplication of mammary gland alveolar cells during lactation

Rut Molinuevo[1,2,4], Julien Menendez[1,2,4], Kora Cadle[1], Nabeela Ariqat [1], Marie Klaire Choy[1], Cayla Lagousis[1], Gwen Thomas[1], Catherine Strietzel[3], J. W. Bubolz[3] & Lindsay Hinck [1,2] ✉

Lactation insufficiency affects many women worldwide. During lactation, a large portion of mammary gland alveolar cells become polyploid, but how these cells balance the hyperproliferation occurring during normal alveologenesis with terminal differentiation required for lactation is unknown. Here, we show that DNA damage accumulates due to replication stress during pregnancy, activating the DNA damage response. Modulation of DNA damage levels in vivo by intraductal injections of nucleosides or DNA damaging agents reveals that the degree of DNA damage accumulated during pregnancy governs endoreplication and milk production. We identify a mechanism involving early mitotic arrest through CDK1 inactivation, resulting in a heterogeneous alveolar population with regards to ploidy and nuclei number. The inactivation of CDK1 is mediated by the DNA damage response kinase WEE1 with homozygous loss of *Wee1* resulting in decreased endoreplication, alveologenesis and milk production. Thus, we propose that the DNA damage response to replication stress couples proliferation and endoreplication during mammary gland alveologenesis. Our study sheds light on mechanisms governing lactogenesis and identifies non-hormonal means for increasing milk production.

Endoreplication is the process by which a diploid cell (2C DNA content) undergoes DNA replication in the absence of cell division to become tetraploid (4C DNA content) or polyploid (>4C DNA content). Developmentally programmed endoreplication occurs in several mammalian tissues during pregnancy and is usually linked to terminal differentiation. In the placenta, trophectoderm cells undergo endoreplication and differentiate into trophoblast giant cells, which penetrate the uterus and promote blastocyst implantation[1–3]. Subsequently, in the uterus, stromal cells of the endometrium endoreplicate and differentiate into decidual cells, which further facilitate blastocyst implantation and vascularization[4–6]. Another example is pregnancy-induced liver growth occurring through hepatocyte hypertrophy that is generated by endoreplication[7]. In the mammary gland (MG), alveolar cells undergo endoreplication at the onset of lactation[8–11]. While these phenomena have long been observed and considered adaptations necessary for tissue expansion during pregnancy, the molecular mechanisms driving these pregnancy-induced endoreplication events remain poorly understood.

The MG plays an essential role in the survival of mammalian species by producing milk required for the nourishment of offspring. During pregnancy, the MG undergoes a profound morphological change known as alveologenesis, in which epithelial luminal progenitors proliferate and subsequently differentiate into polyploid alveolar cells that secrete milk during lactation[8–12]. This polyploidization of the MG is conserved across many mammalian species, including mice and humans, and it is required for efficient milk production[8–11]. Once

[1]Department of Molecular, Cell and Developmental Biology, University of California, Santa Cruz, CA 95064, USA. [2]Institute for the Biology of Stem Cells, University of California, Santa Cruz, CA 95064, USA. [3]Zoetis Inc., 333 Portage Street, Building 300, Kalamazoo, MI 49007, USA. [4]These authors contributed equally: Rut Molinuevo, Julien Menendez. ✉e-mail: lhinck@ucsc.edu

breastfeeding is complete, in a process known as involution, massive cell death clears these milk-producing polyploid cells and tissue remodeling brings the epithelium back to a pre-pregnancy-like state.

The mechanisms by which endoreplication is achieved are diverse and vary between tissues. Endoreplication results in tetraploid cells; however, cells can also undergo further endoreplication and become polyploid. This can be accomplished either by early mitotic arrest or cytokinetic failure[13]. Endoreplication induced by early mitotic arrest occurs when a cell undergoes DNA replication without progressing through mitosis, becoming tetraploid and mononucleated. Endoreplication through early mitotic arrest requires inhibition of the CYCLIN B/CDK1 complex that facilitates progression from the G2 phase to the M phase. In megakaryocytes, CYCLIN B is downregulated during endoreplication leading to CDK1 inactivation[14]. In other tissues, CDK1 is directly inactivated by CDK1 inhibitors. For example, in the placenta, the upregulation of the CDK1 inhibitor P57[Kip2] in response to FGF4 deprivation induces trophoblast stem cells to differentiate into trophoblast giant cells and endoreplicate by preventing progression through mitosis[15]. In the endometrium, upregulation of a different CDK1 inhibitor, P21[Cip1], has been suggested to inactivate CDK1 and induce G2/M arrest during the endoreplication of decidual cells[16]. Alternatively, during endoreplication by cytokinetic failure, a cell progresses through mitosis unperturbed, but fails to divide, resulting in a tetraploid binucleated cell. These binucleated cells can arise in several ways, such as failure to specify a cleavage plane due to insufficient RhoA activation[17] or cleavage furrow ingression failure due to improper anchoring of the actomyosin ring[18]. MG alveolar endoreplication has been suggested to require Aurora A kinase upregulation and cytokinesis failure[11]. Although the role of Aurora A during the G2/M transition and mitotic spindle assembly has been extensively studied, whether it is directly implicated in cytokinesis remains unclear[19]. Therefore, the mechanisms regulating the transition from a proliferative mitotic cell cycle to an endocycle in the MG have yet to be elucidated.

The DNA damage response (DDR) plays a central role in the regulation of the cell cycle, to ensure genomic stability and safeguard inheritance. In the event of DNA damage, the DDR kinases ATM and ATR initiate a signaling cascade that activates cell cycle checkpoints during the G1/S transition, intra-S or the G2/M transition, through inactivation of CDK/CYCLIN complexes[20]. These checkpoints permit the DDR to perform any necessary repairs before giving rise to a daughter cell. DNA damage as a consequence of exogenous genotoxic insults has been shown to trigger endoreplication and terminal differentiation through G2/M checkpoint activation[21–24]. Here, however, we identify an unconventional trigger, physiological DNA damage, which accumulates during the extensive cell proliferation of mid-pregnancy and drives these events at the onset of lactation. This mechanism involves the activation of the DDR to replication stress, and the subsequent activation of the G2/M checkpoint. WEE1 governs this process, revealing a role for this CDK1 inhibitor in the regulation of mammalian endoreplication.

While the G1/S checkpoint safeguards entry into the cell cycle and has been focused on extensively, our findings highlight a potential role for the G2/M checkpoint in coupling rapid tissue development of pregnancy with endoreplication and terminal differentiation. When considering the necessity of expansive growth during pregnancy, it is evident that multiple tissues (liver, endometrium, placenta, mammary gland) depend on a strict balance of proliferation, differentiation, endoreplication, and apoptosis[25]. In the absence of adequate DNA repair, extensive DNA damage most often results in apoptotic cell death. However, apoptosis would be deleterious to these tissues that require rapid rates of proliferation to develop functionally. Therefore, by coupling proliferation with terminal differentiation, we propose that endoreplication through the activation of the G2/M checkpoint presents a developmental advantage for cell survival and tissue function.

## Results

### Endoreplication results in a heterogeneous alveolar population during lactation

Mammary alveolar cells have been previously shown to undergo endoreplication during lactation, a process essential for efficient milk production[8–11]. How these cells become committed to endoreplication and the outcome of this endoreplication (in terms of DNA content and number of nuclei) remains unclear. It has previously been shown that a significant percentage of alveolar cells become tetraploid and binucleated during lactation[11,26]. Through immunohistochemistry (IHC) staining and in situ 3D DNA content analysis of tissue sections from lactation day (LD) 5 MGs, we further detect mononucleated polyploid (>4C) alveolar cells and binucleated alveolar cells containing polyploid nuclei (Fig. 1a). Additionally, we detect the rare occurrence of multinucleated cells (Fig. 1b, arrows). To better understand the heterogeneity of the alveolar population, we performed FACS DNA content analysis of the Cytokeratin-8 positive (CK8+) luminal cell population from MGs of nulliparous, pregnancy day (PD) 17.5, LD2 and LD5 mice (Fig. 1c–f and Supplementary Fig. 1a). As previously reported[11], we observe a substantial increase in the proportion of tetraploid (4C) cells during lactation (Fig. 1e, f). In addition, we identify a polyploid (>4C) subpopulation that arises at the end of pregnancy and expands during lactation (Fig. 1e, f and Supplementary Fig. 1a). By visualizing the FACS-purified CK8+ population harvested from LD5 MGs using immunofluorescent microscopy, we find ~35% of tetraploid (4C) cells are mononucleated, with the remaining ~65% binucleated, whereas polyploid (>4C) cells are observed to be both mononucleated and binucleated in a ~50/50 ratio (Fig. 1g, h). Together, these results demonstrate mammary alveolar cells are heterogeneous during lactation with respect to DNA content and nuclei number.

To further investigate the role of endoreplication during alveologenesis, we took advantage of the HC11 murine mammary cell line as an in vitro lactation model. This cell line resembles the MG in that it undergoes differentiation into milk-producing secretory cells when cultured in the presence of the lactogenic hormones dexamethasone, insulin, and prolactin (DIP). As previously published[27], we find ~80% of differentiated HC11 cells undergo G0/G1 arrest, remaining diploid (2C; Supplementary Fig. 1b, c). In addition, we identify populations of tetraploid (4C) and polyploid (>4C) HC11 cells that increase during differentiation (Supplementary Fig. 1b, c). This observation suggests HC11 cells undergo endoreplication during differentiation, in accordance with the role of endoreplication in milk production. To further investigate this, we treated undifferentiated HC11 cells at 80% confluence with blebbistatin (Blebbi, 30uM), a myosin II inhibitor that prevents cytokinesis and induces endoreplication through cytokinesis failure[28]. FACS DNA content analysis 6 h post-treatment shows blebbistatin efficiently induces mitotic arrest in HC11 cells, increasing the proportion of tetraploid (4C) cells (Supplementary Fig. 2a). By differentiation day 3 (DIP3), cells escape the mitotic arrest imposed by blebbistatin and undergo further endoreplication, becoming polyploid (4 C; Supplementary Fig. 2b). Increased endoreplication is accompanied by an increase in milk-containing domes, milk protein gene β-Casein (*Csn2*) and milk protein Perilipin-2 (PLIN2), detected by RT-qPCR and ICC respectively (Supplementary Fig. 2c–f). Together, these data demonstrate HC11 cells endoreplicate and endoreplication is increased when cell division is blocked, resulting in increased milk production. They are, therefore, a suitable in vitro model to investigate endoreplication during alveolar differentiation.

### CDK1 inhibition drives endoreplication of alveolar cells through an early mitotic arrest

Through visualization of FACS-purified HC11 cell subpopulations, we observe that a large proportion of polyploid (>4C) cells are mononucleated at DIP3 (Fig. 2a). While cytokinetic failure has been suggested to generate binucleated alveolar cells in vivo[11], the presence of

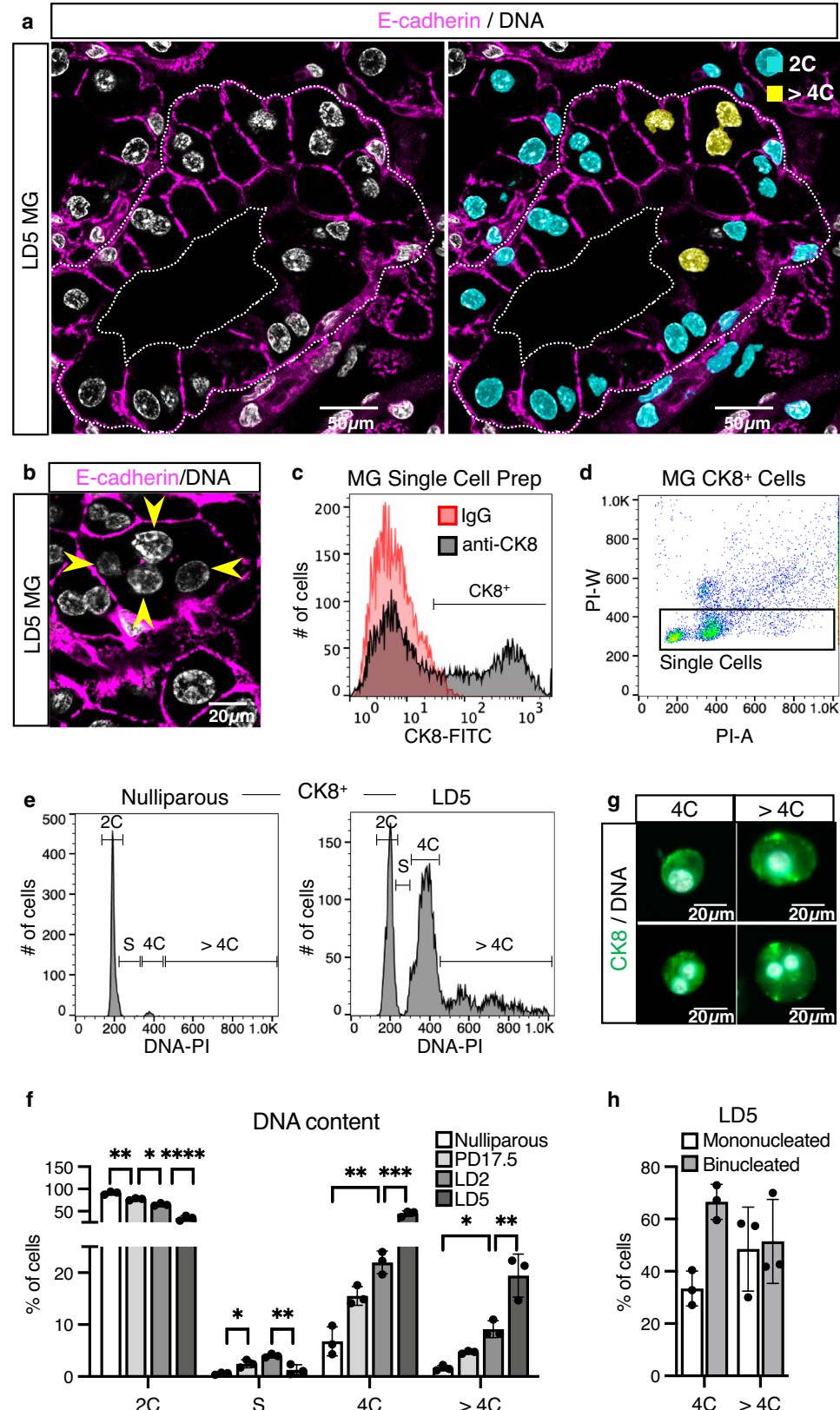

mononucleated polyploid cells both in vitro and in vivo indicates early mitotic arrest at the G2/M transition is also a contributing factor. Endoreplication through early mitotic arrest requires the inactivation of the mitotic regulator CDK1[29]. In addition, there must be a transition in the activity of CDK/CYCLIN complexes. CYCLIN B must be down-regulated to facilitate CDK1 inactivation and G2/M arrest, while the

activity of the CDK2/CYCLIN E complex must persist to allow for DNA replication[25]. Accordingly, we find, during lactogenic differentiation of HC11 cells, CYCLIN B expression is lost while CYCLIN E expression is maintained (Supplementary Fig. 3a, b). In addition, we find that this cyclin switch also occurs in the mammary gland at the onset of lacta-tion, when CYCLIN E is upregulated to a greater extent than CYCLIN B

**Fig. 1 | A heterogenous population of polyploid luminal cells emerges during lactation. a** Optical section from a 3D confocal image showing an alveolar unit from a MG at LD5 (left). Individual, complete nuclei color-labeled by DNA content, as calculated by the 3D sum integrated density of nuclear DNA staining using stromal cell nuclei as a reference for 2C DNA content (right; 2C in blue, >4C in yellow). E-cadherin detected in magenta. Nuclear DNA by Hoechst. White dotted line represents the border of the alveolus. **b** Optical section from a 3D confocal image at LD5 focused on a polyploid multinucleated alveolar cell. Arrows point to each of the 4 nuclei. E-cadherin shown in magenta. Nuclear DNA by Hoechst. **c** FACS analysis of a representative MG single cell preparation used for the identification of the luminal CK8$^+$ population. Red histogram represents the negative isotype control (IgG). Gate indicates the CK8$^+$ population in relation to the negative isotype control. **d** FACS analysis of the MG CK8$^+$ cells for the identification of single cells,

based on propidium iodide area (PI-A) versus width (PI-W). Red-to-blue color scale shows high-to-low density of data points. Black box shows the gating strategy for single-cell identification. **e** Representative FACS histograms for DNA content analysis of the CK8$^+$ population in the nulliparous (left) and LD5 MGs (right). **f** Percentage of CK8$^+$ cells with 2C, 4C or >4C DNA content, or in the S phase, as detected by FACS analysis, from nulliparous, PD17.5, LD2 and LD5 MGs. **g, h** Representative images (**g**) and quantification (**h**) of mononucleated and binucleated CK8$^+$ cells from LD5 MGs, sorted based on DNA content (4C or >4C). Nuclear DNA by propidium iodide. Data are presented as mean ± SD. Data analyzed by one-way ANOVA with Tukey's multiple comparison test (**f**) and by unpaired, two-tailed Student's *t* test (**h**). Data representative of *n* = 3 biologically independent experiments. *p* values: *<0.05, **<0.01, ***<0.001, ****<0.0001.

to allow for endoreplication (Fig. 2b, c). To investigate if CDK1 inactivation is sufficient to induce endoreplication during lactogenic differentiation, we treated undifferentiated HC11 at 80% confluence with the CDK1 inhibitor Ro-3306 (5uM). FACS DNA content analysis 6 h after treatment with Ro-3306 shows CDK1 inhibition efficiently induces mitotic arrest, as detected by the accumulation of tetraploid (4C) cells (Fig. 2d and Supplementary Fig. 3c). By DIP3, cells escape the mitotic arrest imposed by Ro-3306 and undergo further endoreplication, becoming polyploid (>4C; Fig. 2e and Supplementary Fig. 3d). Accordingly, CSN2 expression increases, as detected by RT-qPCR and WB (Fig. 2f, g). These results show CDK1 inhibition and early mitotic arrest are sufficient to drive endoreplication during lactogenic differentiation in HC11 cells, resulting in increased milk production. Through visualization of FACS-purified subpopulations, we observe CDK1 inhibition generates mono-, bi- and multinucleated polyploid (>4C) cells (Fig. 2h). In contrast, treatment with blebbistatin predominantly generates multinucleated polyploid (>4C) cells (Fig. 2h). This shows that the heterogeneity of the MG observed in vivo is recapitulated by CDK1 inhibition, but not by the cytokinesis failure induced by blebbistatin. Altogether, these results suggest early mitotic arrest imposed by CDK1 inhibition is involved in alveolar endoreplication.

## DNA damage during alveologenesis increases mammary alveolar endoreplication

Developmentally programmed endoreplication occurs in different mammalian tissues not only during pregnancy[1-11], but also during organogenesis and tissue regeneration in response to injury[21,22,30-34]. In addition, DNA damage elicited by genotoxic stress has been shown to induce endoreplication and terminal differentiation through activation of the G2/M cell cycle checkpoint in various mammalian tissues[21-24]. In the MG, DNA damage occurs in alveolar cells during pregnancy[35]; however, whether it plays a physiological role during alveologenesis remains unknown. To determine the extent of DNA damage during alveologenesis and lactation, we investigated the phosphorylation of histone H2A.X at Serine-139 (γH2AX), a site that is rapidly phosphorylated in the presence of DNA strand breaks[36]. We find γH2AX is present in CK8$^+$ luminal cells throughout alveologenesis, with the peak occurring at PD10.5 (Fig. 3a, b and Supplementary Fig. 4a). Similarly, we find that the DNA damage marker 53BP1 is also upregulated at mid-pregnancy (PD10.5 and PD15.5; Fig. 3c, d and Supplementary Fig. 4a). In HC11 cells, γH2AX is highest in undifferentiated cells that are actively proliferating and lower in confluent cells during competency and priming (Supplementary Fig. 4b, c). However, we detect an increase in γH2AX when HC11 cells undergo differentiation (Supplementary Fig. 4b, c), and phalloidin staining shows that γH2AX accumulates in cells forming milk domes (Supplementary Fig. 4d). These results suggest DNA damage plays a role in lactogenic differentiation and endoreplication. To investigate, HC11 cells were treated with DMSO-containing vehicle (DMSO) or doxorubicin (Doxo, 50 nM), which induces double strand breaks and DNA damage by inhibiting topoisomerase II during DNA replication. Accordingly, Doxo

results in an increase of γH2AX 24 h post-treatment (Supplementary Fig. 5a, b). FACS DNA content analysis shows the proportion of tetraploid (4C) HC11 cells increases after 24 h of Doxo treatment, indicating the G2/M checkpoint is activated (Supplementary Fig. 5c, d). By DIP3, an increase in the proportion of both tetraploid (4C) and polyploid (>4C) cells is detected (Supplementary Fig. 5e, f). Microscopic visualization of the FACS-purified population confirms that DNA damage-induced endoreplication generated both mono- and binucleated polyploid (>4C) HC11 cells (Supplementary Fig. 5g), recapitulating the heterogeneity observed in the MG (Fig. 1g, h). HC11 cell endoreplication is accompanied by an increase of CSN2, as detected by WB (Supplementary Fig. 5h). We also analyzed the HC11 cells undergoing cell death using propidium iodide staining and FACS, by quantifying the sub-G1 cells at 24 h and DIP3 after DMSO or Doxo treatment (Supplementary Fig. 5i, j). The results show that, by DIP3, DMSO treatment of cells increases the proportion of cells undergoing spontaneous apoptosis when compared to undifferentiated, DMSO-treated HC11 cells (Supplementary Fig. 5i, j). In contrast, while Doxo induces apoptosis in ~ 4% of HC11 cells 24 h after treatment, there is no increase in the proportion of cells undergoing apoptosis at DIP3 (Supplementary Fig. 5i, j). Altogether, these data demonstrate that, in the presence of DNA damage, HC11 cells preferentially undergo endoreplication and lactogenic differentiation, rather than apoptosis.

To examine the in vivo consequences of damaging DNA during pregnancy, we performed contralateral intraductal injection (IDI) of DMSO or Doxo into MGs at PD12.5 (Fig. 3e) to extend the period of DNA damage that peaks at PD10.5 (Fig. 3a–d and Supplementary Fig. 4a). Trypan blue IDI shows that the volume injected is efficiently delivered and distributed throughout the gland (Fig. 3f). We observe increased γH2AX in CK8$^+$ luminal cells 24 h post-injection with Doxo (Fig. 3g, h), suggesting that proliferating cells are susceptible to DNA damage. FACS DNA content analysis of Doxo-injected MGs at PD17.5 shows an increase in the proportion of tetraploid (4C) and polyploid (>4C) luminal populations (14.86% and 47.1% increase in the overall populations, respectively), while the diploid (2C) population decreases (5.23% decrease in the overall population) (Fig. 3i, j). This increase in endoreplication is accompanied by an increase in milk production, as detected by RT-qPCR for milk protein genes *Lalba*, *Wap*, *Csn2* and *Xdh1* (Fig. 3k). Collectively, these results indicate that DNA damage increases alveolar endoreplication through the activation of the G2/M checkpoint and, consequently, milk production. We also analyzed cells undergoing apoptosis by quantifying the percentage of sub-G1cells within the CK8$^+$ population. We find that Doxo treatment decreases the percentage of cells undergoing apoptosis (Fig. 3l, m), consistent with previous studies showing that cells undergoing endoreplication in response to genotoxic stress are more resistant to apoptosis[37]. In the context of MG development during pregnancy, endoreplication may represent the preferred mechanism to prevent proliferation of cells harboring DNA damage, rather than apoptosis, which would be deleterious to proper tissue growth and function.

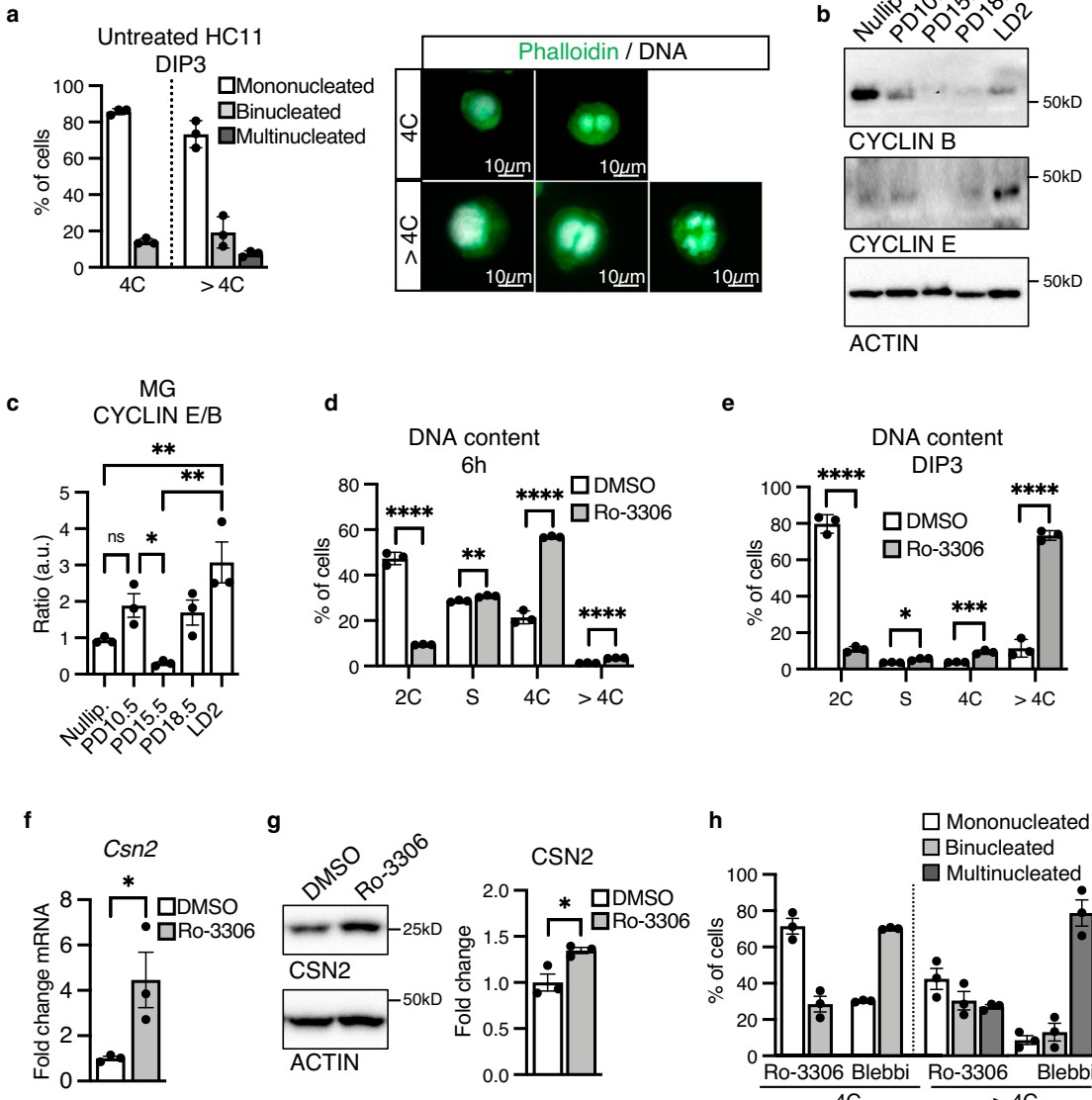

**Fig. 2 | CDK1 inhibition increases endoreplication and milk production in HC11 cells and recapitulates the heterogeneity observed in the MG.**
**a** Quantification (left) and representative images (right) of mono-, bi- and multi-nucleated (more than 2 nuclei) differentiated (DIP3) HC11 cells, sorted based on DNA content (4C or >4C). Nuclear DNA by propidium iodide. Phalloidin shown in green. **b**, **c** Representative western blot (**b**) and quantification (**c**) of CYCLIN E and CYCLIN B expression in nulliparous, PD10.5, PD15.5, PD18.5 and LD2 MGs. Quantification (**c**) shown as CYCLIN E/CYCLIN B ratio. **d** Percentage of HC11 cells with 2C, 4C or > 4C DNA content or in the S phase, as detected by FACS analysis, 6 h after treatment with DMSO or the Cdk1 inhibitor Ro-3306. **e** Percentage of differentiated (DIP3) HC11 cells with 2C, 4C or > 4C DNA content or in the S phase, as detected by FACS analysis, after treatment with DMSO or Ro-3306. **f** Quantification of *Csn2* expression in differentiated (DIP3) HC11 cells after DMSO or Ro-3306 treatment, as detected by RT- qPCR. **g** Western Blot (left) and quantification (right) of CSN2 expression in differentiated (DIP3) HC11 cells after DMSO or Ro-3306 treatment. **h** Percentage of mono-, bi- or multinucleated (more than 2 nuclei) differentiated (DIP3) HC11 cells treated with Ro-3306 or blebbistatin (blebbi) and sorted based on DNA content (4C or >4C). Data presented as mean ± SD (**a**, **d**, **e**) and mean ± SEM (**c**, **f**–**h**). Data analyzed by one-way ANOVA with Tukey's multiple comparison test (**c**) and unpaired, two-tailed Student's *t* test (**d**–**g**). Data representative of $n = 3$ biologically independent experiments. *p* values: *<0.05, **<0.01, ***<0.001, ****<0.0001.

## Replication stress results in activation of the DNA damage response and endoreplication during alveologenesis

To ensure genomic stability and safeguard inheritance, cells possess a DNA damage response (DDR) that monitors genomic integrity throughout the cell cycle. During normal development, cell proliferation frequently results in activation of the DDR due to replication stress[38,39]. Given the tremendous amount of proliferation that occurs during early alveologenesis, we hypothesized that replication stress may be the source of DNA damage driving endoreplication of mammary alveolar cells. By IHC staining, we find that replication protein A (RPA), which binds single-stranded DNA during replication stress[40],

accumulates during mid-pregnancy (PD10.5-PD15.5; Fig. 4a, b and Supplementary Fig. 6a), coinciding with the proliferative phase of alveologenesis and the accumulation of DNA damage markers γH2AX and 53BP1 (Fig. 3a–d and Supplementary Fig. 4a). The response to DNA damage by replication stress is mediated by the kinase ATR, which is activated by auto-phosphorylation at Threonine-1989 (pATR)[41,42]. By IHC staining, we find that, similarly to γH2AX, 53BP1 and RPA (Figs. 3a–d, 4a, b and Supplementary Figs. 4a, 6a), ATR is activated at PD10.5 when proliferation is at its peak (Fig. 4c, d and Supplementary Fig. 6a)[43]. However, while γH2AX expression peaks and decreases (Fig. 3a, b and Supplementary Fig. 4a), pATR persists until the end of

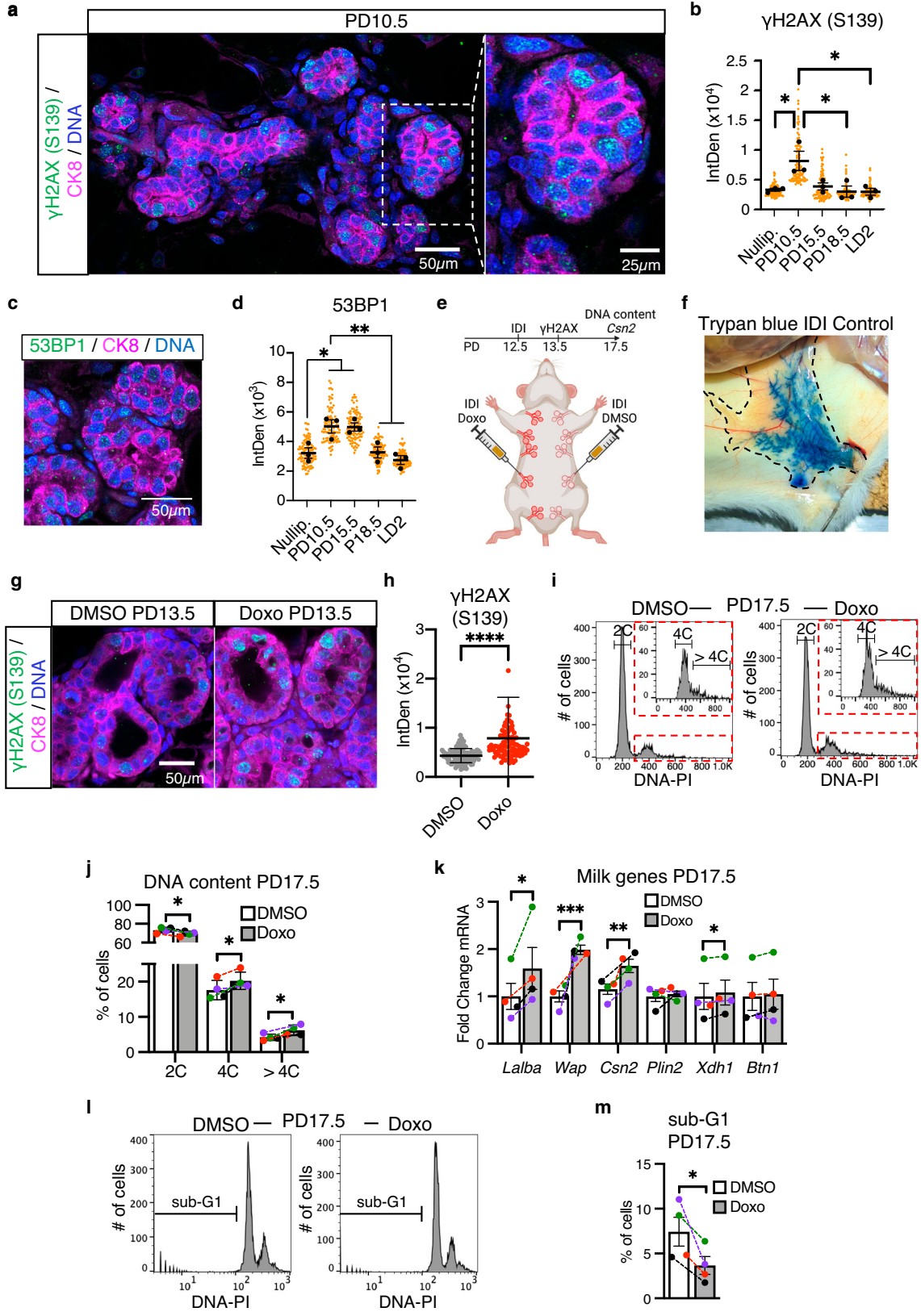

pregnancy, when endoreplication begins (Fig. 4c, d and Supplementary Fig. 6a–c). Contrarily, we find that activation of the DDR kinase ATM, which is induced by auto-phosphorylation at Serine-1981 (pATM), is downregulated as pregnancy progresses and lactation begins (Fig. 4e and Supplementary Fig. 6a, d, e)[44]. In addition, we find activation of CHK1, which is induced by ATR-mediated

phosphorylation at Serine-345 (pCHK1) and responsible for the activation of the G2/M checkpoint, is upregulated at LD2 (Fig. 4f–h and Supplementary Fig. 6a)[45,46]. These data show that replication stress occurs during the proliferative phase of alveologenesis at mid-pregnancy, resulting in DNA damage. Given the spike in expression of pATR in late pregnancy (PD 18.5) and subsequent phosphorylation of its

**Fig. 3 | Physiological DNA damage occurs in the MG during pregnancy.**
**a** Representative image of γH2AX (green) in optically-cleared MG sections at PD10.5. White dashed lines indicate the single magnified alveolus that shows γH2AX in the luminal CK8$^+$ cells (magenta). Nuclear DNA by propidium iodide.
**b** Quantification of nuclear γH2AX in the luminal CK8$^+$ population from nulliparous, PD10.5, PD15.5, PD18.5 and LD2 MGs, as detected by IHC. **c** Representative image of 53BP1 (green) in optically-cleared MG sections at PD10.5. **d** Quantification of nuclear 53BP1 in the luminal CK8$^+$ population from nulliparous, PD10.5, PD15.5, PD18.5 and LD2 MGs, as detected by IHC. **e** Cartoon representing the contralateral intraductal injection (IDI) of MGs performed. Created with BioRender.com.
**f** Intraductal injection with trypan blue at PD17 showing efficient delivery of small molecules throughout the ductal tree. **g** Representative images of γH2AX (green) at PD13.5, 24 h after contralateral IDI with DMSO or Doxo. CK8 shown in magenta. Nuclear DNA by propidium iodide. **h** Quantification of nuclear γH2AX in the luminal CK8$^+$ population after contralateral IDI with DMSO or Doxo, as detected by IHC.
**i** Representative FACS DNA content analysis histograms from CK8$^+$ PD17.5 MGs after contralateral IDI with DMSO or Doxo. Red dashed lines show a histogram magnification corresponding to the 4C and >4C DNA content populations.
**j** Percentage of CK8$^+$ cells with 2C, 4C and >4C DNA content, as detected by FACS

analysis, after contralateral IDI with DMSO or Doxo in PD17.5 MGs. Colored data points and dashed lines represent paired samples. **k** Milk protein gene expression, *Lalba, Wap, Csn2, Plin2, Xdh1* and *Btn1*, in PD17.5 MGs after contralateral IDI with DMSO or Doxo, as detected by RT-qPCR. Colored data points and dashed lines represent paired samples. **l** Representative FACS histograms for DNA content analysis showing the sub-G1 population within the CK8$^+$ MG cells at PD17.5, after intraductal injections with DMSO or Doxo. **m** Percentage of sub-G1 cells, as detected by FACS analysis, within the CK8$^+$ MG population at PD17.5, after intraductal injections with DMSO or Doxo. Colored data points and dashed lines represent paired samples. For (**b, d**), data show values of 100 cells per timepoint from *n* = 3 combined biologically independent experiments (orange dots). Black dots represent the means of each *n* = 3 biologically independent experiments. For (**h**), data show values of 100 cells per treatment (gray or red dots). Data presented as mean ± SEM (**b, d, h, k**) and mean ± SD (**j, m**). Data analyzed by one-way ANOVA with Tukey's multiple comparison test (**b, d**), Mann–Whitney *u* test (**h**), and paired, two-tailed Student's *t* test (**j, k, m**). Data representative of *n* = 3 biologically independent experiments except (**i–m**), representative of *n* = 4. *p* values: *<0.05, **<0.01, ***<0.001, ****<0.0001.

downstream effector CHK1, it appears that ATR, and not ATM, is activated by DNA damage. Moreover, previous studies have shown that loss of *Atm* under *Wap-Cre* control allows for normal alveologenesis and lactogenesis, with structural and lactational defects only appearing at mid-lactation due to increasing cell death by apoptosis[47]. Taken together, these data suggest that the ATR-DDR pathway remains active until the beginning of lactation, when endoreplication occurs, and support the notion that the ATR-DNA damage response governs endoreplication during alveologenesis.

To investigate the role of replication stress on alveolar endoreplication, we induced it in HC11 cells prior to lactogenic differentiation by overexpressing CYCLIN E (*CCNE1*) (Supplementary Fig. 7a). CYCLIN E is an oncogene that accelerates DNA replication during the S phase of the cell cycle, resulting in DNA damage due to replication stress[48,49]. FACS analysis shows *CCNE1* increases DNA replication in HC11 cells as measured by BrdU incorporation (Supplementary Fig. 7b). In accordance with DNA damage accumulation due to replication stress, γH2AX and pATR also increase (Supplementary Fig. 7c–f). Furthermore, FACS DNA content analysis shows *CCNE1* overexpression increases endoreplication by DIP3, as well as CSN2, as detected by WB (Supplementary Fig. 7g–i). To investigate the effect of inducing replication stress in vivo, we performed contralateral intraductal injection (IDI) of hydroxyurea (Hu) or PBS vehicle into MGs at PD8.5 and PD12.5 (Fig. 5a), timepoints that encompass the peak of proliferation and physiological DNA damage at PD10.5 (Fig. 3a–d)[43]. Hu generates replication stress by inhibiting nucleoside synthesis, mimicking one of the likely sources of replication stress occurring in vivo during pregnancy[50]. Our data show that Hu IDI increases γH2AX and pATR in the CK8$^+$ population at PD13.5 by IHC staining (Fig. 5b–d). FACS analysis of DNA content in CK8$^+$ cells reveals an increase in the polyploid (>4C) population at LD2 (48% increase in the overall population) (Fig. 5e), while the diploid (2C) population decreases (5.5% decrease in the overall population) (Supplementary Fig. 8a, b). This increase in endoreplication is accompanied by an increase in milk protein genes *Wap, Csn2, Plin2* and *Btn1* as detected by RT-qPCR (Fig. 5f). Due to observed areas of high and low milk staining, we imaged entire sections (~1/2 of an abdominal MG; ~1cm$^2$ in area) of LD2 MG tissue by stitching together a minimum of 60 fields of view to generate a tiled image. We then quantified the milk staining contained within each alveolus of the tiled image and calculated the average integrated density among all alveoli in the section. We find a ~15% increase in milk in Hu-injected MGs (Fig. 5g–i), suggesting that DNA damage, produced by replication stress, is sufficient to drive endoreplication during lactation.

Next, we examined the consequences of relieving replication stress, by using nucleoside (Nuc) supplementation, which has been

shown to provide such relief in in vitro[49,51]. To investigate in vivo, we performed contralateral IDI of Nucs or PBS vehicle into MGs at PD8.5 and PD12.5 (Fig. 6a), again encompassing the peak of proliferation and physiological DNA damage (Fig. 3a–d). We find Nuc IDI decreases γH2AX and pATR in the CK8$^+$ population at PD13.5 by IHC staining (Fig. 6b–d). FACS analysis of DNA content in CK8$^+$ luminal cells also reveals a decrease in the tetraploid (4C) population at LD2 (22.15% decrease in the overall population), and both the tetraploid (4C) and polyploid (>4C) populations by LD5 (6.05% and 14.79% decrease in the overall populations, respectively) (Fig. 6e, f and Supplementary Fig. 8c–f). Accordingly, the diploid (2C) population increases at both LD2 and LD5 (9% and 49% increase in the overall population, respectively; Fig. 6e, f and Supplementary Fig. 8c–f). These results suggest Nuc IDI relieves replication stress during pregnancy and inhibits the activation of the G2/M checkpoint at LD2, resulting in decreased endoreplication by LD5. This decrease in endoreplication is accompanied by a decrease in milk protein genes at LD2 and LD5, and a 21% decrease in milk by LD5, as detected by RT-qPCR and IHC, respectively (Fig. 6g–j and Supplementary Fig. 8g). Altogether, these findings strongly suggest replication stress during early alveologenesis causes DNA damage and leads to prolonged DDR. This, in turn, triggers the activation of the G2/M checkpoint and endoreplication of alveolar cells at the onset of lactation.

**The DNA damage response regulates endoreplication via WEE1**
CDK1 inactivation during G2/M arrest can occur through several different inhibitors. It is primarily the Cip and Kip family of CDK inhibitors, composed of P21$^{Cip1}$, P27$^{Kip1}$, and P57$^{Kip2}$, that are implicated in endoreplication[29]. In addition, the CDK1 inhibitor WEE1 is required for proper DNA replication and for the activation of the G2/M checkpoint in response to replication stress[52–54]. WEE1 has also been shown to regulate endoreplication in plants[55–57]. To determine which of these inhibitors may be regulating CDK1 activity during alveologenesis, we analyzed their expression during pregnancy and lactation by RT-qPCR. We find *Cdkn1a* and *Cdkn1b*, which encode for P21$^{Cip1}$ and P27$^{Kip1}$, respectively, remain unchanged in comparison to their expression in the nulliparous MG (Supplementary Fig. 9a, b). In contrast, *Wee1* is upregulated during the proliferative period occurring in early pregnancy, and again at LD2, when endoreplication is occurring (Fig. 7a). RT-qPCR of FACS-purified MG populations demonstrates *Wee1* upregulation during lactation occurs specifically in the luminal population (Fig. 7b). Western blotting of whole MG lysates confirms the upregulation of WEE1 at lactation onset, occurring concomitantly with the upregulation of CDK1's inhibitory phosphorylation at Tyrosine-15 (Fig. 7c, d). Furthermore, we find *Wee1* is upregulated in response to IDI

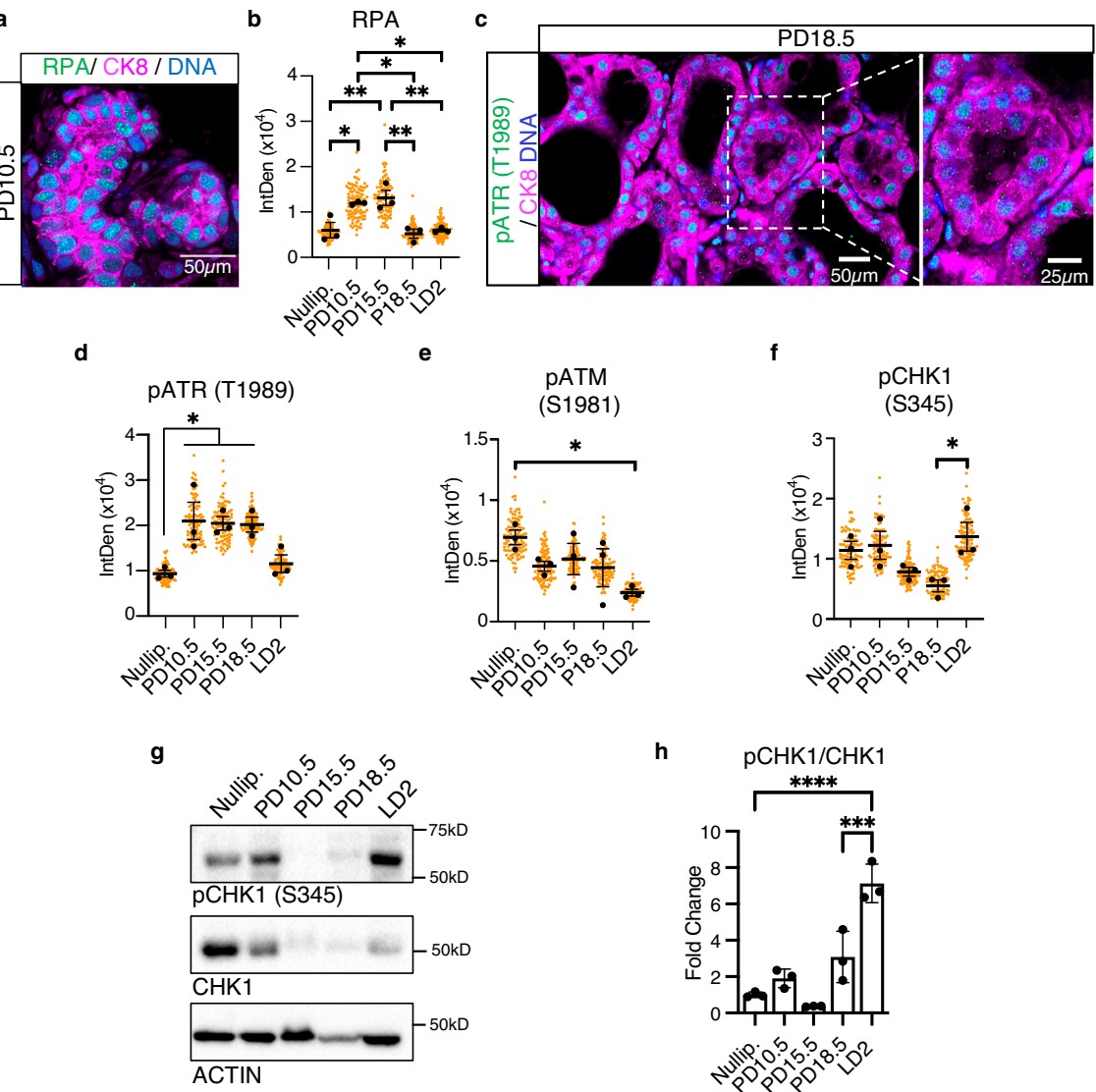

**Fig. 4 | DNA damage increases endoreplication and milk production in vivo.**
**a** Representative image of RPA (green) in optically-cleared MG sections at PD10.5.
**b** Quantification of nuclear RPA in the luminal CK8⁺ population from nulliparous,
PD10.5, PD15.5, PD18.5 and LD2 MGs, as detected by IHC. **c** Representative image of
pATR (green) in optically-cleared MG sections at PD18.5. White dashed lines indi-
cate the single magnified alveolus that shows pATR in the luminal CK8⁺ cells
(magenta). **d** Quantification of nuclear pATR in the luminal CK8⁺ population from
nulliparous, PD10.5, PD15.5, PD18.5 and LD2 MGs, as detected by IHC.
**e, f** Quantifications of nuclear pATM (**e**) and pCHK1 (**f**) in the luminal CK8⁺ popu-
lation from nulliparous, PD10.5, PD15.5, PD18.5 and LD2 MGs, as detected by IHC.

**g, h** Representative western blot (**g**) and quantification (**h**) of pCHK1 and CHK1
expression in nulliparous, PD10.5, PD15.5, PD18.5 and LD2 MGs. Quantification (**h**) is
shown as pCHK1/CHK1 ratio. For (**b, d–f**), data show values of 100 cells per time-
point from $n = 3$ combined biologically independent experiments (orange dots).
Black dots represent the means of each $n = 3$ biologically independent experiments.
Data presented as mean ± SEM (**b, d–f**) and mean ± SD (**h**). Data analyzed by one-
way ANOVA with Tukey's multiple comparison test. Data representative of $n = 3$
biologically independent experiments. $p$ values: *<0.05, **<0.01,
***<0.001, ****<0.0001.

of Doxo or Hu into the pregnant MG and downregulated in response to
IDI of Nucs (Supplementary Fig. 9c), demonstrating a direct correla-
tion between *Wee1* expression and the extent of endoreplication dur-
ing alveologenesis.

To further investigate the role of WEE1 during alveolar endor-
eplication, we turned to an MG organoid ex vivo model. First, to test
whether MG organoids recapitulate our findings in vitro and in vivo, we
treated them with Doxo, Ro-3306, Hu or Nucs then induced lactogenic
differentiation by culturing them in media containing insulin, prolactin
and dexamethasone. After 3 days, we find that Doxo, Ro-3306 or Hu
increases endoreplication (4C and >4C populations) and expression of
milk protein genes *Csn2*, *Wap* and *Lalba* (Fig. 7e, f). In contrast, Nuc
treatment decreases both endoreplication and milk protein gene
expression (Fig. 7e, f). These results demonstrate that MG organoids

fully recapitulate our data and are therefore a suitable ex vivo model
for alveolar endoreplication. Hence, we decided to treat them with the
WEE1 inhibitor Mk-1775, finding that WEE1 inhibition decreases
endoreplication (4C and >4C populations) and milk protein gene
expression (Fig. 7e, f). Next, we performed contralateral IDI of Mk-1775
or PBS vehicle into MGs at PD16.5 to capture the onset of endor-
eplication occurring by PD17.5 (Fig. 7g). FACS analysis of DNA content
in CK8⁺ luminal cells reveals that Mk-1775 IDI decreases the tetraploid
(4C) population at LD2 (19.4% decrease in the overall population), and
both the tetraploid (4C) and polyploid (>4C) populations by LD5 (3.2%
and 16.31% decrease in the overall populations, respectively) (Fig. 7h
and Supplementary Fig. 9d–g). Accordingly, the diploid (2C) popula-
tion increases at both LD2 and LD5 (7.7% and 12.4% increase in the
overall population, respectively) (Supplementary Fig. 9d–g). This

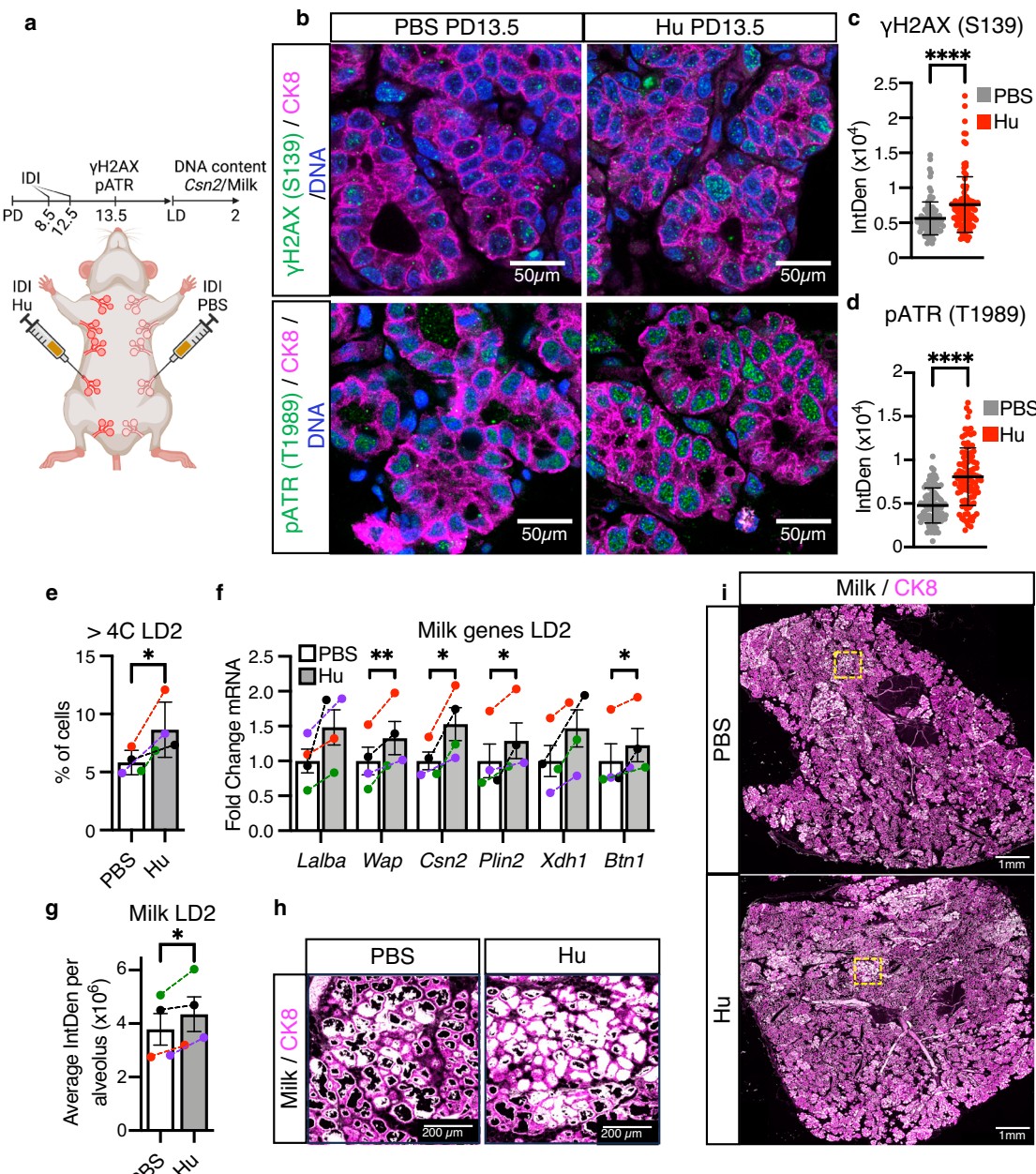

**Fig. 5 | Hydroxyurea induces replication stress and increases endoreplication and milk production in vivo. a** Cartoon representing the contralateral intraductal injection (IDI) of MGs performed. Created with BioRender.com. **b** Representative images of γH2AX (green, top) or pATR (green, bottom) in optically-cleared MG sections after contralateral IDI with PBS or hydroxyurea (Hu). CK8 shown in magenta. Nuclear DNA by propidium iodide. **c, d** Quantifications of nuclear γH2AX (**c**) and pATR (**d**) in the luminal CK8⁺ population after contralateral IDI with PBS or Hu, as detected by IHC. **e** Percentage of CK8⁺ cells with >4C DNA content, as detected by FACS analysis, after contralateral IDI with PBS or Hu in LD2 MGs. Colored data points and dashed lines represent paired samples. **f** Milk protein gene expression, *Lalba, Wap, Csn2, Plin2, Xdh1* and *Btn1*, in LD2 MGs after contralateral IDI with PBS or Hu, as detected by RT-qPCR. Colored data points and dashed lines

represent paired samples. **g** Quantification of milk per alveolus detected in LD2 MG alveoli after contralateral IDI with PBS or Hu, as detected by IHC. Colored data points and dashed lines represent paired samples. **h, i** Representative images of milk (white) in LD2 MGs after contralateral IDI with PBS or Hu. CK8 shown in magenta. Yellow squares indicate the magnified areas shown in (**h**), illustrating the milk contained within the alveoli. For (**c, d**), data show values of 100 cells per treatment (gray or red dots). Data presented as mean ± SEM (**c, d, f, g**) and mean ± SD (**e**). Data analyzed by Mann–Whitney *u* test (**c, d**) and paired, two-tailed Student's *t* test (**e–g**). Data representative of *n* = 4 biologically independent experiments except for (**b–d**), representative of *n* = 3. *p* values: *<0.05, **<0.01, ****<0.0001.

decrease in endoreplication is accompanied by a decrease of milk protein gene expression by LD2 and LD5 and a 21% decrease in milk production by LD5, detected by RT-qPCR and IHC, respectively (Fig. 7i–l and Supplementary Fig. 9h).

Next, we generated a *Wee1* conditional knock-out mouse line to delete the gene specifically in luminal cells. To evaluate whether the *Ck8* promoter is a suitable driver for conditional deletion, we

performed FACS analysis and identify three subpopulations of luminal cells based on their CK8 expression and side scatter (SSC): CK8^low; SSC^low, CK8^high; SSC^low and CK8^high; SSC^high (Supplementary Fig. 10a). Overall, we observe CK8 expression slightly decreasing from PD17.5-LD5 (Supplementary Fig. 10b), corresponding to a decrease in the CK8^high; SSC^high population even as the CK8^low; SSC^low population increases (Supplementary Fig. 10c). DNA content analysis of these

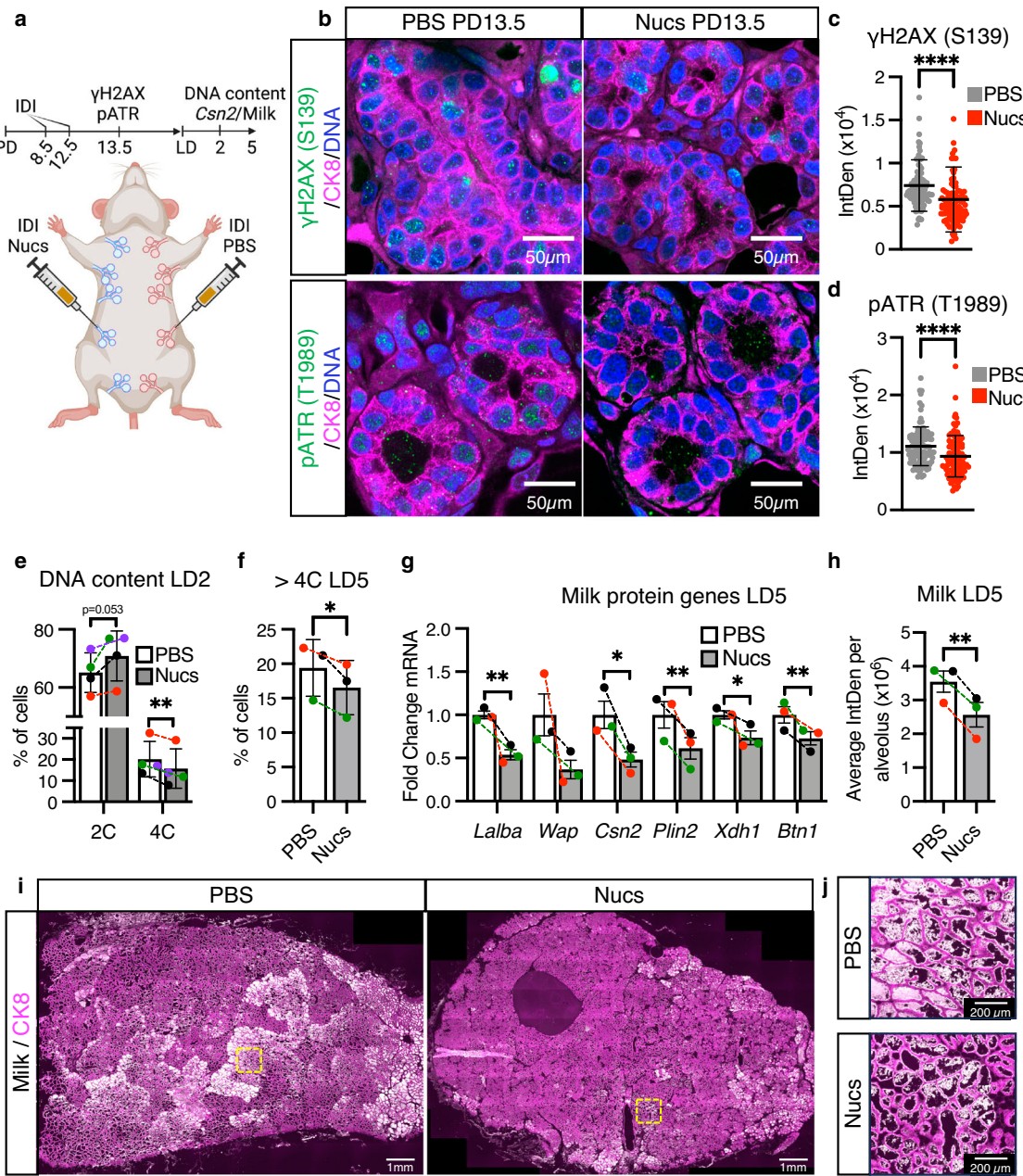

**Fig. 6 | Nucleosides supplementation relieves replication stress and inhibits endoreplication and milk production. a** Cartoon representing the contralateral intraductal injection (IDI) of MGs performed. Created with BioRender.com. **b** Representative image γH2AX (green, top) or pATR (green, bottom) in optically-cleared MG sections after contralateral IDI with PBS or nucleosides (Nucs). CK8 is shown in magenta. Nuclear DNA by propidium iodide. **c, d** Quantifications of nuclear γH2AX (**c**) and pATR (**d**) in the luminal CK8⁺ population after contralateral IDI with PBS or Nucs, as detected by IHC. **e** Percentage of CK8⁺ cells with 2 C and 4 C DNA content, as detected by FACS analysis, after contralateral IDI with PBS or Nucs in LD2 MGs. Colored data points and dashed lines represent paired samples. **f** Percentage of CK8⁺ cells with >4C DNA content, as detected by FACS analysis, after contralateral IDI with PBS or Nucs in LD5 MGs. Colored data points and dashed lines represent paired samples. **g** Milk protein gene expression, *Lalba, Wap, Csn2, Plin2, Xdh1*, and *Btn1*, in LD5 MGs after contralateral IDI with PBS or Nucs, as detected by RT-qPCR. Colored data points and dashed lines represent paired samples. **h** Quantification of milk per alveolus detected in LD5 MG alveoli after contralateral IDI with PBS or Nucs, as detected by IHC. Colored data points and dashed lines represent paired samples. **i, j** Representative images of milk (white) in LD5 MGs after contralateral IDI with PBS or Nucs. CK8 is shown in magenta. Yellow squares indicate the magnified areas shown in (**j**), illustrating the milk contained within the alveoli. For (**c, d**), data show values of 100 cells per treatment (gray or red dots). Data presented as mean ± SEM (**c, d, g, h**) and mean ± SD (**e, f**). Data analyzed by Mann–Whitney *u* test (**c, d**) and paired, two-tailed Student's *t* test (**e–h**). Data representative of *n* = 3 biologically independent experiments except for (**e**), representative of *n* = 4. *p* values: *p* values: *<0.05, **<0.01, ****<0.0001.

combined three subpopulations shows that the great majority of tetraploid (4C) and polyploid (>4C) luminal cells are CK8^high at LD5 (Supplementary Fig. 10d–g). In contrast, at PD17.5 most luminal cells are diploid (2C), regardless of the expression levels of CK8 (Supplementary Fig. 10d–f, h). Thus, because tetraploid and polyploid luminal cells express high levels of CK8, we used an inducible *CreER* system

under the control of the *Ck8* promoter that also carries a *mTmG* reporter. Induction of recombination was performed at PD17.5 to prevent potential deleterious effects caused by *Wee1* loss during early alveologenesis. FACS analysis of GFP expression at LD2 shows that recombination occurs in ~ 99% of CK8⁺ cells from both *Ck8-CreER;mTmG;Wee1^+/+* (*Wee1^+/+*) and *Ck8-CreER;mTmG;Wee1^fl/fl* (*Wee1^fl/fl*)

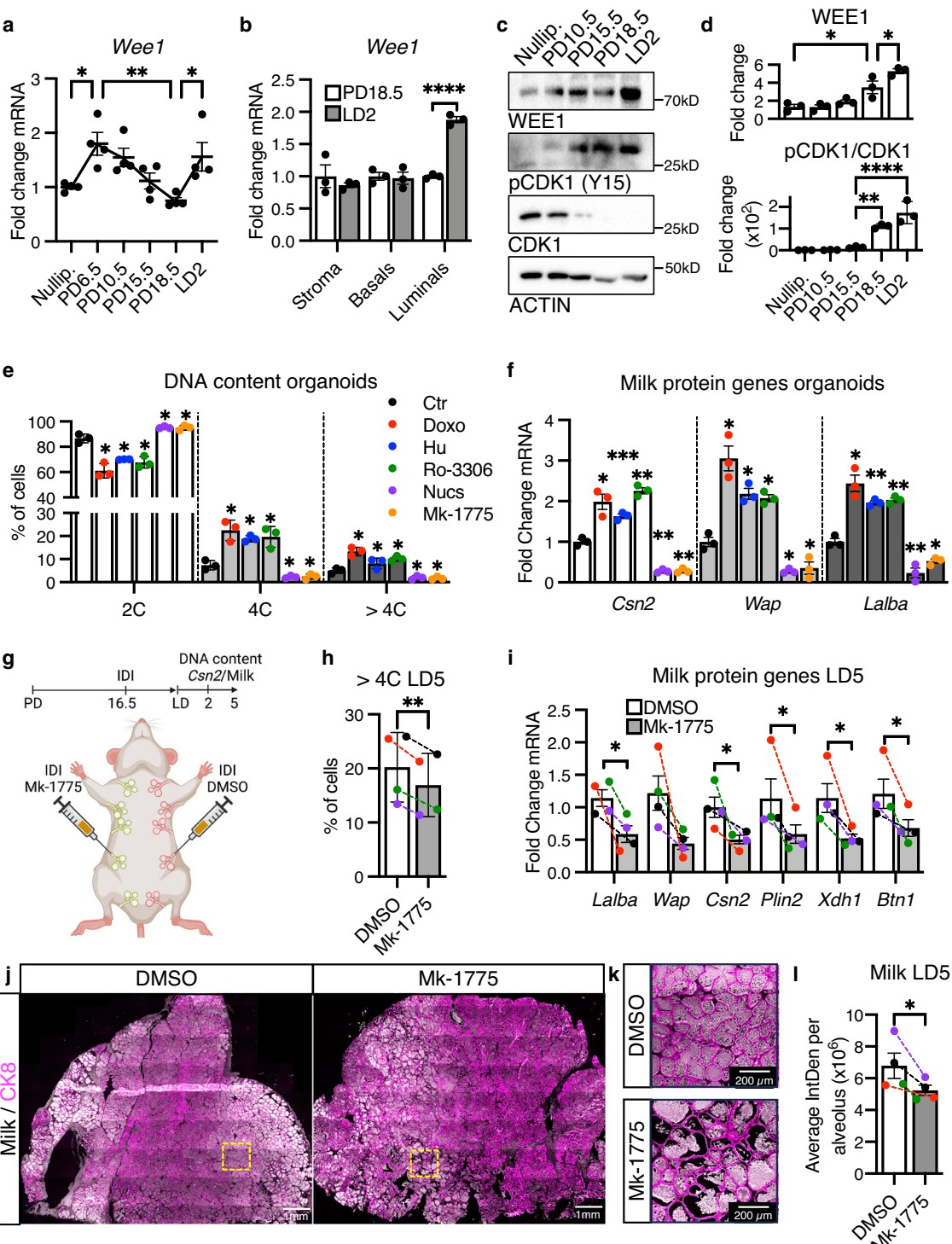

MGs (Supplementary Fig. 11a, b). As expected, in lysates from *Wee1^fl/fl* MGs, we detect decreased WEE1 expression and decreased inhibitory phosphorylation at Tyrosine-15 of WEE1 target CDK1 by western blot (Supplementary Fig. 11c–e).

To determine the effect of *Wee1* loss on milk production, we attempted to evaluate pup weight. When nursed by *Wee1^fl/fl* dams, however, pups do not display milk in their stomachs and do not survive past LD2 (Fig. 8a, b). Accordingly, at LD2, we find that *Wee1^fl/fl* MGs exhibit decreased expression of milk protein genes *Wap*, *Lalba*, *Csn2*, *Plin2*, *Xdh1* and *Btn1* and a ~50% decrease in milk produced as detected by RT-qPCR and IHC, respectively (Fig. 8c–f). The loss of the pups suggests they cannot suckle the little milk produced by the *Wee1^fl/fl*

alveolar structures. Indeed, the *Wee1^fl/fl* MGs are structurally deficient as they have ~55% fewer alveoli containing lumens, and those lumens are ~61% smaller (Fig. 8g, h). This structural deficiency results in an overall ~83% decrease in fat pad filling, as detected by IHC (Fig. 8i). We also find that the alveolar differentiation marker STAT5 and its active form, phosphorylated at Tyrosine-694, are decreased in *Wee1^fl/fl* MGs (Fig. 9a–c). In addition, the involution marker STAT3 is downregulated in *Wee1^fl/fl* MGs, while its active form, phosphorylated at Tyrosine-705, is upregulated (Supplementary Fig. 11f–h), likely the result of accelerated involution in *Wee1^fl/fl* MGs due to the loss of all pups by LD2. Despite these phenotypic differences, FACS analysis reveals the proportion of CK8+ luminal cells remains unchanged between *Wee1^+/+* and

**Fig. 7 | Pharmacological inhibition of WEE1 decreases endoreplication and milk production during lactation. a** *Wee1* expression in whole MG tissue from nulliparous, PD6.5, PD10.5, PD15.5, PD18.5 and LD2 MGs, as detected by RT-qPCR. *n* = 4. **b** *Wee1* expression in sorted luminal, basal and stromal populations from PD18.5 and LD2 MGs, as detected by RT-qPCR. Populations FACS-sorted based on CD24 and CD29 expression. **c, d** Representative western blot (**c**) and quantifications (**d**) of WEE1 (top), pCDK1 and CDK1 (bottom) expression in nulliparous, PD10.5, PD15.5, PD18.5, and LD2 MGs. Quantification (**d, bottom**) shown as pCDK1/CDK1 ratio. **e** Percentage of 2C, 4C and >4C CK8⁺ cells from MG organoids differentiated for 3 days, as detected by FACS analysis, after DMSO treatment (Ctr) or doxorubicin (Doxo), hydroxyurea (Hu), Ro-3306, nucleosides (Nucs) or Mk-1775. **f** Milk protein gene expression, *Csn2, Wap,* or *Lalba*, in MG organoids differentiated for 3 days after DMSO treatment (Ctr) or Doxo, Hu, Ro-3306, Nucs or Mk-1775. **g** Cartoon representing contralateral intraductal injection (IDI) of MGs performed. Created with BioRender.com. **h** Percentage of CK8⁺ cells with >4C DNA content, as detected by FACS analysis, after contralateral IDI with DMSO or MK-1775 in LD5 MGs. Colored data points and dashed lines represent paired samples. **i** Milk protein gene expression, *Lalba, Wap, Csn2, Plin2, Xdh1,* and *Btn1,* in LD5 MGs after contralateral IDI with DMSO or Mk-1775, as detected by RT-qPCR. Colored data points and dashed lines represent paired samples. **j, k** Representative images of milk (white) in LD5 MGs after contralateral IDI with DMSO or Mk-1775. CK8 shown in magenta. Yellow squares indicate magnified areas in (**k**), illustrating milk contained within alveoli. **l** Quantification of milk per alveolus after contralateral IDI with DMSO or Mk-1775, as detected by IHC. Colored data points and dashed lines represent paired samples. Data presented as mean ± SEM (**a, b, i, l**) and mean ± SD (**d, e, f, h**). Data analyzed by one-way ANOVA with Tukey's multiple comparison test (**a, d**) and paired, two-tailed Student's *t* test (**b, e, f, h, i, l**). Data representative of *n* = 4 biologically independent experiments except for (**b–f**), representative of *n* = 3. *p* values: *<0.05, **<0.01, ***<0.001, ****<0.0001.

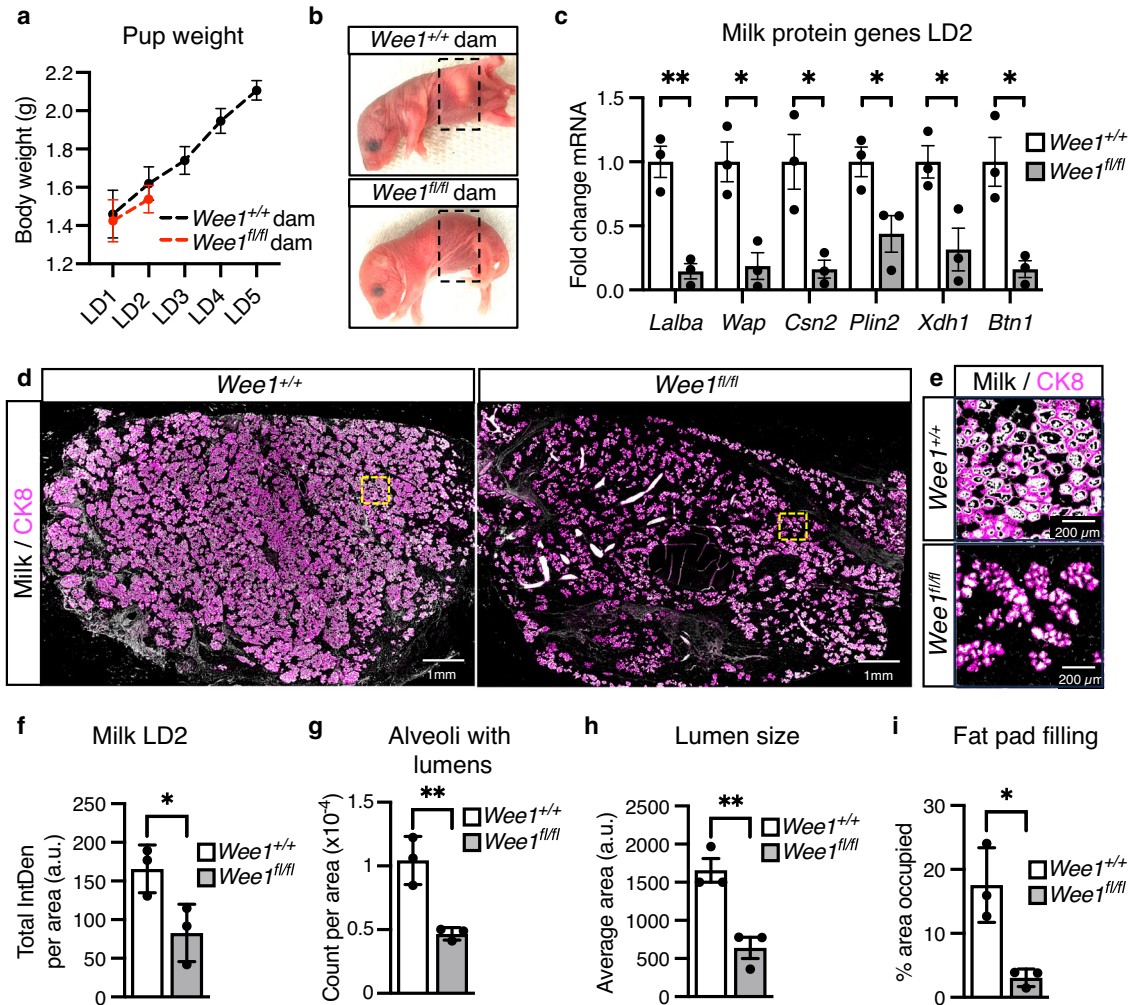

**Fig. 8 | WEE1 regulates milk production during lactation. a** Quantification of the body weight (g) of pups nursed by *Wee1⁺ᐟ⁺* or *Wee1ᶠˡᐟᶠˡ* dams. **b** Representative images of pups nursed by *Wee1⁺ᐟ⁺* (top) or *Wee1ᶠˡᐟᶠˡ* (bottom) dams. Black dashed-line squares indicate the stomach of the pups. **c** Milk protein gene expression, *Lalba, Wap, Csn2, Plin2, Xdh1* and *Btn1,* in *Wee1⁺ᐟ⁺* or *Wee1ᶠˡᐟᶠˡ* LD2 MGs, as detected by RT-qPCR. **d, e** Representative images of milk (white) by IHC in LD2 MGs from *Wee1⁺ᐟ⁺* or *Wee1ᶠˡᐟᶠˡ* mice. CK8 shown in magenta. Yellow squares indicate the magnified areas in (**e**), illustrating the milk contained within the alveoli. **f** Quantification of total milk per MG area in *Wee1⁺ᐟ⁺* and *Wee1ᶠˡᐟᶠˡ* mice at LD2, as detected by IHC. **g–i** Quantification of number of alveoli with lumens per MG area (**g**), alveoli lumen size (**h**), and fat pad filling (**i**) in LD2 MGs from *Wee1⁺ᐟ⁺* or *Wee1ᶠˡᐟᶠˡ* mice, as detected by IHC. For (**a**), data show the mean values of 5 pups per timepoint from each *n* = 3 biologically independent experiments. Data presented as mean ± SD (**a, f, g, i**) and mean ± SEM (**c, h**). Data analyzed by unpaired, two-tailed Student's *t* test. Data representative of *n* = 3 biologically independent experiments. *p* values: *<0.05, **<0.01.

*Wee1ᶠˡᐟᶠˡ* MGs (Fig. 9d). Nevertheless, *Wee1ᶠˡᐟᶠˡ* MGs show an overall ~40% decrease in CK8⁺ luminal cells with high scatter parameters at LD2, reflecting a decrease in cell size and complexity of the luminal population (Fig. 9e, f). Furthermore, DNA content analysis of CK8⁺ luminal cells from *Wee1ᶠˡᐟᶠˡ* MGs reveals a decrease in the tetraploid (4C) and polyploid (>4C) populations (~56% and 67% decrease in the overall population, respectively), while the diploid (2 C) population increases (Fig. 9g and Supplementary Fig. 11i, j). These data show that loss of

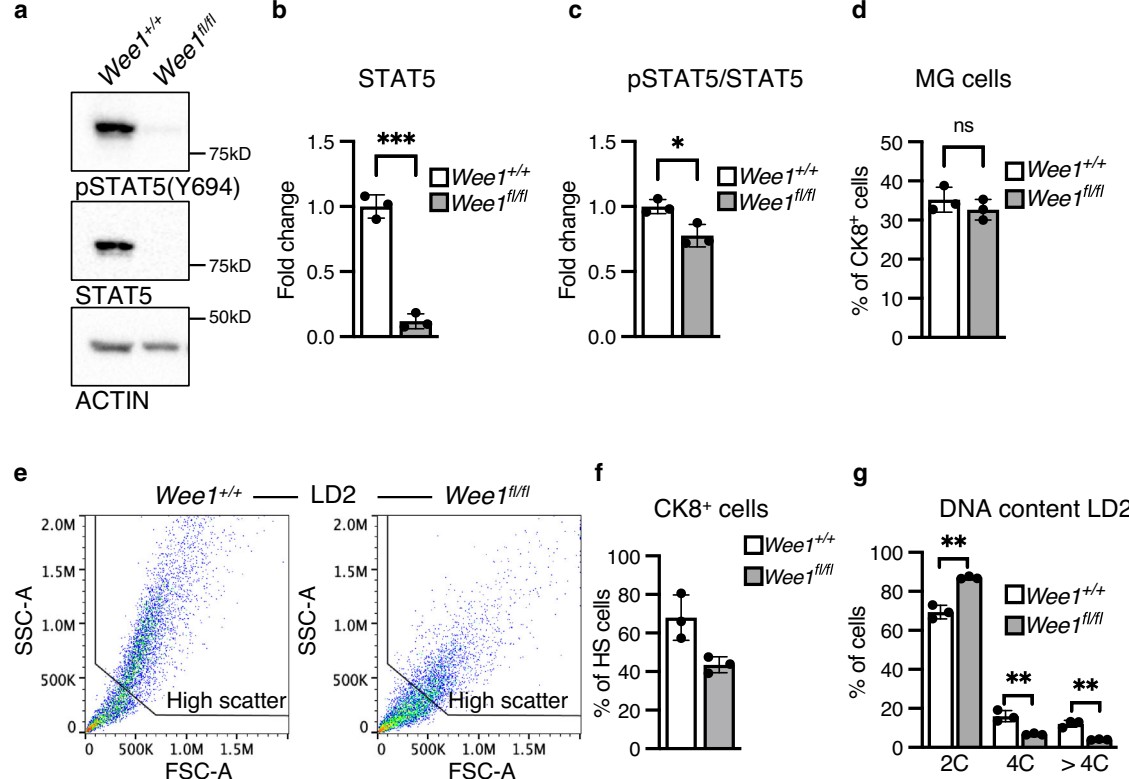

**Fig. 9 | WEE1 regulates alveolar differentiation and endoreplication during lactation. a–c** Representative western blot (**a**) and quantifications of STAT5 (**b**) and pSTAT5 (**c**) expression in *Wee1*⁺/⁺ or *Wee1*ᶠˡ/ᶠˡ LD2 MGs. Quantification (**c**) shown as pSTAT5/STAT5 ratio. **d** Percentage of CK8⁺ cells, as detected by FACS analysis, in *Wee1*⁺/⁺ or *Wee1*ᶠˡ/ᶠˡ LD2 MGs. **e** Representative dot plots showing the SSC-A and FSC-A parameters of the CK8⁺ population from *Wee1*⁺/⁺ or *Wee1*ᶠˡ/ᶠˡ LD2 MGs, as detected by FACS analysis. Red-to-blue color scale shows high-to-low density of data points. Black line shows the gating strategy for identifying cells with high scatter parameters. **f** Percentage of CK8⁺ cells from *Wee1*⁺/⁺ or *Wee1*ᶠˡ/ᶠˡ LD2 MGs with high scatter (HS) parameters, as detected by FACS analysis. **g** Percentage of CK8⁺ cells with 2C, 4C or >4C DNA content, as detected by FACS analysis, in *Wee1*⁺/⁺ or *Wee1*ᶠˡ/ᶠˡ LD2 MGs. Data presented as mean ± SD. Data analyzed by unpaired, two-tailed Student's *t* test. Data representative of *n* = 3 biologically independent experiments. *p* values: *<0.05, **<0.01, ***<0.001.

*Wee1* is sufficient to inhibit endoreplication, impede proper alveolar development and stop adequate milk production, which altogether prevent offspring survival. In sum, our results demonstrate that WEE1 mediates the DDR response to replication stress by activating the G2/M checkpoint and promoting alveolar endoreplication and milk production (Fig. 10).

## Discussion

Functional differentiation of alveolar cells is linked to endoreplication[8–11]. Endoreplication is thought to benefit tissues by (1) amplifying copy number for the more efficient production of RNA and proteins and (2) creating large cells that improve resistance to mechanical tension[58]: both important in the MG, which serves to produce milk and contracts in response of oxytocin. In addition, it has also been suggested that polyploid cells may be more susceptible for removal during involution, once breastfeeding is complete[11]. In this study, we show the pathway to polyploidization in the MG via induction of differentiation through the DDR is also a response that limits proliferation of damaged cells in a context where apoptosis or senescence would be deleterious to tissue integrity and function. Our results demonstrate endoreplication in the MG is achieved through an early mitotic arrest imposed by the activation of the DDR-G2/M checkpoint. Although this type of functional endoreplication resulting in cell differentiation has been shown to occur in response to genotoxic insults or during inflammatory processes[21,22,34,59], we show physiological DNA

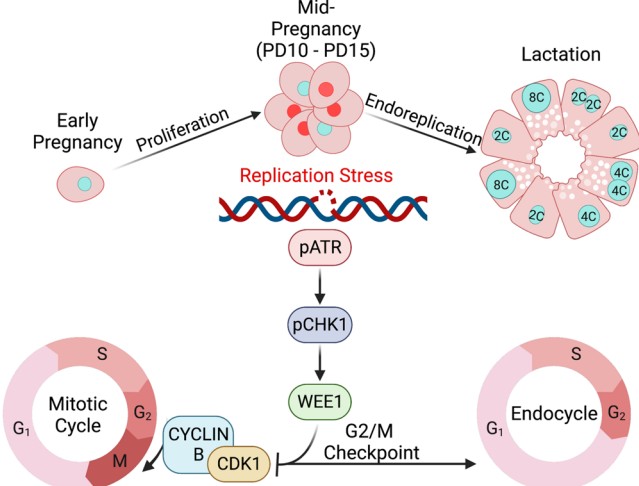

**Fig. 10 | Cartoon representing a model for the regulation of alveolar endoreplication by the ATR-DNA damage response pathway.** Proliferation of alveolar cells results in accumulation of DNA damage during pregnancy and activation of the ATR pathway. As a result, ATR mediates the transition from a mitotic cell cycle into an endocycle through the activation of the CDK1 inhibitor WEE1, which activates the G2/M checkpoint and regulates alveolar endoreplication during lactation. Created with BioRender.com.

damage, accumulated during the massive proliferation of early pregnancy, regulates alveolar functional endoreplication at the onset of lactation. This suggests DNA damage is not only an insult to genomic integrity and a potential source for carcinogenesis, but also an inevitable consequence of cell proliferation that can determine cell fate.

CDK1 inhibition is one of the key drivers of endoreplication, and various CDK1 inhibitors have been shown to regulate endoreplication in mammalian tissues[29]. Our study demonstrates CDK1 activity in the MG during alveologenesis is inhibited by WEE1, leading to activation of the G2/M checkpoint in response to DNA damage. Although WEE1 regulates endoreplication in some plants[55–57] and prevents apoptosis during keratinocyte endoreplication[21], here we show it also plays a central role in MG alveolar endoreplication. Why tissues achieve endoreplication through different CDK1 inhibitors, however, remains unclear. In the case of the MG, the role of WEE1 may be explained by its dual activities maintaining genome integrity and inhibiting CDK1[52,53]. On the one hand, WEE1 monitors DNA replication during proliferation by limiting replication initiation and exhaustion of nucleotide pools due to excessive origin firing[35,54]. On the other hand, it prevents proliferation by activating the G2/M checkpoint through the inhibitory phosphorylation of CDK1[53]. Accordingly, we observed two waves of WEE1 upregulation in the MG: the first occurring during early pregnancy and the second at the onset of lactation. This suggests WEE1 ensures proper DNA replication during the proliferative phase of alveologenesis and then induces endoreplication of damaged cells during the latter phase of differentiation.

While we investigated the role of WEE1 in driving endoreplication of mammary alveolar cells during functional differentiation, other studies have evaluated the role WEE1 in ensuring proper DNA replication during proliferative phases of mammary gland development[26,60]. It has been shown that loss of the KRAB zinc finger protein Roma/Zfp157 results in increased replication stress and WEE1 downregulation[26]. Polyploidization was also observed in the *Roma* knockout. This suggests that, in the presence of genomic instability triggered by the loss of *Roma* and downregulation but not complete loss of WEE1, luminal cells undergo endoreplication to limit uncontrolled proliferation of damaged cells. In a second study, mammary-specific heterozygous loss of *Wee1* in nulliparous mice resulted in increased DNA damage and dysregulation of mitotic progression, leading to the formation of luminal-type tumors after a long lag period[60]. Interestingly, the majority of the tumor cells were polyploid, suggesting that increased DNA damage due to loss of *Wee1* during proliferation drives the endoreplication of these tumorigenic luminal cells. Homozygous knockouts, however, did not develop tumors, and it was suggested that the profound genomic instability due to complete loss of *Wee1* leads to cellular senescence, as opposed to endoreplication[60]. Thus, a role has been established for WEE1 in inducing cellular polyploidization in response to DNA damage as a safeguard mechanism during proliferative phases of development and during tumorigenesis[26,60]. By contrast, our study demonstrates a second role for WEE1 in harnessing DNA damage that is produced by replication stress during the proliferative phase of pregnancy to both drive functional endoreplication during the differentiative phase and facilitate the efficient production of milk during lactation.

Breastfeeding provides a myriad of long-term advantages to both a mother and child[61,62]. Yet lactation insufficiency, defined as the inability of a nursing mother to produce the milk necessary for an infant's daily nutritional needs, is a global public health concern. Past endeavors to address this issue have primarily focused on manipulation of the prolactin pathway to enhance milk production. While some therapeutics have been approved, albeit not in the U.S.A., such as the dopamine antagonist domperidone, negative side effects prevent their widespread adoption[63]. Our model offers an explanation as to why mothers have a differential ability to build a milk supply during lactation. Our studies provide mechanistic insights into how DNA damage accumulation during massive proliferation couples cell generation and organ growth to efficient milk production. We further demonstrate that controlling CDK1 activity has the potential to mitigate lactation insufficiency by providing a non-hormonal means of targeting milk production.

## Methods

### Mouse strains
CD-1 mice were obtained from Charles Rivers. C57BL/6-Wee1<tm1.1 mrl> were generated by Taconic. Briefly, exons 3 to 5 were flanked by loxP sites with the targeting vector generated using C57BL/6J DNA and transfected into the TaconicArtemis C57BL/6NTac ES cell line. Flp-mediated recombination generated the conditional KO allele after which the Flp-transgene was removed by segregation. Ck8-CreER/mTmG mice, a cross of Tg(Krt8-cre/ERT2)17blpn/J (JAX:017947) and B6.129(Cg)-*Gt(ROSA)26Sor*^tm4(ACTB-tdTomato, -EGFP)Luo^/J (JAX: 007676), were generously provided by Dr. Diwakar R Pattabiraman. Genotyping was performed by extracting DNA from ear snips and performing an end-point PCR for the given transgene using the primers CreER: 5′-CAGATGGCGCGGCAACACC-3′ and 5′-GCGCGGTCTGGCAGTAAAAAC-3′; mTmG: 5′-AGG GAG CTG CAG TGG AGT AG-3′, 5′-TAG AGC TTG CGG AAC CCT TC-3′ and 5′-CTT TAA GCC TGC CCA GAA GA-3′; *Wee1*: 5-GCTTCGGAACCTTCCTAATGC-3′ and 5′-TGAAGTCTCACCCTGTCTCG-3. All animal procedures were both approved by and conducted in accordance with the guidelines set by the University of California, Santa Cruz (UCSC) Institutional Animal Care and Use Committee (IACUC).

### Animal studies
Nulliparous analysis was performed using adult (10-12-week-old) female mice. For timed pregnancies and lactation analysis, CD-1 pregnant adult females were obtained from Charles Rivers. Embryos were examined at the time of mammary gland harvesting to confirm pregnancy state. For lactation analysis on the C57BL/6-Wee1<tm1.1 mrl> mice, adult females were checked for the presence of a vaginal plug indicating that mating occurred. Plugged mice were considered PD0.5 on the day of the observed plug. All females used in this study were age matched. For pup weight analysis, litters were culled to 5 pups and weight measured every 24 h.

### Contralateral Intraductal Injections
Mice were anesthetized with isoflurane (VetOne, 501017) Prior to intraductal injections, hair was removed from around the nipples using Nair hair remover. Injections into the duct of the nipple were performed with 33 G needles (Hamilton, 7803-05) and a volume of 40 μl per gland. Right inguinal, abdominal and thoracic glands were injected with doxorubicin (1.5 μg per gland, Cayman Chemical, 15007), hydroxyurea (300 μg per gland, Sigma-Aldrich, H8627), nucleosides (40 μl per gland, Millipore-Sigma, ES-008-D) or Mk-1775 (50 μg per gland, Cayman Chemical, 21266), and left glands were injected with vehicle, either PBS or DMSO (Thermo Scientific, BP231) diluted in PBS.

### Tamoxifen Injections
Tamoxifen (Sigma, T5648) was dissolved in corn oil (Sigma, C8267) at a concentration of 20 mg/ml. Mice were anesthetized with isoflurane (VetOne, 501017). Intraperitoneal injections of 75 mg/kg bodyweight were performed at PD17.5 and LD2 using insulin syringes (Fisher Scientific, 14-826-79).

### Organoid cultures
Primary cell organoids were grown as previously described[64]. Primary cells were mixed and grown in Matrigel Growth Factor Reduced Phenol-Red Free (Corning, CB-40230C) and cultured in basal DMEM/F12 media (Thermo Fisher Scientific, 11039021) supplemented with 1x N-2 (Thermo Fisher Scientific, 17502048), 1x B-27 (Thermo Fisher Scientific, 12587010), 100 ng/ml Nrg1 (R&D, 5898-NR-050), 42.5 ng/ml

R-spondin (Peprotech, 120-38), 1 nM Rho-inhibitor (Y-27632; Cell Signalling, 13624S) and 10 ng/ml epidermal growth factor (EGF; Peprotech, AF-100-15). After 5 days, growing organoids were cultured in differentiation media [DMEM/F12 containing 1x N-2, 1x B-27, 100 ng/ml Nrg1, 42.5 ng/ml R-spondin, 1 nM Rho-inhibitor, 1 µg/ml prolactin (NHPP, oPRL-21), 5 µg/ml dexamethasone (Millipore-Sigma, D4902-1G) and 5 µg/ml insulin (Millipore-Sigma, I6634)] for additional 3 days.

Organoids were treated with 50 nM doxorubicin (Cayman Chemical, 15007), 100 uM hydroxyurea (Sigma, H8627), 1 µM Ro-3306 (Sigma Aldrich, SML0569), 150 nM Mk-1775 (Cayman Chemical, 21266), corresponding DMSO-containing vehicle (Thermo Scientific, BP231), or supplemented with 1x nucleosides (Millipore-Sigma, ES-008-D). Nucleosides supplementation began on the first day of growing in Matrigel, doxorubicin, hydroxyurea and Ro-3306 treatments began one day prior to differentiation, and Mk-1775 treatment began on the first day of differentiation. All treatments were maintained through the growing and/or differentiation process and replenished with every change of media. Organoids were removed from Matrigel incubating for 30 min with ice-cold Cell Recovery Solution (Corning, CB-40253). For FACS DNA content analysis, organoids were incubated for 5 min with pre-warmed Trypsin-EDTA (0.25%; Thermo Fisher Scientific, 25200072) to obtain a single cell suspension, and fixed with 70% ice-cold ethanol.

## Cell cultures

The HC11 cell line was obtained from American Type Culture Collection (ATCC) and routinely checked for mycoplasma (Mycoplasma PCR kit, ABM, G238). Undifferentiated HC11 cells were cultured in growing medium (RPMI-1640; Thermo Fisher Scientific, 72400047), supplemented with 10% FBS, 5 µg/ml insulin (Millipore-Sigma, I6634), 10 ng/ml epidermal growth factor (EGF; Peprotech, AF-100-15) and 1× AntiAnti (Thermo Fisher Scientific, 15240112) at 37 °C with 5% CO2. Cells were grown to confluency and maintained in growing medium for 2 days, until they became competent. Competent HC11 cells were primed for differentiation by culturing them in priming medium [RPMI-1640 supplemented with 5 µg/ml insulin, 1 µM dexamethasone (Millipore-Sigma, D4902-1G) and 1× Anti-Anti] for 24 h at 37 °C with 5% CO2. To induce differentiation, primed HC11 cells were cultured in DIP Medium [RPMI-1640, supplemented with 10% FBS, 5 µg/ml insulin, 1 µM dexamethasone, 1× anti-anti and 3 µg/ml Prolactin (NHPP, oPRL-21)] at 37 °C with 5% CO2.

For endoreplication studies, 80% confluent undifferentiated HC11 were treated with 30 µM blebbistatin (Millipore-Sigma, B0560), 5 µM Ro-3306 (Sigma Aldrich, SML0569), 100 nM doxorubicin (Cayman Chemical, 15007) or corresponding DMSO (Thermo Scientific, BP231) control. Drugs were maintained through the differentiation process and added with every change of media.

Undifferentiated HC11 were transfected with the Rc/CMV cyclin E plasmid (Addgene, #8963) for *CCNE1* overexpression. Briefly, 70% confluent HC11 were transfected using Lipofectamine 3000 kit (Thermo Fisher, L3000015) and OPTI-MEM (GIBCO, 11058021). 48 h after transfection, HC11 were selected using 50 µg/ml of geneticin (Thermo Fisher, 10131035). Geneticin was maintained during HC11 culture and differentiation for stable *CCNE1* expression.

## BrdU incorporation analysis

Undifferentiated HC11 was treated with 10 µM BrdU (Abcam, ab142567) for 2 h at 37 °C with 5% CO2. Cells were washed with 1X DPBS (GIBCO, 14190-250) and harvested using 0.05% Trypsin-EDTA (GIBCO, 25300-062). Cell suspension was washed with 1X DPBS and centrifuged at 1000 rpm for 5 min at 4 °C. Cell pellet was fixed in ice-cold 70% EtOH vortexing vigorously to avoid cell clumps. After fixation cells were washed with washing buffer [1X PBS (GIBCO, 14190136) containing 5% FBS] and centrifuged at 2,000 rpm for 5 min at 4 °C. Cell pellet was treated with 500 µl of 2 M HCl for 20 min at room temperature. Cells were washed with 2 ml of 0.1 M sodium tetraborate (Sigma Aldrich,

221731) and centrifuged at 2,000 rpm for 5 min at 4 °C. Cells were washed once more with 3 ml of 0.1 M sodium tetraborate and one last time with 2 ml of washing buffer. Cells were incubated for 1 h at room temperature with anti-BrdU (Abcam, ab6326; undiluted) or corresponding rat IgG (Thermo-Fisher, 10700) isotype control at equivalent concentration. Cells were washed twice with 2 ml of washing buffer and incubated for 1 h at room temperature in darkness with FITC anti-Rat (Thermo Fisher, A24544; 1:100). After incubation, cells were washed twice and resuspended in 500 µl of 1X PBS. Cells suspensions were filter through a 70 µm cell strainer (Falcon, 08-771-2) and analyzed using a BD LSRII cytometer. Populations were analyzed using FlowJo.

## Western blotting

Whole cell or mammary gland lysates were prepared using 1× NP40 lysis buffer (Thermo Fisher Scientific, FNN0021) supplemented with Pierce Protease and Phosphatase inhibitors (Thermo Fisher Scientific, A32959). Cells were washed with ice-cold PBS (GIBCO, 14190136) and lysed direct in buffer and kept at 4 °C rocking at 70 rpm for 30 min. Lysed cells were collected and centrifuged at 12,000 rpm at 4 °C for 15 min. Protein concentration was quantified using Qubit 4 fluorometer (Thermo Fisher, Q33238). Samples were resolved by SDS page and transferred to polyvinylidene difluoride (PVDF, Millipore-Sigma, IPVH00010) for 60 min at 250 mA. Immunoblots were blocked for 1 h at room temperature using either 5% non-fat milk or 5% BSA TBST. Primary antibodies [anti-GAPDH (SCBT, sc-365062; 1:1000), anti-Actin (SCBT, sc-47778; 1:5000), anti-HSP70 (SCBT, sc-24; 1:1000), anti-Cyclin B1 (SCBT, sc-245; 1:1000), anti-Cyclin E1 (Millipore-Sigma, SAB4503516; 1:1000) and anti-CSN2 (ABclonal, A12749; 1:1000), anti-pATR (GeneTex, GT128145; 1:500), anti-ATR (Cell Signaling, 2790 S; 1:500), anti-pATM (GeneTex, GTX132146; 1:1000), anti-ATM (GeneTex, GTX70103; 1:500), anti-pCHK1 (Cell Signaling, 23485; 1:1000), anti-CHK1 (SCBT, sc-8408; 1:1000), anti-pCdc2 (Cell Signaling, 10A11; 1:1000), anti-Cdc2 (SCBT, sc-454; 1:1000), anti-WEE1 (Abnova, H00007465-M01A; 1:1000, anti-WEE1 (Thermofisher, PA5-29303; 1:500), anti-pSTAT5 (Cell Signaling, 9351 S; 1:1000), anti-STAT5 (SCBT, sc-836; 1:1000), anti-pSTAT3 (Cell Signaling, 9145 S; 1:1000), and anti-STAT3 (Cell Signaling, 9139 S; 1:1000)] were incubated overnight at 4 °C in a rocker. HRP-conjugated secondary antibodies (The Jackson Laboratory) were used for 1 h at room temperature. Immunoblots were developed using Clarity ECL (Bio-Rad), detected using a Bio-Rad ChemiDoc MP Image, and quantified using ImageJ.

## Mammary gland single-cell suspension

Mechanically dissociated inguinal, abdominal, and thoracic mammary fat pads were prepared into cell suspension for flow cytometry or fluorescence-activated cell sorting (FACS). The lymph node was removed from abdominal glands. Glands were chopped using a mechanical tissue chopper and digested for 1 h at 37 °C in digestion media [RPM1 containing 1%FBS, collagenase IA (Sigma, C9891), hyaluronidase (Sigma, H3506) and DNAse I (Worthington, LS002007)]. Tissue was washed with washing buffer (1X PBS containing 2% FBS) and centrifuged at 1,000 rpm for 5 min at 4 °C. Tissue was further digested using pre-warmed 0.25% Trypsin-EDTA (Thermo Fisher, 25200056), washed, and digested with 5 mg/ml of pre-warmed dispase II (Roche, 4942078001). Red blood cells were lysed using Ammonium Chloride Solution (Stem Cell Technologies, 07850). Cells were washed, resuspended and filter through a 70 µm cell strainer (Falcon, 08-771-2) and processed for downstream applications.

## Flow cytometry

For DNA content analysis of the mammary gland or organoid CK8[+] epithelial population, cells suspension was obtained as described above. Cells were fixed in ice-cold 70% EtOH at a final concentration of $10^6$ cells/ml. During fixation, cells were vigorously vortexed for 1 min to avoid the formation of cell aggregates. Cells were washed twice with

washing buffer [1X PBS containing 5% FBS and 0.5% tween 20 (Fisher Chemical, BP337500)] and centrifuged at 2000 rpm for 5 min at 4 °C. Cell pellet was incubated for 1 h at room temperature with anti-CK8 (Developmental Studies Hybridoma Lab, TROMA-1; 1:250), anti-GFP (Thermo Fisher, A01704; 1:250) or corresponding rat (Thermo-Fisher, 10700) or rabbit (Thermo-Fisher, 10500 C) IgG isotype controls at equivalent concentration. Cells were washed twice and incubated for 1 h at room temperature in darkness with FITC anti-Rat (Thermo Fisher, A24544; 1:100) or FITC anti-Rabbit (Thermo Fisher, A16030; 1:100) and APC anti-Rat (Jackson ImmunoResearch, 712-136-153; 1:100). Cells were washed twice and resuspended in propidium iodide solution [1X PBS containing 25 μg/ml of propidium iodide (Thermo Fisher, P3566) and 100 μg/ml of RNAse (Thermo Fisher, 12091021)]. For DNA content analysis on HC11, cells were washed with 1X DPBS (GIBCO, 14190-250) and harvested using 0.05% Trypsin-EDTA (GIBCO, 25300-062). Cell suspension was washed with 1X DPBS and centrifuged at 1,000 rpm for 5 min at 4 °C. Cell pellet was fixed in ice-cold 70% EtOH and then vortexed vigorously to avoid cell aggregates. After fixation, cells were washed with washing buffer [1X PBS (GIBCO, 14190136) supplemented with 5% FBS] and centrifuged at 2000 rpm for 5 min at 4 °C. The pellet was resuspended in propidium iodide solution. Cell suspensions were filtered through a 70 μm cell strainer (Falcon, 08-771-2) and analyzed using a BD LSRII cytometer or a BD FACS Aria II Cell Sorter. Populations were analyzed using FlowJo.

**Visualization of cells sorted based on DNA content**
Mammary gland CK8⁺ epithelial cells or HC11 were sorted based on DNA content using BD FACS Aria II Cell Sorter. After sorting, cells were stained in suspension using Phalloidin-iFluor 488 Reagent (Abcam, ab176753) for 30 min at room temperature in darkness. Cells were spined down onto microscopy slides (Fisher, 12-550-15) using Cytospin 2 (Shandon, 599 × 52) at 500 rpm for 3 min. Cells were mounted using fluoromount-G (Southern Biotech, 0100-01) and visualized using Zeiss Axio Imager Microscope.

**Immunofluorescence on HC11 cells**
HC11 cells were fixed using ice-cold MeOH for 10 min. After washing with 1X PBS, cells were incubated for 1 h at room temperature with primary antibodies [anti-γH2AX (SCBT, sc-517348; 1:100), anti-pATR (Genetex, GTX128145; 1:100), anti-PLIN2 (generously provided by Jim McManaman; 1:100)] in a humid incubation chamber. After incubation, cells were washed three times using 1X PBS and incubated for 1 h at room temperature in darkness using corresponding secondary antibodies [donkey anti-rabbit 488 (Jackson ImmunoResearch, 711-546-152; 1:100), donkey anti-rabbit 647 (Jackson ImmunoResearch, 711-606-152; 1:100), donkey anti-mouse 488 (Jackson ImmunoResearch, 715-546-150; 1:100) and/or donkey anti-mouse 647 (Jackson ImmunoResearch, 715-606-150; 1:100)] and Phalloidin-iFluor 488 Reagent (Abcam, ab176753; 1:1000) when indicated. Cells were washed three times using 1X PBS and incubated with Hoechst 33342 (AnaSpec, AS-83218; 1:1000) for 10 min. Cells were mounted using fluoromount-G (Southern Biotech, 0100-01) and visualized using Zeiss Axio Imager Microscope. The integrated density of PLIN2, nuclear γH2AX and nuclear pATR was quantified using ImageJ.

**Immunofluorescence of paraffin-embedded tissue**
Mammary gland tissue was fixed in 10% neutral buffered formalin (EMD Millipore, MR0458682) at 4 °C overnight. Fixation was quenched using 0.2% glycine (Fisher Scientific, BP381) in PBS, for 1 h at room temperature. Tissue was dehydrated by incubating with 70% EtOH (Decon Labs, V1001) overnight, 95% EtOH for 1 h, 100% EtOH for 1 h (x3) and xylenes (Fisher Scientific, X3P) for 1 h (x3). Dehydrated tissue was soaked in paraffin (VWR, 15159-409) overnight and embedded. Paraffin-embedded tissue was sectioned at a thickness of 5 μm and mounted on Superfrost Plus Microscope Slides (Fisher, 12-550-15).

Sectioned tissue was hydrated by incubating with xylenes for 5 min (x3), 100% ethanol for 2 min (x2), 95% ethanol for 1 min, 70% ethanol for 1 min, 50% ethanol for 1 min, and diH2O for 5 min. Antigen retrieval was performed using antigen unmasking solution (VectorLabs, H3300-250) in a conventional lab microwave. Sections were incubated with blocking buffer containing 10% donkey serum (Equitech-Bio, SD30), 1% BSA (VWR, 97061-422) and 0.3% triton (Millipore Sigma, X100) in PBS overnight at 4 °C. Incubation with primary antibodies [anti-CK8 (Developmental Studies Hybridoma Lab, TROMA-1; 1:500) and anti-mouse milk proteins (Accurate Chemical and Scientific, YNRMTM; 1:500)] was performed overnight at 4 °C. Sections were washed with 0.3% triton in PBS for 30 min (x3) at room temperature. Incubation with secondary antibodies [(donkey anti-Rat 647 (Thermo-Invitrogen; A48272; 1:500) and donkey anti-Rabbit 488 (Thermo-Invitrogen; A32790; 1:500)] was performed for 2 h at room temperature. Sections were washed with 0.3% triton in PBS for 30 min (x3) at room temperature and mounted using fluoromount-G (Southern Biotech, 0100-01). Image acquisition was performed using Zeiss Axio Imager Microscope. Entire sections (~1/2 of an abdominal MG; ~1cm² in area) of MG tissue were imaged by stitching together a minimum of 60 fields of view to create a tiled image. Image analysis was performed using ImageJ. Individual alveolar lumens were segmented by filtering, thresholding and inverting the channel comprising the luminal cell marker CK8. The resulting binary image was used as a mask to quantify the integrated density of the milk staining within each alveolus of the tiled image. Masked objects too small or large to be alveolar lumens were excluded from analysis.

**Immunofluorescence and optical clearing of cryosections**
Mammary gland tissue was fixed in 10% neutral buffered formalin (EMD Millipore, MR0458682) at 4 °C overnight. Fixation was quenched using 0.2% glycine (Fisher Scientific, BP381) in PBS, for 1 h at room temperature. Tissue was incubated with 30% sucrose (Fisher Scientific, BP220-212) in PBS at 4 °C for 48 h and sectioned at a thickness of 100 μm. Sections were washed with PBS for 10 min (x2) at room temperature. Sections were incubated with CUBIC-L[65] at 37 C overnight, and washed with 0.3% triton in PBS for 30 min (x3). Sections were incubated with blocking buffer containing 10% donkey serum (Equitech-Bio, SD30), 1% BSA (VWR, 97061-422) and 0.3% triton (Millipore Sigma, X100) in PBS overnight at 4 °C. Incubation with primary antibodies [anti-γH2AX (Cell Signaling, 2577 S; 1:500), anti-pATR (Genetex, GTX128145; 1:500), anti-pCHK1 (Cell Signaling, 2348; 1:500), anti-pATM (Genetex, GTX132146; 1:500), anti-53BP1 (Cell Signaling, 4937 S; 1:500), anti-RPA (Abcam, ab76420; 1:500), anti-CK8 (Developmental Studies Hybridoma Lab, TROMA-1; 1:500)] was performed overnight at 4 °C. Sections were washed with 0.3% triton in PBS for 1 h (x3) at room temperature. Incubation with secondary antibodies [donkey anti-Rat 488 (Thermo-Invitrogen; A32795; 1:500) and donkey anti-Rabbit 647 (Thermo-Invitrogen; A48269; 1:500)], propidium iodide (Thermo Fisher, P3566) and RNAse (Thermo Fisher, 12091021) was performed for 6 h at room temperature. Sections were washed with 0.3% triton in PBS for 1 h (x3) at room temperature and mounted on poly-L-lysine (Sigma, P8920) coated chamber slides (Ibidi, 80827). Tissues sections were incubated with CUBIC-R[65] at room temperature until cleared (approximately 48 h), and imaged using a ZEISS LSM 880 microscope with Airyscan. Image analysis was performed using ImageJ. Individual luminal cell nuclei were manually identified and segmented by free-hand selection using the channels comprising the luminal cell marker CK8 and the nuclear dye propidium iodide, and the integrated density of nuclear γH2AX, pATR, pCHK1, pATM, 53BP1 or RPA was quantified per nuclei.

**3D DNA content in situ**
Mammary gland tissue was fixed in 10% neutral buffered formalin (EMD Millipore, MR0458682) at 4 °C overnight. Fixation was quenched

using 0.2% glycine (Fisher Scientific, BP381) in PBS, for 1 h at room temperature. Tissue was incubated with 30% sucrose (Fisher Scientific, BP220-212) in PBS at 4 °C for 48 h and sectioned at a thickness of 200 μm. Sections were washed with PBS for 10 min (x2) at room temperature. Sections were incubated with blocking buffer containing 10% donkey serum (Equitech-Bio, SD30), 1% BSA (VWR, 97061-422) and 0.3% triton (Millipore Sigma, X100) in PBS overnight at 4 °C. Incubation with primary antibody anti-E-cadherin (Thermo Fisher, 13-1900; 1:500) was performed overnight at 4 °C. Sections were washed with 0.3% triton in PBS for 1 h (x3) at room temperature. Incubation with secondary antibody donkey anti-Rat 488 (Thermo-Invitrogen; A32795; 1:500), Phalloidin-647 (Invitrogen, A30107; 1:1000) and Hoechst 33342 (AnaSpec, AS-83218; 1:1000) was performed overnight at 4 °C. Sections were washed with 0.3% triton in PBS for 1 h (x3) at room temperature and mounted on poly-L-lysine (Sigma, P8920) coated chamber slides (Ibidi, 80827). Tissue sections were incubated 80% glycerol (Sigma, G9012) in H2O at room temperature for 72 h, and 3D z-stack images were taken using a ZEISS LSM 880 microscope with Airyscan. Image analysis was performed using Cell Profiler. Individual nuclei were segmented in 3D by filtering, thresholding and performing a 3D watershed segmentation using the channel comprising the DNA dye Hoechst. The resulting 3D binary image was used as a mask to quantify the sum integrated density of Hoechst staining within each individual, complete nucleus. Using stromal cell nuclei as a reference for diploid (2C) DNA content, we classified all complete nuclei in the image by DNA content based on their sum integrated density and labeled them by color (2C in blue, >4C in yellow). Incomplete nuclei were excluded from analysis.

### RNA extraction and RT-qPCR
For RNA isolation from FACS-purified populations, mammary gland cell suspensions were blocked using Mouse BD Fc Block™ (BD Biosciences) for 10 min. Cells were subsequently resuspended on 1X PBS at a density of $10^7$ cells/ml and stained with the following antibodies for 30 min on ice: anti-CD24 PE (Stem Cell Technologies, 60099PE.1), anti-CD29 PE-Cy7 (BioLegend, 102222), anti-CD45-APC (BioLegend,105826), Ter119-APC (BD Biosciences, 561033), CD31-ACP (BD Biosciences, 551262). Propidium iodide at a final concentration of 0.5 μg/ml was used for the discrimination of dead cells. Stromal, basal and luminal mammary populations were sorted using a BD FACS Aria II Cell Sorter. Cells were subsequently lysed in TRIzol reagent (ThermoFisher, 15596018) and phase separated according to the manufacturer's protocol with an additional overnight RNA precipitation step in ethanol[66]. The RNA was further purified with TURBO DNase (Ambion, AM1906) treatment. For HC11, organoids and whole-gland tissue RNA isolation, the NucleoSpin RNA extraction kit (Macherey-Nagel, 740955.50) was utilized according to the manufacturer's instructions. Total RNA quality was analyzed by agarose gel electrophoresis and quantified using an ND-1000 spectrophotometer (NanoDrop). cDNA was prepared from 500-1000 ng of total RNA using iScript cDNA synthesis kit (Bio-Rad, 1708841). Quantitative RT-qPCR was performed in triplicates using SsoAdvanced Universal SYBR Green Supermix, (Bio-Rad, 1725272). The reactions were run in a Bio-Rad CFX'Connect Real-Time System and CFX Manager software (Bio-Rad) as follows: 95 °C for 2 min followed by 40 cycles of 95 °C for 15 s, 60 °C for 30 s and 72 °C for 45 s. Results were normalized to *Gapdh*. Primers used in this study are: *Csn2*: 5′-CCTCTGAGACTGA-TAGTATT-3′ and 5′-TGGATGCTGGAGTGAACTTTA-3′; *Wap*: 5′-TCTG CCAAACCAACGAGGAGTG-3′ and 5′-AGAAGCCAGCTTTCGGAACACC-3′; *Lalba*: 5′-GAGTCGGAGAACATCTGTGGCA-3′ and 5-CTTCTCAGAGC ACATGGGCTTG-3′; *Xdh1*: 5′-GCTCTTCGTGAGCACCAGAAC-3′ and 5′-CCACCCATTCTTTTCACTCGGAC-3′; *Plin2*: 5′-GACCGTGCGGACTTGC TC-3′ and 5′-GCCATTTTTTCCTCCTGGAGA-3′; *Btn1*: 5′-AGACAACGAC GACTTCGAGGAG-3′ and 5′-GTACCATCCAGAGGAGGTGCAC-3′; *Gapdh*: 5′-CATGGCCTTCCGTGTTCCTA-3′ and 5′-CCTGCTTCACCACCTTCTT-GAT-3′; *Cdkn1a*: 5′-ATCCAGACATTCAGAGCCACAG-3′ and 5′-ACG

AAGTCAAAGTTCCACCGT-3′; *Wee1*: 5′-TTGGCTGGCTCTGTTGATGA-3′ and 5′-CAGCTAAACTCCCACCATTACAG-3′; *Cdkn1b*: 5′-AACGTGCGAG TGTCTAACGG-3′ and 5′-CCCTCTAGGGGTTTGTGATTCT-3′.

### Statistical analyses
No statistical method was used to predetermine the sample size. Statistical analysis was performed using Prism9 software. Sample size, biological replicates, statistical test, and statistical significance are denoted in the figure legends. For the statistical analysis of the experiments involving contralateral intraductal injections, paired statistical tests were performed.

### Reporting summary
Further information on research design is available in the Nature Portfolio Reporting Summary linked to this article.

### Data availability
The primary data generated in this study are provided in the Source Data file. Any additional data are available from the corresponding author upon request. Source data are provided with this paper.

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

## Acknowledgements

The authors acknowledge core support from the UCSC Institute for the Biology of Stem Cells and California Institute for Regenerative Medicine (CIRM) Shared Stem Cell Labs (RRID:SCR_021353), Microscopy (RRID:SCR_021135), Facilities (CL1-00506-1,2), and Major Facilities (FA1-00617-1) awards and National Institutes of Health confocal grant (1S10OD23528-01). This work was supported by the National Institutes of Health: Eunice Kennedy Shriver National Institute of Child Health and Human Development (NIHCD) under award number R01HD098722 to L.H. and T32HD108079 to J.M, and. Other support for individuals was from a California Institute of Regenerative Medicine (CIRM) training grant (EDUC-12759) to R.M. and the National Institutes of Health, Maximizing Access to Research Careers (MARC) program under award number T34GM140956 to N.A. We thank Ben Abrams, Bari Nazario for technical assistance. We thank Santa Cruz Biotechnology for antibodies. And Dr. Diwakar R Pattabirama for the CK8-CreER; mTmG mice.

## Author contributions

R.M., J.M., and L.H. developed the concepts and supervised the research; R.M., J.M. designed experiments; R.M., J.M., K.C., N.A., M.K.C., C.L., G.T. performed the experiments; R.M., J.M. analyzed data; J.W.B., C.S., L.H. gave technical support and conceptual advice; J.W.B., C.S., L.H. acquired funding; R.M., J.M. wrote the original manuscript; R.M., J.M., L.H. reviewed and edited the manuscript; All authors have read and agreed to the published version of the manuscript. R.M. J.M. contributed equally to the work.

## Competing interests

L.H., R.M., J.M. have applied for patents related to this paper. L.H. received research funding from Zoetis, Inc., C.S., J.W.B., are employees of Zoetis Inc. All other authors declare no competing interests.
