## [Peer Review File · Nature Communications]

Physiological DNA damage Promotes Functional Endoreplication of Mammary Gland Alveolar Cells During LactationReviewer #1 (Remarks to the Author):

In this manuscript, the authors aim to investigate the mechanism by which some mammary alveolar cells become polyploid and frequently binucleate at, or during, lactation. This is an interesting and important topic. It was established over 40 years ago that "Functional differentiation in mouse mammary gland epithelium is attained through DNA synthesis, inconsequent of mitosis" using explant mammary gland cultures from virgin mice (PMID: 7286443). More recent studies, using genetically altered mouse models, showed that the number of binucleate cells can be diminished by ablation of Aurora A kinase (Rios, AC et al., Essential role for a novel population of binucleated mammary epithelial cells in lactation. *Nature Comms.* 2016 Apr 22;7:11400) or increased by ablation of the KRAB-domain zinc finger protein Zfp157 (Ho, TLF et al., The KRAB Zinc Finger Protein Roma/Zfp157 Is a Critical Regulator of Cell-Cycle Progression and Genomic Stability *Cell Rep.* 2016 Apr 26;15(4):724-734). In this manuscript, the authors' experimental approach encompasses the use of a mammary epithelial cell line (HC11), that can be partially differentiated, and a genetically altered mouse model of Wee1 deletion. While the data presented in this manuscript provide interesting insights into a possible mechanism for the generation of polyploid and binucleate mammary epithelial cells, a number of unanswered questions remain and additional experiments would provide a firmer foundation for the authors' conclusions.

Major concerns:

1. The data in Figure 1 provide a more detailed analysis of DNA content in luminal alveolar cells from nulliparous, pregnancy day 17.5, and days 2 and 5 of lactation (day2L and day5L) than previously described. They identify a mononucleate population that nevertheless are tetraploid. The question arises as to whether these cells would become binucleate in due course or whether they are 'trapped' earlier in mitosis. It is interesting that there is an approximately 2-fold increase in the percentage of both 4C and >4C cells between day2L and day5L. This suggests further rounds of DNA replication and it would be interesting to address this by carrying out timed injections of EdU.

It is worth noting that the use of CK8 to FACS purify cells will not isolate all luminal alveolar cells as levels of CK8 are low in a proportion of alveolar cells (Davis, FM et al., Single-cell lineage tracing in the mammary gland reveals stochastic clonal dispersion of stem/ progenitor cell progeny *Nat Commun.* 2016 Oct 25;7:13053). This is also clear from the image in Supplementary Figure 4. Could this skew the analysis?

2. I am curious as to why the authors chose to use the HC11 mouse mammary epithelial cell line as a model of pregnancy/lactation for their studies. There are now robust organoid models of mammary gland development (eg. Jamieson PR, Dekkers JF, Rios AC, Fu NY, Lindeman GJ, Visvader JE: Derivation of a robust mouse mammary organoid system for studying tissue dynamics. *Development* 2017, 144:1065–1071) and indeed, recently an organoid model for lactation/involution was described (Sumbal J, Chiche A, Charifou E, Koledova Z, Li H: Primary mammary organoid model of lactation and involution. *Front. Cell Dev. Biol* 2020, 8:68). Given the complex interactions between different cells, and in particular between hormone receptor expressing (ER+/PR+/Gata3+) and non-expressing (pStat5+) luminal cells, 3D organoid models seem to me to be essential for these studies. Organoids would allow an analysis also of knockout mouse models and CRISPR-mediated gene deletion.

Although HC11 cells respond to treatment with prolactin, dexamethasone and insulin by upregulating expression of beta-casein (encoded by Csn2) and form 'domes', it is unlikely that these cells differentiate to the extent that occurs in late pregnancy after only 3 days treatment. There is a hierarchy of milk protein gene expression with beta-casein being induced early in pregnancy followed by whey acidic protein (WAP) and then alpha-lactalbumin just before birth. The authors show only expression of beta-casein. Previous studies have shown that WAP is induced at the mRNA level after 3 days of DIP treatment and that the majority of cells (approx. 80%) were arrested at the G0/G1 stage (Sornapudi TR et al., Comprehensive profiling of transcriptional networks specific for lactogenic differentiation of HC11 mammary epithelial stem-like cells. *Scientific Reports* 8, Article number: 11777 (2018)). Could the authors explain how cells arrested before S phase (as they show in this manuscript) replicate their DNA more than once? I note that

HC11 cells were treated with DIP for 5 days in Supplementary Figure 2 but only 3 days in other experiments. I suggest that time course of differentiation is carried out over a longer time period and expression of beta-casein, WAP and alpha-lactalbumin analysed by both QRT-PCR and immunoblot. It is important that cell cycle regulators such as cyclinsD1 and D3, Cdk6, Cdc20, securin, Wee1, AuroraA, Plk1 and p21 are analysed by immunoblot. This would allow a more complete examination of the molecular events associated with binucleation in relation to the cycle and the differentiation process.

The data presented in Supplementary Figure 2 do not convince me that this HC11 model is a good model of endoreplication during alveolar differentiation. Figure 2C shows a huge increase in the number of >4C DIP3 cells following inhibition of Cdk1 activity, but less than a 1.5-fold change in beta-casein levels. I would suggest that it is an overstatement to equate this 1.5 fold change with increased milk production.

These data are inadequate on their own and would be much strengthened by the inclusion of studies in an organoid model.

3. In Figure 3, the response to DNA damage is analysed. How many HC11 cells undergo apoptosis in response to DNA damage? In Supplementary Figure 6a, it appears that a substantial number of cells are undergoing apoptosis. Adding propidium iodide to the assay would allow dying cells to be distinguished. The interesting point about endoreplication/ endoreduplication in mammary gland is that differentiation allows the survival of cells harboring a level of DNA damage that would otherwise induce cell cycle arrest and/or apoptosis. Analysis of gammaH2AX provides evidence of DNA damage but it would be interesting to look at other markers of the DNA damage response such as pS428-ATR, pS15-p53, pS345CHK1 and foci of 53BP1.

I do not really understand the rationale for injecting doxorubicin into mouse mammary glands but perhaps I have missed something. Doxo induces cell death and impairs mitochondria and seems a blunt tool to examine the consequences of physiological DNA damage on lactogenic differentiation. Would it not be better to use a mouse model deficient in DNA damage repair? The changes in beta-casein expression are again only about 1.5-fold. A more extensive analysis of milk protein expression could be carried out. WAP should be detectable at PD17.5 by immunoblots. Are a subset of luminal cells more vulnerable to DNA damage? It would appear so from Figure 3d. I am surprised at the number of gammaH2AX positive cells at PD10.5. Does this represent temporarily stalled replication forks? It would be surprising if these cells died before PD13.5.

4. Why have the authors only examined ATR and not also ATM? The pATR immunostaining of mammary gland is not very convincing. An immunoblot should be included here along with other markers of the DNA damage response. Again, only a 2-fold elevation in beta-casein expression is detected.

Given that Rios et al., (2016) showed that Aurora A levels increased during lactation but were not elevated at day17.5G, and furthermore that AuroraA was essential for the generation of binucleate cells and for milk production, could the authors explain why they propose that DNA damage during pregnancy is responsible for endoreplication and milk production? Replication stress has been shown to occur in the absence of Zfp157 which was accompanied by a significant increase in tetraploid cells during lactation. I am not sure how studies using pharmacological agents to cause DNA damage shed light on the mechanism of physiological DNA damage in the mammary gland.

5. The changes observed in Figure 6, although statistically significant, are minor and I doubt of much biological significance.

6. The data on the inhibition of Wee1 activity in lactating mammary glands are not very convincing. Is a 3.2% change likely to be important? The fold change in beta-casein expression is marginal.

7. The conditional Wee1 knockout mouse is more interesting. A confocal analysis of mammary gland would be worth including here to correlate cells that have deleted Wee1 with GFP and thus allow an in situ analysis of DNA content. Why was only one allele of Wee1 deleted? In Figure 8e, it

looks as if one gland has been suckled recently and the other not. How did the authors control for milk removal by the pups. Did pups suckled by Wee1fl/+ dams thrive?

It is not clear to me why proliferation during pregnancy would result in DNA damage and the generation of polyploid and binucleate cells that continue to proliferate. Given a proposed rationale for cells undergoing a further round of DNA replication at the onset of lactation is to produce more ribosomes for the extensive increase in protein production required during lactation, it does not make sense to me that endoreduplication/ endoreplication would occur during mid-pregnancy where one would expect cells harboring DNA damage to exit the cell cycle and stop proliferating. Could the authors explain their observations? The title refers only to lactation.

Minor comments:

PLIN2 is perilipin-2 not periplin-2

The labelling of axes in some of the supplementary figures is a little difficult to read as there are black boxes on the scale.

Reviewer #2 (Remarks to the Author):

Physiological DNA damage Promotes Functional Polyploidization of Mammary Gland Alveolar Cells During Lactation.

In this study, the authors showed that DNA damage and activation of ATR mediated DNA damage response pathway accumulates could promote polyploidization of mammary alveolar cells during lactation. They identified a mechanism involving early mitotic arrest through CDK1 inactivation, resulting in a heterogeneous alveolar population with regards to ploidy and nuclei number. The inactivation of CDK1 is mediated by the DNA damage response kinase WEE1 with heterozygous loss of Wee1 resulting in decreased endoreplication and milk production. Thus, the authors propose the DNA damage response to replication stress couples proliferation and endoreplication during mammary gland alveologenesis. This is an interesting study, although it remains preliminary and some more experiments are still needed to support their conclusions.

Major points

1. In the last part the authors tried to show that the DNA damage response regulates endoreplication via WEE1. Although the authors did not claim WEE1 is only one, it gave an impression that WEE1 is a primary factor for regulating endoreplication. But WEE1 is not identified by an unbiased screening rather than through a candidate approach. Indeed, the WEE1 inhibitor treatment or WEE1+/- conditional knockout only partially decreased the 4C population, which might suggest the involvement of some other factors. In this regard, the author should also test WEE1 in HC11 cells to see if WEE1 inhibitor and knockdown could affect partially or completely, the spontaneous or induced tetraploid cell formation.
2. CHK1 is also an important ATR downstream gene that is involved in DDR. Why it was not checked? It should also be examined as a candidate,
3. An unbiased screening using CRISPR-Cas9 mediated approach might be good for identify some more candidate genes, although WEE1 could still be focused as a main factor.
4. In this case, the current data just showing "DNA damage response regulates endoreplication via WEE1" is preliminary. The molecular mechanism underlying such a regulation should be investigated.
5. Authors claims that the ATR-DDR is the major mediator for the endoreplication. However, there is no experiment data to exclude ATM. Inhibitors for ATM and ATR should be used in the study and their effects be compared to justify the claim.
6. Some figures are too much simplified with only a few panels although some data are placed in the supplementary figures. However, it makes it difficult to find the key data. For example, Figure 3 indicated that "doxorubicin-injected MGs at PD17.5 shows an increase in the proportion of 4C and > 4C luminal populations (14.86% and 47.1% increase in the overall population, respectively)", which is a piece of key data, but it was not in Figure 3. Figure 3i only showed minor

changes. It is suggested to enrich these figures by adding some data from related Supplementary figures.

Minor points

1. Figure 1a: Please indicate in the figure legends regarding how the blue and yellow colour are generated, and also the stages of the sample (P17.5?).
2. Suggest also show a P10.5 samples as a control in this figure.

Reviewer #3 (Remarks to the Author):

This paper addresses the role of the DNA damage response pathway in polyploidy in functionally differentiated mammary epithelial cells. The authors show increased ploidy in mammary luminal epithelial cells throughout differentiation and this response is dependent in part upon replication stress and Wee1 kinase activity. Major concerns are that the majority of the observations have been previously published (Visvader et al Nat Comm, 2016). This study showed increased ploidy throughout mammary gland pregnancy and lactation and was dependent upon Aurora A kinase, which has recently been shown to also play a role in replication stress and activation of the CDK1 pathway (Wang et al, Reproductive Biol and Endo, 2021).

Specific comments:

1. The number of mice in each litter was not documented.
2. The quantification of milk in the gland from staining is not a reliable measure of mammary gland function. Milk expression is highly dynamic and dependent upon the litter size and when the pups last nursed, which was not documented. The use of normalized litters and pup weight would provide a better measurement of mammary gland function.
3. It isn't clear why the authors utilized the CK8 Tamoxifen-induced Cre when other more reliable lactation and mammary specific Cre drivers are available (i.e. WAP-Cre). The use of the CK8 Cre may give rise to non-mammary specific phenotypes. Moreover, the WAP-Cre would also allow for a more detailed impact on mammary gland function by measuring changes in pup weights.
4. The authors do not show controls (i.e. dye) for the IDI experiments how much of the injected volume is distributed throughout the gland.
5. The addition of additional markers for differentiation and involution should be included (p-STAT5, p-STAT3, etc).

Reviewer #4 (Remarks to the Author):

Physiological DNA damage Promotes Functional Polyploidization of Mammary Gland Alveolar Cells During Lactation.
Manuscript 378304 Nat Comm

Overview:

This is an interesting manuscript describing a potential mechanism for DNA-damage induced differentiation of mammary epithelial cells. The induction of DNA damage in a tissue, and its response in order to preserve genome integrity is an important developmental and medical question. The response of the mammary gland to DNA damage has not yet been well described and this manuscript provides some interesting insight into the process as it relates to endoreplication and milk production. The authors have described a mechanism of DNA damage induced Wee1 inactivation of CDK1 leading to endoreplication and multinucleated alveolar cells.

Major points

Strengths

1. The research uses multiple models of mammary epithelial cell differentiation, including a well characterized cell line (HC11), CD-1 outbred mice, and an inducible reporter mouse line (Ck8-

CreER/mTmG), to knockout Wee1 in Ck8 expressing cells, together providing a robust line of inquiry.

2. The experiments in general are well controlled.

3. The observations are important to the field.

Queries

4. The model described, and visualized in Fig 8, is not clear based on the experimental conditions. The authors induce DNA damage using doxorubicin, in order to mimic replication induced stress. In this case they should have chosen a compound that generates primarily single stranded DNA breaks. Doxorubicin, however, creates predominantly double strand DNA breaks, as it is a Topoisomerase II poison. The authors appear to be inducing a different set of breaks and stimulating a different cellular response to DNA damage than what they describe in their model, which is ATR based recognition of DNA damage. The schematic also shows double strand DNA breaks. The authors should use a compound that more accurately stimulates single strand breaks. [Capranico G, De Isabella P, Penco S, Tinelli S, Zunino F. Role of DNA breakage in cytotoxicity of doxorubicin, 9-deoxydoxorubicin, and 4-demethyl-6-deoxydoxorubicin in murine leukemia P388 cells. *Cancer Res.* 1989;49(8):2022–2027.]

5. The authors use gamma H2AX, which can be a marker for double strand DNA breaks, and is also phosphorylated by ATM on serine 139. The authors should determine if ATM is phosphorylated in the mammary gland at pregnancy day 10.5 as well as ATR.

6. Page 6, "In the MG, DNA damage occurs in alveolar cells during pregnancy 35,..." What type of DNA damage is known to occur in alveolar cells during pregnancy? Is it predominantly single or double stranded breaks? Which DNA damage pathway is expected to be induced? Is it both ATM and ATR-based?

7. The gamma H2AX staining is unclear. Very often images show staining as the whole nucleus, when it should be in punctate repair foci. Greater resolution is required.

8. Has the knockout of Wee1, or at least its reduced level of protein, been confirmed by western, or using immunofluorescence? The reduction in protein should be shown, rather than just at the transcript level.

Minor issues

1. Page 3 – the discussion in the introduction regarding the coupling of differentiation, cell checkpoints (particularly the G1S checkpoint in the last paragraph) and DNA damage in the mammary gland would benefit from more detailed reference citation.

2. What is meant by 'differential' ability to build a milk supply (page 12).

3. There is very little information on the C57BL/6-Wee1^{-/-} line.

4. Was HC11 STR tested?

Reviewer #1:

1A) This suggests further rounds of DNA replication and it would be interesting to address this by carrying out timed injections of EdU.

We agree with Reviewer #1 that polyploid cells would require further rounds of DNA replication. This important point was addressed by Rios and colleagues, in which they showed the results of injecting pregnant dams with a single dose of EdU at 16.5 or 18.5 days pregnant and then analyzing them after defined periods of chase (Figure 2b)¹. By 18.5 days pregnant, binucleated tetraploid EdU-positive luminal cells were apparent after 6h, and up to 3 and 12 days of chase. They concluded that tetraploid cells arise through DNA re-replication, and not cell fusion. Because this set of studies has been completed, we did not repeat the experiment.

1B) It is worth noting that the use of CK8 to FACS purify cells will not isolate all luminal alveolar cells as levels of CK8 are low in a proportion of alveolar cells (Davis, FM et al., Nat Commun. 2016 Oct 25;7:13053). This is also clear from the image in Supplementary Figure 4. Could this skew the analysis?

This is a great question. We are aware that the expression of CK8 varies in luminal cells and, indeed, we detect lower CK8 expression by FACS in mammary gland cell preps at LD5 compared to PD17.5 (Response Figure 1a). In addition, we also detect three subpopulations of luminal cells based on their expression of CK8 and their side scatter (SSC) properties: CK8^{low}; SSC^{low}, CK8^{high}; SSC^{low} and CK8^{high}; SSC^{high} (Response Figure 1b), which were analyzed as one population in the original manuscript. We find at LD5 that the CK8^{low}; SSC^{low} population increases while the CK8^{high}; SSC^{high} population decreases, in comparison to PD17.5. No differences in the CK8^{high}; SSC^{low} populations were detected (Response Figure 1c-h).

To address Reviewer #1's specific point, we analyzed the luminal subpopulations at PD17.5 and LD5 independently. We observe that tetraploid (4C) and polyploid (>4C) luminal cells are largely CK8^{high} at LD5. In contrast, at PD17.5, most luminal cells are diploid (2C) regardless of the expression levels of CK8 (Response Figure 1c-h). Therefore, the data support the use of CK8 for FACS isolation in our studies of cells because this marker allows for effective analysis of the vast majority of luminal cells undergoing endoreplication. In the revised manuscript, we include this analysis of CK8 expression in luminal cells in Revised Supplementary Figure 11 to address Reviewer #3 who asked us to explain why we chose the CK8-CreER system for the *Wee1* conditional knock-out.

Figure 1. Tetraploid and polyploid cells emerge in a CK8^{high} luminal subpopulation at LD5.

a) Detection of CK8 expression in PD17.5 (blue) or LD5 (red) whole mammary gland (MG) cell preps. CK8 positive cells were detected by using an IgG negative control (grey). **b)** Dot plot representing the CK8 negative (black), CK8^{low}; SSC^{low} (green), CK8^{high}; SSC^{low} (blue) and CK8^{high}; SSC^{high} (red) populations detected by FACS from PD17.5 or LD5 MGs. **c and d)** Representative FACS histograms for DNA content analysis of the CK8^{low}; SSC^{low}, CK8^{high};

SSC^{low} and CK8^{high}; SSC^{high} populations from PD17.5 (**g**, blue histograms) or LD5 (**h**, red histograms) MGs. **e**) Quantification of the percentage of CK8^{low}; SSC^{low}, CK8^{high}; SSC^{low} and CK8^{high}; SSC^{high} cells from PD17.5 (blue) or LD5 (red) MGs, detected by FACS. **f-h**) Quantification of FACS DNA content analysis showing the percentage of diploid (2C), tetraploid (4C) and polyploid (>4C) cells within the CK8^{low}; SSC^{low} (**f**), CK8^{high}; SSC^{low} (**g**) and CK8^{high}; SSC^{high} (**h**) populations from PD17.5 (blue) or LD5 (red) MGs. Data are shown as the mean \pm SD; n=3 biological replicates. For (**e-h**) two-tailed Student's t- test. p values: * < 0.05, **< 0.01, *** < 0.001, **** < 0.0001.

2A) *I am curious as to why the authors chose to use the HC11 mouse mammary epithelial cell line as a model of pregnancy/lactation for their studies.*

To spare the number of animals utilized in our studies, we routinely use the HC11 cell line to pilot experiments before moving *in vivo*, as required by our IACUC protocol. In addition, although the number of HC11 cells undergoing endoreplication during lactogenic differentiation is low, these cells recapitulate our observations in the mammary gland of increased endoreplication in response to doxorubicin-induced DNA damage, as shown in our original manuscript (Supplementary Figure 6). That said, we agree with Reviewer #1 that our studies would benefit from experiments on mammary organoids, which our lab has the expertise to perform^{2,3}. In the revised manuscript, we present data on mammary organoids, testing the effect of doxorubicin, hydroxyurea, nucleosides, CDK1 and WEE1 inhibitors on alveolar endoreplication and milk production (Response Figure 2). You can see that there are more cells with 4C and >4C DNA content and increased expression of milk protein genes (*Csn2*, *Wap*, *Lalba*) when organoids are treated with drugs that either increase DNA damage (doxorubicin and hydroxyurea) or inhibit Cdk1 (Ro-3306). In contrast, treatment with drugs that either reduce DNA damage (nucleosides) or inhibit Wee1 (Mk-1775), result in fewer cells with 4C and >4C DNA content and reduced expression of milk protein genes (*Csn2*, *Wap*, *Lalba*). These results fully corroborate our findings *in vivo* where we used intraductal injections to deliver these drugs to mammary tissue during pregnancy. In the revised manuscript, we include these data in Revised Figure 6.

Figure 2. Mammary gland organoids recapitulate the phenotype observed *in vivo*.

a) Quantification of FACS DNA content analysis showing the percentage of diploid (2C), tetraploid (4C) and polyploid (>4C) CK8 positive cells from the mammary gland (MG) organoids untreated (Ctr) or treated with doxorubicin (Doxo), hydroxyurea (Hu), Ro-3306, nucleosides (Nucs) or Mk-1775, using the concentrations indicated. **b)** Quantification of the expression of the milk protein genes *Csn2*, *Wap* or *Lalba* in MG organoids untreated or treated as indicated. Data are shown as the mean \pm SD; n=3 biological replicates. two-tailed Student's t- test. p values: * < 0.05, ** < 0.01, *** < 0.001.

2B) Could the authors explain how cells arrested before S phase (as they show in this manuscript) replicate their DNA more than once?

We do not believe the data show that the cells are arrested before S phase. As shown in the original manuscript (Supplemental Figure 1c), there is no difference in the percentage of cells in S phase between the stages of priming and DIP5. In contrast, we observe that the percentage of cells with 4C DNA content decreases (trending) while the percentage of cells with >4C DNA content significantly increases. Thus, coupled with the observation that the percentage of cells with 2C DNA content does not change between priming and DIP5, our interpretation is that the cells that had 4C DNA content during the priming phase underwent **further endoreplication** and became polyploid by DIP5.

2C) Reviewer #1 requested additional data on the HC11 cells, but then stated: “The data presented in Supplementary Figure 2 do not convince me that this HC11 model is a good model of endoreplication during alveolar differentiation.”

As explained above, while we piloted our experiments and tested some hypotheses using HC11 cells, we agree they are just a cell line, and we further elucidated mechanisms governing endoreplication in the original manuscript by using intraductal injection and by generating/analyzing a *Weel* conditional knock-out mouse. However, to further address Reviewer #1’s point, we performed an extensive analysis using mammary gland organoids as an additional *ex vivo* model (Response Figure 2). We find that mammary gland organoids recapitulate the phenotypes observed *in vivo*. In the revised manuscript, these data are included in Revised Figure 6.

3A) In Figure 3, the response to DNA damage is analysed. In Supplementary Figure 6a, it appears that a substantial number of cells are undergoing apoptosis. Adding propidium iodide to the assay would allow dying cells to be distinguished.

This is an important point. We analyzed HC11 cells undergoing cell death using propidium iodide staining and FACS, quantifying the percentage of sub-G1 cells 24h after DMSO (vehicle) or doxorubicin treatment, and 3 days after differentiation (DIP3) (Response Figure 3). These data show that in DMSO treated cells by DIP3 there is an increase in the percentage of cells undergoing spontaneous apoptosis when compared to undifferentiated, DMSO-treated HC11 cells. Doxorubicin induces apoptosis in ~4% of HC11 cells 24h after treatment and this remains unchanged at DIP3 (Response Figure 3a, b). These data, together with the DNA content analysis presented in the original manuscript (Supplementary Figure 6 c-f), which shows increased percentage of tetraploid (4C) and polyploid populations (>4C) at DIP3, demonstrate that HC11 cells preferentially undergo endoreplication and lactogenic differentiation in the presence of DNA damage, rather than apoptosis.

To further address Reviewer #1’s comment, we also analyzed cells undergoing cell death after intraductal injection of DMSO or doxorubicin by collecting mammary glands and performing propidium iodide staining and subsequent FACS analysis to quantify the percentage of sub-G1 cells within the CK8 positive population (Response Figure 3c, d). Interestingly, these data show that doxorubicin treatment decreases the percentage of cells undergoing apoptosis. This is consistent with previous studies showing that cells undergoing endoreplication in response to genotoxic stress are more resistant to apoptosis⁴. In the context of mammary gland development during pregnancy, endoreplication may represent the preferred mechanism to prevent proliferation of cells harboring DNA damage due to replication stress, rather than apoptosis, which would be deleterious to proper tissue growth and function. In the revised manuscript, we include these data in support of this idea — that endoreplication via induction of differentiation through the DNA damage response limits proliferation of damaged cells in a context where apoptosis would be deleterious to tissue integrity and function (Revised Supplementary Figures 5 and 6).

Figure 3. Mammary gland cells preferentially undergo endoreplication and lactogenic differentiation, and not apoptosis, in response to DNA damage. **A)** Representative FACS histograms for DNA content analysis showing the sub-G1 population of HC11 cultures treated with DMSO or doxorubicin (Doxo) for 24h (top histograms) or after 3 days of differentiation (DIP3, bottom histograms). **B)** Quantification of the percentage of HC11 within the sub-G1 population, 24h after treatment with DMSO or Doxo, or 3 days after differentiation (DIP3). **C)** Representative FACS histograms for DNA content analysis showing the sub-G1 population within the CK8 positive MG cells at PD17.5, after intraductal injections with DMSO or Doxo. **D)** Quantification of the percentage of sub-G1 cells within the CK8 positive MG population at PD17.5, after intraductal injections with DMSO or Doxo. Data are shown as the mean \pm SD; n=3 biological replicates. For (b) one-way ANOVA Tukey's test. For (d) two-tailed paired Student's t- test. P values: * < 0.05, ** < 0.01, *** < 0.001, **** < 0.0001.

3B) *The interesting point about endoreplication/ endoreduplication in mammary gland is that differentiation allows the survival of cells harboring a level of DNA damage that would otherwise induce cell cycle arrest and/or apoptosis. Analysis of gammaH2AX provides evidence of DNA damage but it would be interesting to look at other markers of the DNA damage response such as pS428-ATR, pS15-p53, pS345CHK1 and foci of 53BP1.*

We appreciate the suggestions. In the original manuscript we presented evidence of DNA damage accumulation by immunostaining for γ H2AX, a histone marker broadly used for the detection of DNA damage. We show γ H2AX accumulating at mid-pregnancy during the proliferative phase of alveologenesis (Figure 3a, b and Supplementary Figure 4). We also show activation of the ATR-DNA damage response to replication stress during pregnancy persisting until late pregnancy when endoreplication begins (Figure 4a, b and Supplementary Figure 7a).

In the revised manuscript, we present additional immunohistochemical analyses for pS345-CHEK1 (the activating site phosphorylated by ATR), pS1981-ATM (the auto-phosphorylated activating site), and 53BP1 (Response Figure 4; ⁵⁻⁷). These data show ATR-targeted pS345-CHEK1 is upregulated at the onset of lactation (LD2), when endoreplication accelerates (Response Figure 4a, f). This is consistent with the role of CHEK1 in activating the G2/M checkpoint in response to DNA damage ⁶. In addition, we show that pS1981-ATM is downregulated during lactation (Response Figure 4b, f), supporting our model that pT1989-ATR, which is still expressed at the beginning of lactation (Response Figure 4c, f), is the primary regulator of the endoreplication response to replication stress. The DNA damage marker 53BP1 is also upregulated at mid-pregnancy (PD10.5 and PD15.5; Response Figure 4d, f), during the proliferative phase of alveologenesis – similar to our observation about γ H2AX (Response Figure 4e, f).

We did not examine the presence of pS428-ATR because this site is not necessary for CHEK1 activation ⁸, but rather it is important for the anti-apoptotic role cytoplasmic ATR plays at the mitochondria ⁹ — a role that is independent of its checkpoint function and thus independent of the role we identify in endoreplication. We also did not include an examination of p53 (pS15-p53) because we recently published data showing that p53 expression remains unchanged in the mammary gland during pregnancy and lactation ¹⁰. Furthermore, we show in the original manuscript that the expression of the p53 transcriptional target p21 (*Cdkn1a*) also remains unchanged (Supplementary Figure 9a). Therefore, these data suggest that p53 does not play a role in the regulation of mammary endoreplication during alveologenesis.

Together, our analyses show that DNA damage, detected by γ H2AX and 53BP1 expression, accumulates during the proliferative phase of alveologenesis at mid-pregnancy. This DNA damage results in the activation of ATR (not ATM) and the ATM target CHEK1. This pathway remains active until the beginning of lactation, when endoreplication occurs. These data strongly support the model outlined in our original manuscript that the ATR-DNA damage response regulates endoreplication during alveologenesis. In the revised manuscript, these data are included in Revised Figure 3 and Revised Supplementary Figure 7.

Figure 4. DNA damage response regulators and effectors activated during pregnancy and lactation. A-e) Quantification of nuclear (a) pS345-CHK1, (b) pS1981-ATM, (c) pT1989-ATR,

(d) 53BP1 and (e) γ H2AX Integrated Density (IntDen) in the luminal CK8⁺ population from nulliparous (Nullip.), PD10.5, PD15.5, PD18.5 and LD2. Data show values from 100 cells per time-point of n=3 biological replicates combined (orange dots). Black dots represent the means of each n=3 biological replicates. Black horizontal bars represent the mean of means. Error bars represent S.E.M. One-way ANOVA Tukey's test. p values: * < 0.05, **< 0.01. f) Detection of pS345-CBK1, pS1981-ATM, pT1989-ATR, 53BP1 and γ H2AX (green) in optical-cleared MG sections from nulliparous (Nullip.), PD10.5, PD15.5, PD18.5 and LD2. Luminal CK8⁺ cells in magenta. Nuclear DNA by propidium iodide. Images are representative of n=3 biological replicates.

3C) *I do not really understand the rationale for injecting doxorubicin into mouse mammary glands but perhaps I have missed something. Doxo induces cell death and impairs mitochondria and seems a blunt tool to examine the consequences of physiological DNA damage on lactogenic differentiation.*

Our rationale for performing these experiments was to investigate the increase in γ H2AX staining that we observe 1) *in vivo* in the mammary gland where it peaks at PD10.5 (Figure 3a, b and Supplementary Figure 4 in the original manuscript), as well as 2) in milk producing domes of differentiated HC11 cells (Supplementary Figure 5c in the original manuscript). Upon observing this endogenous DNA damage, we decided to investigate the effects of increased DNA damage (i.e. gain-of-function) on endoreplication and lactogenic differentiation by using the drug, doxorubicin, that broadly damages DNA. Indeed, we discovered that doxorubicin increases endoreplication and lactogenic differentiation (Figure 3c-g and Supplementary Figure 6c-j in the original manuscript); however, it induces double stranded breaks rather than the single-stranded breaks that are more characteristic of physiological DNA damage. Reviewer #4 also raised a similar point about our choice to generate DNA damage with doxorubicin. We are aware that doxorubicin is a blunt tool; thus, in the original manuscript we presented additional studies using hydroxyurea (Figure 5 and Supplementary Figure 8a-b), an agent that generates replication stress by inhibiting nucleosides synthesis, which is more reflective of physiological DNA damage. Again, using this drug, we observe increased endoreplication and lactogenic differentiation with DNA damage. To address the Reviewers' comments regarding doxorubicin, we make clearer the types of DNA damage being induced by the different drugs in the revised manuscript. We also move the data on doxorubicin to Revised Supplementary Figures 5 and 6 in order to focus the data presented in the main figures on hydroxyurea and nucleosides, which increase and decrease replication stress, respectively (Revised Figures 4 and 5).

3D) *The changes in beta-casein expression are again only about 1.5-fold. A more extensive analysis of milk protein expression could be carried out. WAP should be detectable at PD17.5 by immunoblots.*

In the revised manuscript, we present a more extensive analysis of milk protein gene expression using RT-qPCR (Response Figure 5). We find that intraductal injection of agents causing DNA damage, hydroxyurea (Hu) or doxorubicin (Doxo), increase *Wap* expression, in addition to *Csn2*. Furthermore, *Lalba* and *Xdh1* are increased by Doxo treatment, and *Plin2* and *Btn1* by Hu treatment (Response Figure 5a, b). In contrast, intraductal injection of nucleosides, which reduce replication stress, decreases the expression of this panel of milk protein genes (Response Figure

5c, d). This decrease in milk protein gene expression was also observed with the WEE1 inhibitor Mk-1775 (Response Figure 5e, f). All these data are consistent with our model in which CHK1 negatively regulates WEE1 downstream of DNA damage and ATR. By inhibiting WEE1 or reducing replication stress with nucleosides, we relieve the inhibition of CDK1; the G2/M checkpoint is not triggered, resulting in diminished endoreplication and decreased milk production. In contrast, increasing DNA damage has the opposite effect, stimulating the DNA damage response and activating WEE1, which inhibits CDK1 and triggers the G2/M checkpoint, leading to endoreplication and increasing polyploidy and milk production. In the revised manuscript, these data are included in Revised Figures 4-6 and Revised Supplementary Figures 6, 9 and 10.

Figure 5. DNA damage regulates milk production via WEE1. a-f) Quantification of milk genes expression (*Lalba*, *Wap*, *Csn2*, *Plin2*, *Xdh1* and *Btn1*) after contralateral intraductal injections with DMSO/PBS as control or (a) doxorubicin (Doxo), (b) hydroxyurea (Hu), (c and d) nucleosides (Nucs) or (e and f) WEE1 inhibitor Mk-1775, as detected by RT-qPCR. Data shown as the mean \pm S.E.M; n=3 or n=4 biological replicates; two-tailed paired Student's t-test. p values: * < 0.05, ** < 0.01.

3E) Are a subset of luminal cells more vulnerable to DNA damage? It would appear so from Figure 3d. I am surprised at the number of gammaH2AX positive cells at PD10.5. Does this represent temporarily stalled replication forks? It would be surprising if these cells died before PD13.5.

This is an important observation. Indeed, there does appear to be a subset of more highly labelled cells after doxorubicin injection. Doxorubicin induces DNA damage by inhibiting topoisomerase II during DNA replication. Considering that this analysis of γ H2AX was conducted 24h after administering the drug by intraductal injection, our interpretation is that the highly labelled subset of cells represents cells that were in S phase and actively replicating their DNA at the time of injection. Thus, our data indicate that proliferative luminal cells are the ones more susceptible to DNA damage. This point is clarified in the revised manuscript.

To further address this point in the revised manuscript, we performed additional immunohistochemical analysis of replication protein A (RPA), which binds single-stranded DNA and helps facilitate the recruitment and activation of ATR at sites of replication stress - stalled or collapsed replication forks ^{11,12}. We observe RPA throughout alveologensis, with the peak occurring at PD10.5 similar to our data on γ H2AX and pT1989-ATR (Response Figure 6). The presence of these proteins suggests that a DNA damage response to replication stress (i.e. stalled or collapsed replication forks) is being mounted. Regarding the question about whether the cells are dying, as explained above, we performed FACS analysis to identify the percentage of sub-G1 cells within the CK8 positive population after DMSO or doxorubicin intraductal injections (Response Figure 3). These data show that doxorubicin actually **decreases the percentage of cells undergoing apoptosis**. This result is consistent with previous studies showing that cells undergo endoreplication in response to genotoxic stress are more resistant to apoptosis ¹³; this has been termed the DNA damage differentiation response ¹⁴. In the revised manuscript, we present these new data in Revised Figure 3 and Revised Supplementary Figures 5-7.

Figure 6. Replication protein A (RPA) accumulates during pregnancy due to replication stress. a-c)

Quantification of nuclear (a) γ H2AX, (b) pT1989-ATR, and (c) RPA Integrated Density (IntDen) in the luminal CK8⁺ population from nulliparous (Nullip.), PD10.5, PD15.5, PD18.5 and LD2. Data show values from 100 cells per time-point of n=3 biological replicates combined (orange dots). Black dots represent the means of each n=3 biological replicates. Black horizontal bars represent the mean of means. Error bars represent S.E.M. One-way ANOVA Tukey's test. p values: * < 0.05. **d)** Detection of γ H2AX, pT1989-ATR, and RPA (green) in optical-cleared MG sections from nulliparous (Nullip.), PD10.5, PD15.5, PD18.5 and LD2. Luminal CK8⁺ cells in magenta. Nuclear DNA by propidium iodide. Images are representative of n=3 biological replicates.

4A) Why have the authors only examined ATR and not also ATM? The pATR immunostaining of mammary gland is not very convincing. An immunoblot should be included here along with other markers of the DNA damage response.

We focused on ATR due to the fact it is activated in response to lengths of single-stranded DNA produced by replication stress¹⁵. In response to Reviewers' comments, we also performed immunohistochemistry and immunoblotting for pT1989-ATR, pS1981-ATM and pS345-CHK1 on tissue collected from nulliparous, PD10.5, PD15.5, PD18.5 and LD2 (Response Figure 4a-c, f, and Response Figure 7). We find that pT1989-ATR is upregulated at the end of pregnancy (PD18.5) when endoreplication starts (Response Figure 4c, f, and Figure 7a, d). Accordingly, ATR-targeted pS345-CHK1 is upregulated at the onset of lactation (LD2), when endoreplication accelerates (Response Figure 4a, f, and Figure 7b, e). This is consistent with the role of CHK1 in activating the G2/M checkpoint in response to DNA damage. Further, we observe that pS1981-ATM is downregulated during lactation (Response Figure 4b, f, and Figure 7c, f), supporting our model that pT1989-ATR is the primary regulator of the endoreplication response to replication stress. In the revised manuscript, we have included these data in Revised Figure 3 and Revised Supplementary Figure 7.

Quantification of the (d) pT1989-ATR/ATR, (e) pS345-CHK1/CHK1 and (f) pS1981-ATM/ATM ratios detected by Western Blot in whole MG lysates from nulliparous (Nullip.), PD10.5, PD15.5, PD18.5 and LD2 mice. Data shown as the mean \pm S.D.; n=3 biological replicates; two-tailed Student's t-test. p values: * < 0.05, ** < 0.01, *** < 0.001 and **** < 0.0001.

I am not sure how studies using pharmacological agents to cause DNA damage shed light on the mechanism of physiological DNA damage in the mammary gland.

We intraductally injected pharmacological agents, both doxorubicin and hydroxyurea, in order to induce DNA damage greater than the physiological damage we observed during alveologenesis. We see this as a pharmacological gain-of-function approach. These experiments demonstrate the principle that DNA damage induces endoreplication and increases milk production. To investigate the role of physiological replication stress in endoreplication and milk production, we intraductally injected nucleosides, which relieve replication stress^{16,17}. We see this as a pharmacological loss-of-function approach. Indeed, we find that intraductal injection of nucleosides had the opposite effect of doxorubicin and hydroxyurea – reducing endoreplication and decreasing milk production (Figure 6 in the original manuscript).

5. The changes observed in Figure 6, although statistically significant, are minor and I doubt of much biological significance.

We appreciate that the changes may seem minor to Reviewer #1, but we ask that the Reviewer consider the experiment in its biological context. First, we are intraductally injecting nucleosides to reduce replication stress. To our knowledge, using intraductal injection to reduce replication stress in the mammary gland has never been published. Indeed, we got the idea to try the experiment because Bester and colleagues showed that nucleoside supplementation in **tissue culture** relieved replication stress in primary fibroblasts cells by overexpressing Cyclin E¹⁶. Please note that nucleosides only relieve one source of replication stress, which is otherwise generated in many ways —by nicks, gaps, ssDNA, DNA lesions, fragile sites, conflicts between replication and transcription, and limitation of essential replication factors (both components of the replication machinery and **nucleosides**). Even though nucleoside supplementation only addresses one of these many sources, we showed that relieving this type of stress during pregnancy significantly reduced the percentage of polyploid cells. In the original manuscript, we show that the percentage of polyploid cells detected in untreated glands at LD5 was ~19%, while treatment with nucleosides brings this percentage down to ~16% (Figure 6f). The Reviewer suggests these changes are minor, but they represent a ~15% reduction of the overall polyploid population that we show is sufficient to reduce *Csn2* expression by more than 50% and decrease overall milk production by 30% (Figure 6g-j).

Altogether, Figure 6 in the original manuscript shows that relieving replication stress via nucleoside injection **alone** can reduce endoreplication and decrease both polyploidization and milk production. This strongly suggests that replication stress during alveologenesis is at the core of the DNA damage response that drives the differentiation and polyploidization of alveolar cells during pregnancy. Our results are also notable because they demonstrate that small changes in the polyploid population result in larger changes in milk production, highlighting the biological importance of this population for efficient milk production. To further address the Reviewer's comment in the revised manuscript, we explain more clearly the extent to which nucleoside supplementation is expected to relieve replication stress to better contextualize our results. Additionally, we use organoids to show similar relief of replication stress and concomitant reductions in endoreplication, polyploidization and milk production in response to pharmacological manipulation (Response Figure 2). In the revised manuscript, these data are included in Revised Figure 6.

6) *The data on the inhibition of Wee1 activity in lactating mammary glands are not very convincing. Is a 3.2% change likely to be important? The fold change in beta-casein expression is marginal.*

While the decrease in the tetraploid (4C) population is 3.2%, the polyploid population (>4C) decreases 16.31%. These changes in endoreplication decreased *Csn2* expression by ~25% and overall milk production by ~20%. In the revised manuscript, we show a more extensive analysis of milk protein gene expression after intraductal injection of the WEE1 inhibitor Mk-1775 (Response Figure 5e, f). These data show consistent decreases in all milk protein genes tested, confirming that decreased endoreplication as a result of WEE1 inhibition results in decreased milk production. In addition, our results in the *Wee1* conditional knockout mouse are robust and consistent with these data (Response Figures 8-10). In the revised manuscript, these new data are included in Revised Figures 6-8 and Revised Supplementary figures 10-12.

7) a) *The conditional Wee1 knockout mouse is more interesting. A confocal analysis of mammary gland would be worth including here to correlate cells that have deleted Wee1 with GFP and thus allow an in situ analysis of DNA content.*

To address the Reviewer's comment, we performed FACS DNA content analysis to quantitatively evaluate DNA content in many more cells than in situ analysis would allow. In the revised manuscript, we show this analysis using *Wee1*^{+/+} and *Wee1* knock-out (*Wee1*^{fl/fl}) mice at LD2. We find that 98% and 99% of CK8+ cells are also GFP+, respectively (Response Figure 8a, b). Since virtually all the CK8+ cells are also GFP+; we did not find differences in the DNA content of the CK8+/GFP+ population compared to the CK8+ population alone (data not shown). The data on the double positive cells indicate that depletion of *Wee1* reduces the 4C and >4C DNA content in the CK8+ population (Response Figure 8c-e). In the revised manuscript, we include the analysis of the double CK8+/GFP+ population, instead of the CK8+ population alone, in Revised Figure 8 and Revised Supplementary Figure 12.

Figure 8. Loss of Wee1 inhibits endoreplication of mammary alveolar cells during lactation. a) CK8 and GFP expression of *Wee1* knock-out (*Wee1^{fl/fl}*)

mammary cells at LD2, detected by FACS. **b)** Quantification of the percentage of GFP positive (GFP⁺) within the CK8 positive population (CK8⁺) from *wild-type*

(*Wee1^{+/+}*) or *Wee1^{fl/fl}* MGs at LD2. **c)** Gating strategy for the identification of CK8⁺/GFP⁺ single cells based on propidium iodide with (PI-W) and area (PI-A). **d)** Representative FACS histograms for DNA content analysis showing the diploid (2C), tetraploid (4C) and polyploid (>4C) cells within the CK8⁺/GFP⁺ population from *Wee1^{+/+}* or *Wee1^{fl/fl}* MGs at LD2. **e)** Quantification of the percentage of 2C, 4C or >4C cells within the within the CK8⁺/GFP⁺ population from *Wee1^{+/+}* or *Wee1^{fl/fl}* MGs at LD2. Data shown as the mean ± S.D.; n=3 biological replicates; two-tailed Student's t-test. p values: ** < 0.01.

b) Why was only one allele of Wee1 deleted? In Figure 8e, it looks as if one gland has been suckled recently and the other not. How did the authors control for milk removal by the pups. Did pups suckled by Wee1fl/+ dams thrive?

To address the Reviewers' comments, in the revised manuscript, we present data from experiments performed after homozygous *Wee1* deletion (*Wee1^{fl/fl}*), culling the litters such that each dam had 5 pups. We attempted to evaluate pup weight, however, when nursed by *Wee1^{fl/fl}* dams, pups do not display milk in their stomachs and do not survive past LD2 (Response Figure 9a, b). We also find decreased expression of milk protein genes (*Csn2*, *Wap*, *Lalba*, *Plin2*, *Xdh1* and *Btn1*) and reduced overall milk production at LD2 (Response Figure 9c-f). These data, together with the DNA content analysis showing that *Wee1* loss decreases the 4C and >4C populations (Response Figure 8d, e), demonstrate that *Wee1* plays a crucial role in alveolar expansion and endoreplication. While we do not observe a complete loss of endoreplication, likely due to incomplete homozygous deletion of *Wee1* in our conditional knockout model, the loss of the pups suggests that they cannot suckle the little milk produced by the *Wee1^{fl/fl}* alveolar structures. Indeed, we also find the *Wee1^{fl/fl}* mammary glands are structurally deficient as they have ~55% fewer alveoli containing lumens, and those lumens are ~61% smaller (Response

Figure 9g,h). This structural deficiency results in an overall ~83% decrease in fat pad filling, as detected by IHC (Response Figure 9i). In the revised manuscript, we have included these data in Revised Figure 7.

Figure 9. Wee1 loss inhibits endoreplication and milk production. **a**) Quantification of the body weight (g) of pups nursed by *Wee1*^{+/+} or *Wee1*^{fl/fl} dams. **b**) Representative pictures of pups nursed by *Wee1*^{+/+} (top) or *Wee1*^{fl/fl} (bottom) dams. Black broken-line squares indicate the stomach of the pups. **c**) Quantification of milk genes expression (*Lalba*, *Wap*, *Csn2*, *Plin2*, *Xdh1* and *Btn1*) in *Wee1*^{+/+} or *Wee1*^{fl/fl} MGs at LD2. **d** and **e**) Detection of milk (white) in LD2 MGs from *Wee1*^{+/+} or *Wee1*^{fl/fl} mice. CK8 shown in magenta. Yellow squares indicate the magnified area shown in **(e)**, illustrating the milk contained within the alveoli. Images are representative of n=3. **f-i**) Quantification of **(f)** total milk Integrated Density (IntDen) per MG area, **(g)** number of alveoli with lumens per MG area, **(h)** alveolar lumen size, and **(i)** percentage of MG fat pad occupied by epithelia, as detected by IHC in *Wee1*^{+/+} or *Wee1*^{fl/fl} LD2 MGs. Data shown as the mean ± S.E.M; n=3 for biological replicates; two-tailed Student's t-test. p values: * < 0.05, ** < 0.01.

Despite these phenotypic differences between *Wee1*^{+/+} and *Wee1* knock-out mammary glands, FACS analysis reveals that the proportion of CK8+ luminal cells remains unchanged between these genotypes (Response Figure 10a). This strongly suggests that defects observed in alveoli formation are not due to defects in proliferation, but rather due to defects in endoreplication. Endoreplication likely allows for tissue growth and expansion in the absence of cell proliferation, since tetraploid (4C) and polyploid cells (>4C) are larger than diploid (2C) cells¹⁸. Indeed, by FACS analysis of the scatter properties of CK8+ at LD2, we find there is a decrease in the percentage of cells with high scatter parameters (directly correlating to cell size and complexity) in the *Wee1*^{fl/fl}, compared to *Wee1*^{+/+}, mammary glands (Response Figure 10b, c). Altogether, these data suggest that mammary gland endoreplication is crucial, not only for milk production, but also for the proper development of alveolar structures during lactation. In the revised manuscript, these new observations are included in Revised Figures 7 and 8.

Figure 10. Loss of *Wee1* impairs alveolar formation during lactation. **a)** Quantification of the percentage of cells from *Wee1*^{+/+} or *Wee1*^{fl/fl} LD2 MGs that are CK8 positive, as detected by FACS. **b)** Representative dot plots showing the SSC-A and FSC-A parameters of the CK8 positive population from *Wee1*^{+/+} or *Wee1*^{fl/fl} LD2 MGs, as detected by FACS. Gate represents the cells with high scatter parameters. **c)** Quantification of the percentage of CK8 positive cells from *Wee1*^{+/+} or *Wee1*^{fl/fl} LD2 MGs with high scatter (HS) parameters, as detected by FACS. Data shown as the mean \pm S.D.; n=3 biological replicates; two-tailed Student's t-test. p values: * < 0.05; ** < 0.01.

c) It is not clear to me why proliferation during pregnancy would result in DNA damage and the generation of polyploid and binucleate cells that continue to proliferate. Given a proposed rationale for cells undergoing a further round of DNA replication at the onset of lactation is to produce more ribosomes for the extensive increase in protein production required during lactation, it does not make sense to me that endoreduplication/ endoreplication would occur during mid-pregnancy where one would expect cells harboring DNA damage to exit the cell cycle and stop proliferating. Could the authors explain their observations? The title refers only to lactation.

We agree that it would not make sense for cells with DNA damage to continue to proliferate. Indeed, based on our studies, this is not the model we propose. While we observe that the peak of

DNA damage occurs at mid-pregnancy (PD10.5), it extends to PD15.5 (Response Figure 4d-f). Our data further show that the response to this DNA damage extends even later as pT1989-ATR expression peaks at PD18.5 and pS345-CHK1 at LD2 (Response Figure 4a, c, f). Since polyploidization begins occurring in the final days of pregnancy and during lactation¹ (Figure 1 in the original manuscript), our data that shows *Wee1* expression increasing between PD17.5 and LD2 is in agreement with polyploidization limiting proliferation and enhancing the differentiation of alveolar cells shortly after birth. In the revised manuscript, we make the timing of events clearer.

Reviewer #2:

1) *In the last part the authors tried to show that the DNA damage response regulates endoreplication via WEE1. Although the authors did not claim WEE1 is only one, it gave an impression that WEE1 is a primary factor for regulating endoreplication. But WEE1 is not identified by an unbiased screening rather than through a candidate approach. Indeed, the WEE1 inhibitor treatment or WEE1 +/- conditional knockout only partially decreased the 4C population, which might suggest the involvement of some other factors. In this regard, the author should also test WEE1 in HC11 cells to see if WEE1 inhibitor and knockdown could affect partially or completely, the spontaneous or induced tetraploid cell formation.*

Our rationale for performing a candidate approach rather than an unbiased screening is that, for endoreplication to occur, a cell must first undergo a prolonged arrest of the mitotic cell cycle at the G2/M transition. This G2/M arrest requires inhibition of both CDK1 activity and progression into the M phase. G2/M arrest-driven endoreplication events are thus mediated by different CDK1 inhibitors, with the specific endogenous inhibitor responsible being dependent on the tissue context^{19,20}. Thus, rather than an unbiased screening for genes that may be differentially regulated during the process of endoreplication, we specifically sought to identify the endogenous CDK1 inhibitor responsible for G2/M arrest in the context of alveolar endoreplication. In our original manuscript, we show that *Wee1* is the only endogenous CDK1 inhibitor upregulated during late pregnancy and early lactation (Figure 7 and Supplementary Figure 9), when alveolar endoreplication occurs, and that this upregulation of *Wee1* is specific to the luminal compartment. Thus, we focused on this particular CDK1 inhibitor.

We appreciate Reviewer #2's suggestion of testing our model on HC11 cells by using the WEE1 inhibitor. Reviewer #1, however, did not find HC11 cells to be a suitable model to test our hypothesis. Therefore, we performed experiments *ex vivo* in mammary gland organoids treated with the WEE1 inhibitor (Response Figure 2). The data show that WEE1 pharmacological inhibition decreases the percentages of cells with 4C and >4C DNA from ~7.5% to ~2.5% and ~5% to ~1.7%, for overall decreases of ~66% in both populations. In the revised manuscript, we include these data in Revised Figure 6.

Having piloted our experiments with a pharmacological inhibitor of WEE1, in the revised manuscript, we present data from the *Wee1* knock-out (*Wee1^{fl/fl}*) animals (Response Figures 8-10). Pups nursed by *Wee1^{fl/fl}* dams did not display milk in their stomachs and did not survive past LD2 (Response Figure 9a, b). As a result, we performed DNA content analysis at this earlier timepoint. We find that loss of *Wee1* reduces the percentage of cells with 4C and >4C DNA from ~16% to ~7% and from ~12% to ~4%, for an overall decreases of ~56% and ~67%, respectively (Response Figure 8d, e). While we do not observe a complete loss of endoreplication, this is

likely due to incomplete homozygous deletion of *Wee1* in our conditional knockout model. Nevertheless, this loss of *Wee1* decreases the expression of milk protein genes (*Csn2*, *Wap*, *Lalba*, *Plin2*, *Xdh1* and *Btn1*) and overall milk production at LD2 (Response Figure 9c-f). The loss of the pups suggests that they cannot suckle the little milk produced by the *Wee1^{fl/fl}* alveolar structures. Indeed, we also find the *Wee1^{fl/fl}* mammary glands are structurally deficient as they have ~55% fewer alveoli containing lumens, and those lumens are ~61% smaller (Response Figure 9g,h). This structural deficiency results in an overall ~83% decrease in fat pad filling, as detected by IHC (Response Figure 9i).

Despite these phenotypic differences, FACS analysis reveals the proportion of CK8+ luminal cells remains unchanged between *WT* and *Wee1^{fl/fl}* mammary glands (Response Figure 10a). This strongly suggests that the defects observed in alveoli formation are not due to defects in proliferation, but rather due to defects in endoreplication. Endoreplication likely allows for tissue growth and expansion in the absence of cell proliferation, since tetraploid (4C) and polyploid cells (>4C) are larger than diploid (2C) cells¹⁸. Indeed, by FACS analysis of the scatter properties of CK8+ at LD2, we find there is a decrease in the percentage of cells with high scatter parameters (directly correlating to cell size and complexity) in the *Wee1^{fl/fl}*, compared to *Wee1^{+/+}*, glands (Response Figure 10b, c). Altogether, these data suggest that mammary gland endoreplication is crucial, not only for milk production, but also for the proper development of alveolar structures during lactation. In the revised manuscript, we have included these data in Revised Figures 7-8.

2) *CHK1* is also an important *ATR* downstream gene that is involved in *DDR*. Why it was not checked? It should also be examined as a candidate.

Thank you for this suggestion. In the revised manuscript, we examine the presence of pS345-CHK1, the activating site of CHK1 that is phosphorylated by ATR, during pregnancy and early lactation^{5,6}. By both immunohistochemistry and immunoblotting, we find that pS345-CHK1 is upregulated between PD18.5 and LD2, when phosphorylation at this site reaches its peak (Response Figures 4a, f and 7b, e). This upregulation follows the peak of pT1989-ATR, the site at which ATR is activated by auto-phosphorylation (Response Figures 4c, f and 7a, d), and coincides with the acceleration of endoreplication (Figure 1f in the original manuscript)²¹. Together, these data strongly suggest that the ATR pathway is responsible for the induction of endoreplication during alveologenesis in response to replication stress, and that this occurs through the downstream signaling effector CHK1. This is consistent with the previously published role of CHK1 in phosphorylating and activating WEE1, allowing WEE1 to inhibit CDK1 activity and induce mitotic arrest at the G2/M checkpoint (Response Figure 11, ²²). In the revised manuscript, we have included these data in Revised Figure 3 and Revised Supplementary Figure 7.

Figure 11. Schematic representation of the role of WEE1 in the G2/M checkpoint. (Matheson et al., *TIPS* 2016, Figure 1) ²³.

3) An unbiased screening using CRISPR-Cas9 mediated approach might be good for identify some more candidate genes, although WEE1 could still be focused as a main factor.

As addressed above, we are focusing on the arrest of the mitotic cell cycle at the G2/M checkpoint via CDK1 inhibition, which is required to facilitate endoreplication. For this reason, we opted for a candidate approach to identify the CDK1 inhibitor involved in mammary alveolar endoreplication, rather than an unbiased screening approach.

4) In this case, the current data just showing “DNA damage response regulates endoreplication via WEE1” is preliminary. The molecular mechanism underlying such a regulation should be investigated.

As addressed above, the revised manuscript includes a more extensive analysis of an array of DNA damage response kinases and signaling effectors (Response Figures 4 and 7). In the original manuscript, we present data showing *Wee1* expression was upregulated between PD18.5 and LD2 in the mammary gland luminal compartment and modulated in response to agents that increased or relieved DNA damage (Figure 7a-c). In addition, we have now collected new data by immunoblotting, showing that WEE1 is upregulated by LD2 at the protein level (Response Figure 12a, b). WEE1 upregulation occurs concomitantly with the inhibitory phosphorylation of CDK1 on Tyr15 (Response Figure 12a, c). Moreover, we show downregulation of both WEE1 and pTyr15-CDK1 in *Wee1^{f/f}* mammary glands (Response Figure 12d-f).

With the additional data presented in the revised manuscript, we believe our data support a model in which ATR is activated in response to replication stress, generated by the massive proliferation of luminal cells during early pregnancy. ATR then phosphorylates and activates its canonical downstream signaling effector CHK1, which, in turn, phosphorylates and activates WEE1 by LD2. This signaling cascade culminates in the inhibition of CDK1 by WEE1, resulting in mitotic arrest at the G2/M checkpoint. It has been previously established that prolonged G2/M arrest leads to a cyclin switch in endoreplicative tissues, specifically a switch whereby Cyclin E is upregulated, while Cyclin B is downregulated ²⁴. Indeed, in the original manuscript, we showed in that the induction of differentiation and endoreplication in HC11 results in a ~25-fold increase in the ratio of Cyclin E/Cyclin B by DIP2, in comparison to the primed control (Supplementary Figure 3b). In the revised manuscript, we present new data showing that this cyclin switch from Cyclin B to Cyclin E expression occurs in the mammary gland during lactation (Response Figure 12g, h). Together, we believe our data support a molecular

mechanism by which the DNA damage response-induced activation of WEE1 drives alveolar endoreplication through CDK1 inactivation. In the revised manuscript, these data are included in Revised Figures 2 and 6 and Revised Supplementary Figure 12.

Figure 12. CDK1 inhibition occurs at the onset of lactation. **a-c)** Western blot (**a**) and quantification (**b** and **c**) of WEE1, CDK1 and pCDK1 (Y15) expression in nulliparous (Nullip.), PD10.5, PD15.5, PD18.5 and LD2 MGs. **d-f)** Western blot (**d**) and quantification (**e** and **f**) of WEE1, CDK1 and pCDK1 (Y15) expression in *Wee1*^{+/+} or *Wee1*^{fl/fl} MGs at LD2. **g** and **h)** Western blot (**g**) and quantification (**h**) of CYCLIN E and CYCLIN B expression in nulliparous (Nullip.), PD10.5, PD15.5, PD18.5 and LD2 MGs. Data shown as the mean ± S.D.; n=3 biological replicates; two-tailed Student's t-test. p values: * < 0.05; ** < 0.01.

5) Authors claims that the ATR-DDR is the major mediator for the endoreplication. However, there is no experiment data to exclude ATM. Inhibitors for ATM and ATR should be used in the study and their effects be compared to justify the claim.

As addressed in response to Reviewer #1, we focused on ATR due to the fact it is activated in response to single-stranded DNA produced by replication stress¹⁵. In the revised manuscript, we have included additional immunohistochemistry and immunoblotting analysis of pS1981-ATM, the auto-phosphorylated, activating site of ATM, in nulliparous, PD10.5, PD15.5, PD18.5 and LD2 mammary gland tissue (Response Figures 4 and 7; 7). By immunohistochemistry, we find that pS1981-ATM is highest in nulliparous tissue and is downregulated at the onset of lactation,

when endoreplication accelerates (Response Figure 4b, f). This is confirmed by immunoblotting, where we see a steady decrease in the ratio of pS1981-ATM:ATM from nulliparous tissue, through pregnancy, to the onset of lactation (Response Figure 7c, f). In contrast, pT1989-ATR, the auto-phosphorylated, activating site of ATR, is upregulated at PD10.5, when mammary gland epithelial proliferation is at its peak, and persists until late pregnancy, when endoreplication begins to occur (Response Figures 4c, f and 7a, d; ²¹). Taken together with the analysis of pS345-CHK1 (Response Figures 4a, f and 7b, e), as addressed above, these data support our model that the ATR pathway and not the ATM pathway is responsible for the induction of endoreplication during alveologenesis in response to replication stress, and that this occurs through the downstream signaling effector CHK1. In the revised manuscript we have included these data in Revised Figure 3 and Revised Supplementary Figure 7.

We appreciate Reviewer #2's suggestion of using ATM and ATR inhibitors to identify the pathway that is activated during mammary alveologenesis. However, it has been shown that ATR inhibition results in increased DNA damage due to replication stress, due to the essential role of ATR in safeguarding DNA integrity during replication, and the activation of the ATM pathway ²⁵. Even more, this study also shows that the combined use of ATR, ATM and DNA-PK inhibitors is needed to completely shut down the DNA damage response, most likely due to the cross talk between the different DNA damage response pathways ²⁵. Therefore, the use of these inhibitors is unlikely to bring new insights into which pathway (ATR or ATM) is responsible for the induction of endoreplication during alveologenesis.

6) Some figures are too much simplified with only a few panels although some data are placed in the supplementary figures. However, it makes it difficult to find the key data. For example, Figure 3 indicated that "doxorubicin-injected MGs at PD17.5 shows an increase in the proportion of 4C and > 4C luminal populations (14.86% and 47.1% increase in the overall population, respectively)", which is a piece of key data, but it was not in Figure 3. Figure 3i only showed minor changes. It is suggested to enrich these figures by adding some data from related Supplementary figures.

In the original manuscript, we presented the data as raw values rather than normalized values. We showed that the percentage of luminal cells with >4C DNA content increases from ~4% to ~6% with doxorubicin treatment, for an overall increase of 47.1% (Figure 3f). It is true, however, that the data regarding the percentage of luminal cells with 4C DNA content was located in the Supplementary Figures. We agree with Reviewer #2 that some figures would be enriched by the inclusion of related Supplementary Figures, however, the figure limit forces us to show what we believe to be the key pieces of data. In this case, we wanted to place the focus on the polyploid (>4C DNA content) population, rather than the tetraploid (4C DNA content) population, because it cannot be determined without a reasonable doubt whether a tetraploid cell has undergone endoreplication or whether it will eventually divide to produce two diploid (2C DNA content) cells. Additionally, while we did observe robust effects on endoreplication and milk production with doxorubicin treatment, Reviewers #1 and #4 considered doxorubicin a "blunt tool" because it causes broad DNA damage. Consequently, in the revised manuscript we have placed all the doxorubicin data into the Supplement (Revised Supplementary Figures 5 and 6). In the original manuscript, we also examined the consequences of physiological DNA damage due to replication stress by performing intraductal injection experiments in which we depleted or supplemented nucleoside pools through treatment with hydroxyurea or nucleosides, respectively (Figures 5 and 6 in the original manuscript). We have now complemented these *in vivo*

experiments with *ex vivo* experiments on mammary organoids (Response Figure 2) and included them in the revised manuscript in Revised Figure 6. All these experiments show that increasing (hydroxyurea treatment) or decreasing (nucleosides treatment) replication stress results in increased or decreased DNA damage, ATR activation, endoreplication and milk production, respectively (Revised Figures 4, 5 and 6 and Supplementary Figures 9 and 10). Taken together, the hydroxyurea and nucleoside injection results are more demonstrative of our model in which replication stress during early pregnancy drives endoreplication during late pregnancy and lactation.

Reviewer #3:

This paper addresses the role of the DNA damage response pathway in polyploidy in functionally differentiated mammary epithelial cells. The authors show increased ploidy in mammary luminal epithelial cells throughout differentiation and this response is dependent in part upon replication stress and Wee1 kinase activity. Major concerns are that the majority of the observations have been previously published (Visvader et al Nat Comm, 2016). This study showed increased ploidy throughout mammary gland pregnancy and lactation and was dependent upon Aurora A kinase, which has recently been shown to also play a role in replication stress and activation of the CDK1 pathway (Wang et al, Reproductive Biol and Endo, 2021).

Reviewer #3 is correct in that there was a 2016 publication in *Nature Communications* showing that a large proportion of mammary alveolar cells become tetraploid and binucleated during lactation and that binucleation is required for efficient milk production¹. This paper also suggested that tetraploid binucleated alveolar cells were achieved through Aurora A kinase-mediated cytokinesis failure at the abscission checkpoint. However, a major conclusion of our manuscript is that the population of polyploid cells generated during pregnancy is much more heterogeneous than that described in the Rios paper. The mammary gland during pregnancy is **not composed of only tetraploid binucleated alveolar cells**. And, while the abscission checkpoint mechanism identified by Rios and colleagues may explain the generation of those tetraploid binucleated alveolar cells, it does not explain how the cells we identify — **mononucleated polyploid and binucleated >4C polyploid cells** — are created. Addressing the mechanism responsible for this heterogeneous polyploidization, we show that physiological DNA damage, caused during pregnancy by replication stress, stimulates the DNA damage kinase, ATR. This, in turn, activates WEE1 that inhibits CDK1, resulting in cell cycle arrest and the observed heterogeneous polyploidization.

Reviewer #3 also suggests that a second paper contains results that are similar to ours²⁶. This paper by Wang and colleagues examined a different model system — decidualization in uterine stromal cells — and showed using pharmacological inhibition that Aurora A regulates the phosphorylation CDK1 at Tyr15 through STAT5 and PLK1. Interestingly, although the paper doesn't mention WEE1 or replication stress, Tyr15 is, indeed, an inhibitory site that is phosphorylated by WEE1^{27,28}. Thus, the mechanism identified by Rios and colleagues, in which Aurora A is activated, may also feed into the mechanism we identify — possibly through the STAT5-PLK1 pathway identified by Wang and colleagues. If this is the case, then the endpoint for both pathways is the inhibition of CDK1 via WEE1, which our manuscript identifies. Thus, this activation of WEE1 may explain the generation of both the tetraploid binucleated alveolar cells identified by Rios — **and** the mononucleated polyploid and binucleated >4C polyploid cells

that our manuscript identifies. **We note that neither the Rios nor Wang papers mention replication stress as the source of DNA damage, neither identify heterogenous populations of polyploid cells and neither identify WEE1 as the CDK1 inhibitor responsible for endoreplication occurring at the G2/M checkpoint.** Thus, our manuscript offers numerous new insights, which is likely why no other reviewer raised concerns about its novelty.

Specific comments:

1. *The number of mice in each litter was not documented.*

Thank you for pointing this out. In the revised manuscript we present data from experiments performed after homozygous *Wee1* knock-out (*Wee1^{fl/fl}*), culling the litters such that each dam had 5 pups. We find that pups nursed by *Wee1^{fl/fl}* dams do not display milk in their stomachs and do not survive past LD2 (Response Figure 9a, b). We also find decreased expression of milk protein genes (*Csn2*, *Wap*, *Lalba*, *Plin2*, *Xdh1* and *Btn1*) and overall milk production at LD2 (Response Figure 9c-f). These data, together with the DNA content analysis showing that *Wee1* loss decreases the 4C and >4C populations (Response Figure 8), demonstrate that *Wee1* plays a crucial role in alveolar expansion and endoreplication. In the revised manuscript we have indicated the number of pups being nursed and included these results in Revised Figures 7 and 8 and Revised Supplementary Figures 11 and 12

2. *The quantification of milk in the gland from staining is not a reliable measure of mammary gland function. Milk expression is highly dynamic and dependent upon the litter size and when the pups last nursed, which was not documented. The use of normalized litters and pup weight would provide a better measurement of mammary gland function.*

Thank you for your suggestion. We did not include pup weight in the original manuscript due to the fact that previously published papers have shown that increased IHC staining of total mouse milk proteins was reflective of increased pup weight^{1,2}. Furthermore, for experiments in which we intraductally injected drugs, these were contralateral injections, so control and treated mammary glands were harvested from the same mother, and thus exposed to the same number of suckling pups. In the revised manuscript, we present our data on the *Wee1^{fl/fl}* conditional knockout animals (see response to point 1). We culled the litters to 5 pups, but none displayed milk in their stomachs or survived past LD2 (Response Figure 9a, b). In the revised manuscript, we have included these data in Revised Figure 7.

3. *It isn't clear why the authors utilized the CK8 Tamoxifen-induced Cre when other more reliable lactation and mammary specific Cre drivers are available (i.e. WAP-Cre). The use of the CK8 Cre may give rise to non-mammary specific phenotypes. Moreover, the WAP-Cre would also allow for a more detailed impact on mammary gland function by measuring changes in pup weights.*

We considered using a different Cre driver, but employed the *CK8-CreER* driver for a number of reasons. First, we performed our DNA content FACS analysis on the CK8⁺ population, because we had technical difficulties staining for the secreted protein WAP. Therefore, we wanted *Wee1* to be deleted in the same CK8 population that we analyzed by FACS. By analyzing the different CK8⁺ populations within the mammary gland, we found that the alveolar cells undergoing

endoreplication between PD17.5 and LD5 express high levels of CK8 (Response Figure 1, CK8^{hi} subpopulation). Hence, we are confident that using the CK8-CreER driver allows us to specifically target alveolar cells undergoing endoreplication. We have now included this analysis in the revised manuscript in Revised Supplementary Figure 11. Second, the expression of *Wap* has been shown to occur as early as PD10, when the mammary epithelium is still undergoing massive proliferation. As a key cell cycle regulator, WEE1 is undoubtedly implicated in the safeguarding of genomic integrity during this proliferation, and its depletion this early in pregnancy would likely have detrimental effects on the development of the mammary gland during pregnancy. Thus, we reasoned that our investigation into the role of WEE1 during alveolar endoreplication would be best served by selectively depleting *Wee1* after the hyperproliferation phase of mid-pregnancy, but before the onset of endoreplication during late-pregnancy and early lactation. The constitutively active *Wap-Cre* model would not allow for this, so, instead, we chose the *CK8-CreER* model. Finally, while it is true using this model could give rise to non-mammary phenotypes, this concern is mitigated by the fact that we induce *Wee1* depletion at PD17 and analyze the mammary gland tissue at LD2, resulting in a relatively short pulse-chase period of ~4 days.

4. The authors do not show controls (i.e. dye) for the IDI experiments how much of the injected volume is distributed throughout the gland.

We did not include controls in the original manuscript due to the fact that previously published studies have shown dye controls that demonstrate the efficacy and accuracy of this technique using similar volumes of solutions^{29,30}. In the revised manuscript, we show trypan blue dye controls demonstrating that 40uL of solution intraductally injected at PD17 is distributed throughout the gland (Response Figure 13). In the revised manuscript, we have now included this control in Revised Supplementary Figure 6.

Trypan blue IDI Control

Figure 13. Trypan Blue intraductal injection control. Intraductal injection with trypan blue at PD17 shows efficient delivery of small molecules throughout the ductal tree.

5. The addition of additional markers for differentiation and involution should be included (p-STAT5, p-STAT3, etc).

To address the Reviewer #3's comment, we examined the expression of pSTAT5/STAT5 and pSTAT3/STAT3 in lysates harvested from LD2 mammary glands of *Wee1^{-/+}* and *Wee1^{fl/fl}* mice. We find a substantial reduction in pSTAT5/STAT5 in lysates from *Wee1^{fl/fl}* mammary glands, consistent with reduced alveologenesis and lactogenesis (Response Figure 14a-c). We also observe increased pSTAT3/STAT3 that is consistent with early involution, which is likely occurring because pups are not able to suckle milk from *Wee1^{fl/fl}* dams (Response Figure 14a, d and e). Altogether, the results from the examination of these markers of differentiation (pSTAT5/STAT5) and involution (pSTAT3/STAT3) are consistent with the observed phenotype of the *Wee1^{fl/fl}* animals. In the revised manuscript, we have included these data in Revised Figure 8 and Revised Supplementary Figure 12.

Figure 14. Loss of Wee1 inhibits STAT5 activation while promoting STAT3 pathway. a-e Western blot (a) and quantification (b-e) of pSTAT5 (Y694), STAT5, pSTAT3 (Y705) and STAT3 expression in in *Wee1^{-/+}* or *Wee1^{fl/fl}* MGs at LD2. Data shown as the mean \pm S.D.; n=3 biological replicates; two-tailed Student's t-test. p values: * < 0.05; ** < 0.01.

Reviewer #4

4. The model described, and visualized in Fig 8, is not clear based on the experimental conditions. The authors induce DNA damage using doxorubicin, in order to mimic replication induced stress. In this case they should have chosen a compound that generates primarily single stranded DNA breaks. Doxorubicin, however, creates predominantly double strand DNA breaks, as it is a Topoisomerase II poison. The authors appear to be inducing a different set of breaks and stimulating a different cellular response to DNA damage than what they describe in their model, which is ATR based recognition of DNA damage. The schematic also shows double strand DNA breaks. The authors should use a compound that more accurately stimulates single strand breaks.

[Capranico G, De Isabella P, Penco S, Tinelli S, Zunino F. Role of DNA breakage in cytotoxicity of doxorubicin, 9-deoxydoxorubicin, and 4-demethyl-6-deoxydoxorubicin in murine leukemia P388 cells. *Cancer Res.* 1989;49(8):2022–2027.]

In the original manuscript, we treated with doxorubicin to broadly damage DNA and showed that this increases endoreplication and milk production. However, this is not the central finding of our research; therefore, in the revised manuscript we move all data collected with doxorubicin to the Supplement (Revised Supplementary Figures 5 and 6). Instead, the central finding is that replication stress resulting in physiological DNA damage is sufficient to increase endoreplication and milk production. We demonstrated this in the original manuscript by performing intraductal injection experiments in which we depleted or supplemented nucleoside pools by injecting hydroxyurea or nucleosides, respectively (Figures 5 and 6 and Supplementary Figure 8). In the revised manuscript, we have complemented these *in vivo* experiments with *ex vivo* experiments on mammary organoids (Response Figure 2). All these experiments show that increasing (hydroxyurea treatment) or decreasing (nucleoside treatment) replication stress results in increased or decreased, respectively, DNA damage, ATR activation, endoreplication and milk production. In the revised manuscript, we include these data in Revised Figures 4-6 and Revised Supplementary Figures 9 and 10. We also make clearer the distinction in rationale behind the doxorubicin and hydroxyurea treatments, as well as corrected the schematic (Revised Figure 9) to show single-stranded DNA that would be generated by replication stress.

5. *The authors use gamma H2AX, which can be a marker for double strand DNA breaks, and is also phosphorylated by ATM on serine 139. The authors should determine if ATM is phosphorylated in the mammary gland at pregnancy day 10.5 as well as ATR.*

While γ H2AX is often characterized as being phosphorylated at the sites of double-stranded breaks, it has also been established that ATR phosphorylates H2AX at sites of single-stranded DNA produced by replication stress³¹. As addressed in response to Reviewer #1, we focused on ATR due to the fact it is activated in response to single-stranded DNA produced by replication stress¹⁵. In the revised manuscript, we include additional immunohistochemistry and immunoblotting of pS1981-ATM (the auto-phosphorylated, activating site of ATM) on tissue collected from nulliparous, PD10.5, PD15.5, PD18.5 and LD2⁷. By immunohistochemistry, we find that the presence of pS1981-ATM is highest in nulliparous tissue and is downregulated at the onset of lactation, when endoreplication accelerates (Response Figure 4b, f). This is confirmed by immunoblotting, where we see a steady decrease in the ratio of pS1981-ATM:ATM from nulliparous tissue, through pregnancy, to the onset of lactation (Response Figure 7c, f). In contrast, by immunohistochemistry pT1989-ATR (the auto-phosphorylated, activating site of ATR) is upregulated at PD10.5, when mammary epithelial proliferation is at its peak, and this immunostaining persists until late pregnancy, when endoreplication begins to occur (Response Figure 4c, f; ²¹). By immunoblotting, we observe high expression at PD10.5, a dip in expression at PD15.5, but recovery of high expression at PD.18.5 before pT1989-ATR dips again at LD2 (Response Figure 7a, d). Thus, taken together, our data show that the ATR pathway, and not the ATM pathway, is activated in response to DNA damage during the hyperproliferation of mid-pregnancy occurring in the mammary gland. In the revised manuscript, we have included these data in Revised Figure 3 and Revised Supplementary Figure 7.

6. *Page 6, "In the MG, DNA damage occurs in alveolar cells during pregnancy 35,..." What type of DNA damage is known to occur in alveolar cells during pregnancy? Is it predominantly single or double stranded breaks? Which DNA damage pathway is expected to be induced? Is it both ATM and ATR-based?*

DNA damage in alveolar cells during pregnancy has not been well characterized. Xu and colleagues reported the presence of γ H2AX⁺ cells in pregnant mammary epithelia (Supplementary Figure 1d) and suggested in their discussion that this DNA damage may be generated by replication stress due to fast cell proliferation³². Here, we have done a much more extensive study and we show that this is, indeed, the case. Because γ H2AX is marker of broad damage (both double- and single-stranded breaks), in the revised manuscript we include an immunohistochemical analysis of RPA, a protein complex that binds single-stranded DNA and facilitates the recruitment of ATR to these sites (Response Figure 6c, d;¹¹). We find that RPA is highest at PD10.5, which coincides with the peak of γ H2AX and upregulation of pT1989-ATR (Response Figure 6a, b, d). Furthermore, we examine the presence of pS345-CHK1, the activating site of CHK1 that is phosphorylated by ATR, during pregnancy and early lactation, and find that it is upregulated at the beginning of lactation (Response Figure 4a, f). Additionally, we include analysis of pS1981-ATM as well. We find that pS1981-ATM is highest in nulliparous mammary epithelia and is downregulated with pregnancy and the onset of lactation (Response Figure 4b, f and 7c, f). Together, these data suggest that predominantly single-stranded DNA gaps or breaks occur due to replication stress during the hyperproliferation of mid-pregnancy, and this leads to activation of the ATR-mediated, and not the ATM-mediated, DNA damage response pathway. These data are included in the revised manuscript in Figure 3 and Supplementary Figure 7.

7. The gamma H2AX staining is unclear. Very often images show staining as the whole nucleus, when it should be in punctate repair foci. Greater resolution is required.

Replication stress has been shown to lead to pan-nuclear gamma-H2AX staining in multiple cell lines, including mammary epithelial cell lines (HBL100 and MCF7)^{31,33}. Because HC11 cells are a highly proliferative mammary epithelial cell line, they display robust γ H2AX staining due to the DNA damage incurred by rapid proliferation. In the revised manuscript, we try to alter the brightness and contrast of these HC11 immunostainings to make clearer that punctate foci are observed, but it appears that we are also seeing pan-nuclear gamma-H2AX in these HC11 cells undergoing replication stress (Revised Supplementary Figures 4 and 5a). Please note that, in the original manuscript, H2AX appears as punctate foci in our *in vivo* immunostainings of optically-cleared tissue sections imaged using super-resolution confocal microscopy (Figures 3, 5 and 6 and Supplementary Figure 4).

8. Has the knockout of Wee1, or at least its reduced level of protein, been confirmed by western, or using immunofluorescence? The reduction in protein should be shown, rather than just at the transcript level.

In response to Reviewer #4's comment, we have now confirmed reduced WEE1 protein levels in *Wee1^{fl/fl}* mammary gland tissue by western blot (Response Figure 12d, e). Furthermore, we also observe decreased levels of pY15-CDK1, the inhibitory site that is phosphorylated by WEE1, confirming reduced WEE1 activity (Response Figure 12d, f;^{27,28}). In the revised manuscript, these data are now included in Revised Supplementary Figure 12.

Minor issues

1. Page 3 – the discussion in the introduction regarding the coupling of differentiation, cell

checkpoints (particularly the G1S checkpoint in the last paragraph) and DNA damage in the mammary gland would benefit from more detailed reference citation.

We appreciate the suggestion of Reviewer #4 and have cited references for this paragraph in the revised manuscript.

2. What is meant by 'differential' ability to build a milk supply (page 12).

In our Discussion we stated: “Our model offers an explanation as to why MGs have a differential ability to build a milk supply during pregnancy.” By “differential ability to build a milk supply”, we mean that the ability of a mother to sufficiently lactate and nourish offspring differs or varies according to circumstances and relevant factors. Given our findings, we believe differing levels of replication stress during the period of luminal cell proliferation at mid-pregnancy and the resulting differences in alveolar endoreplication represent one factor that may determine the amount of mother’s milk. Thus, our model offers an explanation as to why mothers can have a differential ability to build a milk supply during lactation.

3. There is very little information on the C57BL/6-Wee1<tm1.1 mrl> line.

We only identified one published Wee1 KO mouse line³⁴, but the authors of this paper would not share the line with us. Therefore, we purchased the Wee1 - Model 6592 - cKO C57BL/6-Wee1tm1.1 mrl conditional knockout from Taconic Biosciences. Below are the details of the mouse line. We explain the design of the mouse line more clearly in the Methods of the revised manuscript.

- The targeting vector has been generated using C57BL/6J DNA and transfected into TaconicArtemis C57BL/6NTac ES cell line.
- Exon 1 contains the translation initiation codon.
- Exons 3 to 5 have been flanked by loxP sites (size of loxP-flanked region: 1.9 kb).
- The conditional KO allele has been generated after Flp-mediated recombination by crossing chimeras to a Flp-Deleter on a C57BL/6 background.
- The conditional KO line has been derived using C57BL/6NTac animals and the Flp-transgene was removed by segregation.
- The constitutive KO allele can be generated by Cre-mediated recombination. Deletion of exons 3 to 5 should result in loss of function of the *Wee1* gene and generates a frame shift.

4. Was HC11 STR tested?

STR testing of mouse cell lines, such as the HC11 line, has not progressed to the extent of STR testing of human cell lines because many mouse strains are inbred and therefore isogenic, making it difficult to differentiate individual cell lines. Currently, HC11 cells have not been included in the NCBI Biosample directory of mouse STR profiles, so there is no verified reference for comparison.

For this manuscript, we purchased the HC11 cells directly from ATCC (CRL-3062™), just prior to initiating the studies. We expanded the line, froze numerous vials and thaw new,

low-passage cells regularly. We did not, however, STR test the cells when they were initially received in 2019; this technology was just becoming commercially available at the time (Almeida et al PLoS ONE 14(6): e0218412). The cell line purchased from ATCC behaves as expected and differentiates over the published time course in response to dexamethasone, insulin and prolactin.

References:

- 1 Rios, A. C. *et al.* Essential role for a novel population of binucleated mammary epithelial cells in lactation. *Nat Commun* **7**, 11400 (2016). <https://doi.org:10.1038/ncomms11400>
- 2 Cazares, O. *et al.* Alveolar progenitor differentiation and lactation depends on paracrine inhibition of notch via ROBO1/CTNNB1/JAG1. *Development* **148** (2021). <https://doi.org:10.1242/dev.199940>
- 3 Rubio, S., Cazares, O., Macias, H. & Hinck, L. Generation of Mosaic Mammary Organoids by Differential Trypsinization. *J Vis Exp* (2020). <https://doi.org:10.3791/60742>
- 4 Mehrotra, S., Maqbool, S. B., Kolpakas, A., Murnen, K. & Calvi, B. R. Endocycling cells do not apoptose in response to DNA rereplication genotoxic stress. *Genes Dev* **22**, 3158-3171 (2008). <https://doi.org:10.1101/gad.1710208>
- 5 Zhao, H. & Piwnicka-Worms, H. ATR-Mediated Checkpoint Pathways Regulate Phosphorylation and Activation of Human Chk1. *Molecular and Cellular Biology* **21**, 4129-4139 (2001). <https://doi.org:10.1128/mcb.21.13.4129-4139.2001>
- 6 Liu, Q. *et al.* Chk1 is an essential kinase that is regulated by Atr and required for the G₂/M DNA damage checkpoint. *Genes & Development* **14**, 1448-1459 (2000). <https://doi.org:10.1101/gad.14.12.1448>
- 7 Bakkenist, C. J. & Kastan, M. B. DNA damage activates ATM through intermolecular autophosphorylation and dimer dissociation. *Nature* **421**, 499-506 (2003). <https://doi.org:10.1038/nature01368>
- 8 Liu, S. *et al.* ATR Autophosphorylation as a Molecular Switch for Checkpoint Activation. *Molecular Cell* **43**, 192-202 (2011). <https://doi.org:10.1016/j.molcel.2011.06.019>
- 9 Hilton, B. A. *et al.* ATR Plays a Direct Antiapoptotic Role at Mitochondria, which Is Regulated by Prolyl Isomerase Pin1. *Mol Cell* **60**, 35-46 (2015). <https://doi.org:10.1016/j.molcel.2015.08.008>
- 10 Xu, A. F. *et al.* Subfunctionalized expression drives evolutionary retention of ribosomal protein paralogs. *Elife* **12** (2023). <https://doi.org:10.7554/eLife.78695>
- 11 Zou, L. & Elledge, S. J. Sensing DNA damage through ATRIP recognition of RPA-ssDNA complexes. *Science* **300**, 1542-1548 (2003). <https://doi.org:10.1126/science.1083430>
- 12 Byun, T. S., Pacek, M., Yee, M.-C., Walter, J. C. & Cimprich, K. A. Functional uncoupling of MCM helicase and DNA polymerase activities activates the ATR-dependent checkpoint. *Genes & Development* **19**, 1040-1052 (2005). <https://doi.org:10.1101/gad.1301205>

- 13 Mehrotra, S., Maqbool, S. B., Kolpakas, A., Murnen, K. & Calvi, B. R. Endocycling cells do not apoptose in response to DNA rereplication genotoxic stress. *Genes & Development* **22**, 3158-3171 (2008). <https://doi.org:10.1101/gad.1710208>
- 14 Gandarillas, A., Molinuevo, R. & Sanz-Gómez, N. Mammalian endoreplication emerges to reveal a potential developmental timer. *Cell Death Differ* **25**, 471-476 (2018). <https://doi.org:10.1038/s41418-017-0040-0>
- 15 Flynn, R. L. & Zou, L. ATR: a master conductor of cellular responses to DNA replication stress. *Trends Biochem Sci* **36**, 133-140 (2011). <https://doi.org:10.1016/j.tibs.2010.09.005>
- 16 Bester, A. C. *et al.* Nucleotide deficiency promotes genomic instability in early stages of cancer development. *Cell* **145**, 435-446 (2011). <https://doi.org:10.1016/j.cell.2011.03.044>
- 17 Halliwell, J. A. *et al.* Nucleosides Rescue Replication-Mediated Genome Instability of Human Pluripotent Stem Cells. *Stem Cell Reports* **14**, 1009-1017 (2020). <https://doi.org:10.1016/j.stemcr.2020.04.004>
- 18 Orr-Weaver, T. L. When bigger is better: the role of polyploidy in organogenesis. *Trends Genet* **31**, 307-315 (2015). <https://doi.org:10.1016/j.tig.2015.03.011>
- 19 Ullah, Z., Kohn, M. J., Yagi, R., Vassilev, L. T. & DePamphilis, M. L. Differentiation of trophoblast stem cells into giant cells is triggered by p57/Kip2 inhibition of CDK1 activity. *Genes Dev* **22**, 3024-3036 (2008). <https://doi.org:10.1101/gad.1718108>
- 20 Tan, J. *et al.* Evidence for coordinated interaction of cyclin D3 with p21 and cdk6 in directing the development of uterine stromal cell decidualization and polyploidy during implantation. *Mech Dev* **111**, 99-113 (2002). [https://doi.org:10.1016/s0925-4773\(01\)00614-1](https://doi.org:10.1016/s0925-4773(01)00614-1)
- 21 Nam, E. A. *et al.* Thr-1989 phosphorylation is a marker of active ataxia telangiectasia-mutated and Rad3-related (ATR) kinase. *J Biol Chem* **286**, 28707-28714 (2011). <https://doi.org:10.1074/jbc.M111.248914>
- 22 Elbæk, C. R., Petrosius, V. & Sørensen, C. S. WEE1 kinase limits CDK activities to safeguard DNA replication and mitotic entry. *Mutat Res* **819-820**, 111694 (2020). <https://doi.org:10.1016/j.mrfmmm.2020.111694>
- 23 Matheson, C. J., Backos, D. S. & Reigan, P. Targeting WEE1 Kinase in Cancer. *Trends Pharmacol Sci* **37**, 872-881 (2016). <https://doi.org:10.1016/j.tips.2016.06.006>
- 24 Zhang, Y., Wang, Z. & Ravid, K. The cell cycle in polyploid megakaryocytes is associated with reduced activity of cyclin B1-dependent cdc2 kinase. *J Biol Chem* **271**, 4266-4272 (1996). <https://doi.org:10.1074/jbc.271.8.4266>
- 25 Molinuevo, R., Freije, A., Contreras, L., Sanz, J. R. & Gandarillas, A. The DNA damage response links human squamous proliferation with differentiation. *J Cell Biol* **219** (2020). <https://doi.org:10.1083/jcb.202001063>
- 26 Wang, P. C., Chen, S. T. & Yang, Z. M. Effects of Aurora kinase A on mouse decidualization via Stat3-plk1-cdk1 pathway. *Reprod Biol Endocrinol* **19**, 162 (2021). <https://doi.org:10.1186/s12958-021-00847-5>
- 27 McGowan, C. H. & Russell, P. Human Wee1 kinase inhibits cell division by phosphorylating p34cdc2 exclusively on Tyr15. *EMBO J* **12**, 75-85 (1993). <https://doi.org:10.1002/j.1460-2075.1993.tb05633.x>
- 28 McGowan, C. H. & Russell, P. Cell cycle regulation of human WEE1. *The EMBO Journal* **14**, 2166-2175 (1995). <https://doi.org:10.1002/j.1460-2075.1995.tb07210.x>

- 29 Krause, S., Brock, A. & Ingber, D. E. Intraductal Injection for Localized Drug Delivery to the Mouse Mammary Gland. *Journal of Visualized Experiments* (2013). <https://doi.org:10.3791/50692>
- 30 Oliemuller, E., Newman, R. & Howard, B. A. Intraductal Injections into the Mouse Mammary Gland. *Methods Mol Biol* **2471**, 221-233 (2022). https://doi.org:10.1007/978-1-0716-2193-6_12
- 31 Ward, I. M. & Chen, J. Histone H2AX is phosphorylated in an ATR-dependent manner in response to replicational stress. *J Biol Chem* **276**, 47759-47762 (2001). <https://doi.org:10.1074/jbc.C100569200>
- 32 Xu, X. *et al.* BRCA1 represses DNA replication initiation through antagonizing estrogen signaling and maintains genome stability in parallel with WEE1-MCM2 signaling during pregnancy. *Hum Mol Genet* **28**, 842-857 (2019). <https://doi.org:10.1093/hmg/ddy398>
- 33 Gagou, M. E., Zuazua-Villar, P. & Meuth, M. Enhanced H2AX Phosphorylation, DNA Replication Fork Arrest, and Cell Death in the Absence of Chk1. *Molecular Biology of the Cell* **21**, 739-752 (2010). <https://doi.org:10.1091/mbc.e09-07-0618>
- 34 Vassilopoulos, A. *et al.* WEE1 murine deficiency induces hyper-activation of APC/C and results in genomic instability and carcinogenesis. *Oncogene* **34**, 3023-3035 (2015). <https://doi.org:10.1038/onc.2014.239>

Reviewer #1 (Remarks to the Author):

The authors have carried out a considerable amount of additional work, and consequently, have submitted a much improved manuscript. Most of my comments have been addressed to my satisfaction. However, the mechanism of the physiological (as stated in the title) DNA damage, particularly in early pregnancy, has not been demonstrated. I still think that the use of chemicals rather than genetic deletion of a candidate gene(s), other than WEE1, does not address the mechanism that occurs in the mammary gland. This is particularly relevant to the early pregnancy stages. Given previous work published on this topic, the authors could present new insights.

Reviewer #2 (Remarks to the Author):

The authors have addressed some of the points this reviewer raised but missed some others.

1. This reviewer still believes the claim that "doxorubicin-injected MGs at PD17.5 shows an increase in the proportion of 4C and > 4C luminal populations (14.86% and 47.1% increase in the overall population, respectively)" is a piece of key data, and it should be shown in the formal figure.

2. Because authors claimed that the ATR-DDR is the major mediator for the endoreplication, this reviewer suggested to compare potential effects of ATM and ATR inhibitors to obtain data to support this claim. The authors did not do it and cited a paper, which showed "that the combined use of ATR, ATM and DNA-PK inhibitors is needed to completely shut down the DNA damage response, most likely due to the cross talk between the different DNA damage response pathways". Therefore, they indicated that "the use of these inhibitors is unlikely to bring new insights into which pathway (ATR or ATM) is responsible for the induction of endoreplication during alveologenesis". This is not a proper argument, as the read out here is the "endoreplication", but not the "shut down" of the DDR. If neither ATR inhibitor nor ATM inhibitor could affect endoreplication because of the functional redundancy, the claim that "the ATR-DDR is the major mediator for the endoreplication" is questionable. If the authors think there will be too much work to carry this out in vivo, can this be done at least in the mammary organoids?

3. Previously, Vassilopoulos et al. generated a Wee1 mammary specific knockout mouse model and found that Wee1 plays a critical role in G2-M checkpoint and the Wee1 deficiency resulted in some abnormality in the mammary epithelial cells (Oncogene. 2015 Jun 4; 34(23): 3023-3035.). Although different Cre was used, this study might need to be cited and discussed, as it might strengthen the conclusion here.

This reviewer also raised the following 2 minor points, but they were not addressed:

1. Figure 1a: Please indicate in the figure legends regarding how the blue and yellow colour are generated, and also the stages of the sample (P17.5?).

2. Suggest also show a P10.5 samples as a control in this figure.

Reviewer #3 (Remarks to the Author):

This is a re-submission of a paper focused on determining the role of the DNA damage response pathway in polyploidy in functionally differentiated mammary epithelial cells. The authors have addressed many of the questions from the previous review including rationalization for using the CK8 conditional mouse model and providing new data showing changes in STAT5A/STAT3; however, there are still concerns that need to be addressed.

1. Measurement of pup weights is considered the gold standard for assessing lactation function. Small changes (1.5 fold) in milk protein expression does not indicate a significant impact in lactation performance. As indicated in the previous review, milk expression is highly dynamic and dependent upon the litter size and when the pups last nursed. The author commented on normalizing litter size for the Wee1^{fl/fl} studies, but provided no new information on the litter size or information on the last time the pups nursed for the injection studies. For this reviewer, the lack

of this information makes the data in Fig 5/6/7 not reliable. Moreover, areas assessed for milk in the gland (Fig 5H-I; Fig6I-J; Fig 7g-h) is highly selective and goes to issues of rigor. Why were these areas selected? Other areas could just have easily been selected to show the opposite result. 2. There are issues with the Western blots in the rebuttal. For example, in rebuttal Figure 7a and 7B, the authors show increased pATR i(nullip) and pATM (PD 10.5), but there isn't a band for the ATR and ATM loading control. This suggests issues with the sample and or assay that needs to be repeated to be interpretable

Reviewer #4 (Remarks to the Author):

This manuscript has been significantly revised, with the results of several important new high-quality experiments. The results are novel and impactful, as they address a potentially paradigm shifting concept of the role of DNA damage in cell fate. My previous concerns have been fully addressed and I highly support the publication of this manuscript.

Reviewer #1 (Remarks to the Author):

The authors have carried out a considerable amount of additional work, and consequently, have submitted a much improved manuscript. Most of my comments have been addresses to my satisfaction. However, the mechanism of the physiological (as stated in the title) DNA damage, particularly in early pregnancy, has not been demonstrated. I still think that the use of chemicals rather than genetic deletion of a candidate gene(s), other than WEE1, does not address the mechanism that occurs in the mammary gland. This is particularly relevant to the early pregnancy stages. Given previous work published on this topic, the authors could present new insights.

Thank you for acknowledging the work we put into the revision of our manuscript. Our study investigates the hypothesis that replication stress caused by prodigious proliferation occurring during pregnancy results in activation of the G2/M checkpoint and DNA re-replication in the absence of cell division (endoreplication). We proposed this hypothesis after identifying in the pregnant and lactating mammary gland a complex population of polyploid cells, including (but not limited to) mononucleated and binucleated cells with >4C nuclei. To test this hypothesis, we decided to intraductally inject pharmacological agents that could 1) generate DNA damage (Doxo), 2) generate replication stress (Hu) or 3) relieve DNA damage (Nucs). By doing these manipulations, we provide evidence that DNA damage promotes the generation of this complex population of polyploid cells, whereas relieving replication stress inhibits this process. In our opinion, this evidence would be difficult to obtain using genetic mutations that generate DNA damage because such mutations have been shown to result in pleiotropic consequences¹. Similarly, to our knowledge, there are no genetic mutations that solely relieve replication stress. In contrast, the approaches we pursued have been used successfully. Nucleoside supplementation has been shown to relieve replication stress²⁻⁴. Doxo has been used to study DNA damage⁵⁻⁷, and Hu has been used to investigate replication stress⁸⁻¹⁰.

The reviewer also appears to be suggesting that we target the specific kinases in the DNA damage response pathway. While there are genetic models for the DNA repair enzymes *ATM* and *ATR*, obtaining clear cut data from these models has proven difficult due to the broad range of these enzyme's activities and the pleiotropic consequences of knocking them out¹¹. For example, as our data in the manuscript and much data in the literature suggest, *ATR* plays a key role in responding to DNA damage by phosphorylating *CHK1*. This, in turn, reduces *CDK1* activity and arrests cell cycle progression to allow for DNA repair. At the same time, *ATR* also performs additional critical functions by preferentially phosphorylating other substrates, including many replication proteins (e.g., *RPA*, *MCM2*, *RFC* etc.) that promote replication fork stability, genomic integrity and DNA repair. *ATR* also shares many overlapping substrates with *ATM* (e.g. *H2AX*, *CTIP*, *FANCD2*)^{12,13}, and together they coordinate DNA repair with other DNA metabolic events (transcription, mitosis, replication)¹¹. This is likely why, when we tried to pharmacologically inhibit *ATR* in response to Reviewer's 2 comments (see below), we observe, as others have reported^{7,14,15}, that blocking *ATR* induces phosphorylation of γ H2AX, which suggests the induction, not suppression, of DNA damage. Indeed, consistent with increased DNA damage, we observe more cells with >4C content and elevated levels of *Csn2* and *Lalba*.

CDK1 is the other kinase that we identified as a downstream effector of the DNA damage response to replication stress in the mammary gland, and that the Reviewer may be referring to when they suggest we perform genetic deletion. Previous discoveries, however, have both already provided a solid foundation for our investigations and limited the novelty of insight that

could be gained by pursuing these studies. For example, analysis of the liver in a *Cdk1* conditional knockout model showed that endoreplication occurs¹⁶. This result, the accumulation of polyploid cells, was again found in human embryonic stem cells upon *Cdk1* knockdown by siRNA¹⁷. Thus, the literature demonstrates that loss of *Cdk1* leads to endoreplication and polyploidization. Consequently, we put our focus on investigating the novel aspects of the DNA damage response pathway in the mammary gland. Our efforts were rewarded when our pharmacological approach identified WEE1 as a candidate CDK1 inhibitor. While previous studies have shown roles for the various CDK1 inhibitors in regulating endoreplication in mammalian tissues, such a role for the CDK1 inhibitor WEE1 has only been observed in plants^{18,19}. Thus, our demonstration by pharmacological inhibition and by generation of a *Wee1* conditional knockout mouse model allowed us to identify a novel role for WEE1 in regulating endoreplication in mammals.

Reviewer #2 (Remarks to the Author):

1. *This reviewer still believes the claim that “doxorubicin-injected MGs at PD17.5 shows an increase in the proportion of 4C and > 4C luminal populations (14.86% and 47.1% increase in the overall population, respectively)” is a piece of key data, and it should be shown in the formal figure.*

Thank you for your recommendation. These data are now included in a new Figure (Figure 3) of the revised manuscript.

2. *Because authors claimed that the ATR-DDR is the major mediator for the endoreplication, this reviewer suggested to compare potential effects of ATM and ATR inhibitors to obtain data to support this claim. The authors did not do it and cited a paper, which showed “that the combined use of ATR, ATM and DNA-PK inhibitors is needed to completely shut down the DNA damage response, most likely due to the cross talk between the different DNA damage response pathways”. Therefore, they indicated that “the use of these inhibitors is unlikely to bring new insights into which pathway (ATR or ATM) is responsible for the induction of endoreplication during alveologenesis”. This is not a proper argument, as the read out here is the “endoreplication”, but not the “shut down” of the DDR. If neither ATR inhibitor nor ATM inhibitor could affect endoreplication because of the functional redundancy, the claim that “the ATR-DDR is the major mediator for the endoreplication” is questionable. If the authors think there will be too much work to carry this out in vivo, can this be done at least in the mammary organoids?*

We tried to address your comments, but the experiments proved difficult to perform. Initially, we thought the best approach would be by intraductal injection, allowing us to collect *in vivo* data. We identified pharmacological inhibitors that had previously been used in cell lines and in animals (delivery by oral gavage): one directed against ATM (Ku-60019)²⁰ and the other against ATR (AZ20)²¹⁻²³. Unfortunately, we discovered that intraductal injection of these inhibitors was not possible due to their poor solubility in aqueous solutions, which resulted in their precipitation out of solution at higher concentrations. To circumvent this issue of solubility, we turned, as the reviewer suggested, to organoid experiments, which permit the effective use of lower concentrations.

For the organoid experiments, we tested 3 ATR inhibitor concentrations (150 nM, 300 nM, and 3 μ M) based on previous studies^{7,15} (Response Figures 1-3). For the ATM inhibitor, we tested 3 concentrations (189 nM, 378 nM, 3.78 μ M) that would provide equivalent inhibition of

the target kinase as the concentrations used for the ATR inhibitor, calculated based on the manufacturer provided IC50 value (Response Figures 1-3). Only the highest concentration (3 μ M) of the ATR inhibitor successfully inhibited ATR and none of the concentrations we tested inhibited ATM (Response Figure 3). Successful inhibition of ATR, however, did not inhibit the DNA damage response pathway as may have been expected. And, while it affected endoreplication and milk production, supporting our claim that the *ATR-DDR is the major mediator for the endoreplication*, it didn't result in a reduction in these read-outs. Instead, we observe, as others have reported^{7,14,15}, that inhibiting ATR induces phosphorylation of γ H2AX, indicating the induction of DNA damage likely due to the pleiotropic effects of ATR inhibition (see more on this topic in the Response to Reviewer 1 above). Thus, consistent with increased DNA damage, we also observe more cells with >4C content (i.e. more endoreplication) and elevated levels of *Csn2* and *Lalba* (Response Figure 3).

The literature shows that ATR/ATM kinases are at the center of numerous pathways^{12,13}. While a key role for ATR is to arrest the cell cycle at the G2/M checkpoint in order to allow time for DNA repair, it also plays an integral role in the intra-S checkpoint during normal cell cycle progression and in response to DNA damage. Specifically, ATR safeguards genomic integrity through the phosphorylation of downstream effectors that are involved in the regulation of origin firing, replication fork stability, replisome function, replication factor and dNTP availability, and activation of DNA repair mechanisms^{11,24}. Indeed, ATR is essential for the survival of proliferating cells and its depletion results in embryonic lethality in mice and cell death in human cells²⁵⁻²⁷. For this reason, efforts in the field to investigate its functions by deletion or pharmacological inhibition have proven difficult.

Our experimental results suggest that ATR is the initiating kinase of the DNA damage response to replication stress in mammary luminal cells. Our findings are also bolstered by other studies showing that ATR responds to replication stress in the mammary gland. For example, it has been shown that loss of *Brca1* during pregnancy leads to the collapse of the replication fork and DNA damage due to replication stress²⁸. *Brca1* loss also impairs the function of ATR/CHK1 in the intra S-phase checkpoint. Interestingly, the authors also find that WEE1 is involved in regulating this checkpoint; we address these dual roles for WEE1 in maintaining genome integrity (intra S-phase checkpoint) and inhibiting CDK1 (G2/M checkpoint) in the revised discussion in response to your comment below. While this study on *Brca1* loss did not examine ATR/CHK function at the G2/M checkpoint, another study has shown that ATR and not ATM is the DNA damage response kinase activated by replication stress in HL-60 cells. This study showed that induction of replication stress through exposure to hydroxyurea (Hu) not only fails to result in the activation of ATM, but reduces ATM phosphorylation below the constitutive levels observed in untreated cells²⁹. Together, these studies bolster the notion that ATR is the DNA damage kinase that responds to replication stress in mammary luminal cells. And, taken together with our experiments in which we show ATR and not ATM is upregulated during pregnancy; the phosphorylation of the canonical effector of ATR, CHK1, which is responsible for the activation of the G2/M checkpoint; and hydroxyurea treatment and nucleoside supplementation promote or inhibit alveolar endoreplication, respectively, the data in aggregate strongly suggest that ATR and not ATM is the DNA damage response kinase responsible for driving endoreplication in mammary alveolar cells. Nevertheless, this has been difficult to prove by manipulating ATR/ATM expression or activity, so consequently we have softened the conclusion in our manuscript by (1) not specifying the ATR-DNA damage response pathway in the abstract (line 11), (2) by removing the specification of

ATR at the end of the Introduction (line 81) and (3) changing our conclusion about ATR in the Results section.

From: These data show that replication stress occurs during the proliferative phase of alveologenesis at mid-pregnancy, resulting in DNA damage. This DNA damage activates ATR, but not ATM, which then phosphorylates its downstream effector CHK1. This pathway remains active until the beginning of lactation, when endoreplication occurs, suggesting that the ATR-DNA damage response governs endoreplication during alveologenesis.

To: These data show that replication stress occurs during the proliferative phase of alveologenesis at mid-pregnancy, resulting in DNA damage. Given the spike in expression of pATR in late pregnancy (PD 18.5) and subsequent phosphorylation of its downstream effector CHK1, it appears that ATR, and not ATM, is activated by DNA damage. The pathway remains active until the beginning of lactation, when endoreplication occurs, suggesting that the ATR-DNA damage response governs endoreplication during alveologenesis.

Figure 1. a-c) Western blot **(a)** and quantification of pT1989-ATR/ATR **(b)** and pS1981-ATM/ATM **(c)** in lysates from murine primary organoids after treatment with DMSO vehicle control, 150nM ATR inhibitor (AZ20), 189nM ATM inhibitor (Ku-60019) or both inhibitors. Actin used as loading control. **d)** Percentage of murine primary organoid cells with 2C, 4C or >4C DNA content, as detected by FACS DNA content analysis, after treatment with DMSO vehicle control, 150nM ATR inhibitor (AZ20), 189nM ATM inhibitor (Ku-60019) or both inhibitors. **e and f)** Expression of milk protein genes *Csn2* **(e)** and *Lalba* **(f)**, as detected by RT-qPCR, after treatment with DMSO vehicle control, 150nM ATR inhibitor (AZ20), 189nM ATM inhibitor (Ku-60019) or both inhibitors. Data are shown as the mean \pm SD; n=3 biological replicates; One-way ANOVA with Tukey's comparison test. p values: * < 0.05.

Figure 2. a-c) Western blot (**a**) and quantification of pT1989-ATR/ATR (**b**) and pS1981-ATM/ATM (**c**) in lysates from murine primary organoids after treatment with DMSO vehicle control, 300nM ATR inhibitor (AZ20), 378nM ATM inhibitor (Ku-60019) or both inhibitors. Actin used as loading control. **d)** Percentage of murine primary organoid cells with 2C, 4C or >4C DNA content, as detected by FACS DNA content analysis, after treatment with DMSO vehicle control, 300nM ATR inhibitor (AZ20), 378nM ATM inhibitor (Ku-60019) or both inhibitors. **e and f)** Expression of milk protein genes *Csn2* (**e**) and *Lalba* (**f**), as detected by RT-qPCR, after treatment with DMSO vehicle control, 300nM ATR inhibitor (AZ20), 378nM ATM inhibitor (Ku-60019) or both inhibitors. Data are shown as the mean \pm SD; n=3 biological replicates; One-way ANOVA with Tukey's comparison test.

Figure 3. a-c) Western blot (**a**) and quantification of pT1989-ATR/ATR (**b**) and pS1981-ATM/ATM (**c**) in lysates from murine primary organoids after treatment with DMSO vehicle control, 3uM ATR inhibitor (AZ20), 3.78uM ATM inhibitor (Ku-60019) or both inhibitors. Actin used as loading control. **d)** Percentage of murine primary organoid cells with 2C, 4C or >4C DNA content, as detected by FACS DNA content analysis, after treatment with DMSO vehicle control, 3uM ATR inhibitor (AZ20), 3.78uM ATM inhibitor (Ku-60019) or both inhibitors. **e and f)** Expression of milk protein genes *Csn2* (**e**) and *Lalba* (**f**), as detected by RT-qPCR, after treatment with DMSO vehicle control, 3uM ATR inhibitor (AZ20), 3.78uM ATM inhibitor (Ku-60019) or both inhibitors. **g)** Representative images of γ H2AX (green) in murine primary organoids after treatment with DMSO vehicle control or 3uM ATR inhibitor (AZ20). CK8 shown in magenta and nuclear DNA shown in blue. **h)** Quantification of nuclear γ H2AX in the luminal CK8+ population of murine primary organoids after treatment with DMSO vehicle control or 3uM ATR inhibitor (AZ20), as detected by IHC. Data are shown as the mean \pm SD; n=3 biological replicates; One-way ANOVA with Tukey's comparison test. p values: * < 0.05, **<0.01, ***<0.001, ****<0.0001.

3. Previously, Vassilopoulos et al. generated a *Wee1* mammary specific knockout mouse model and found that *Wee1* plays a critical role in G2-M checkpoint and the *Wee1* deficiency resulted in some abnormality in the mammary epithelial cells (*Oncogene*. 2015 Jun 4; 34(23): 3023–3035.). Although different Cre was used, this study might need to be cited and discussed, as it might strengthen the conclusion here.

Thank you for your recommendation. This study is now cited and discussed in the revised discussion of the revised manuscript. Please also see below:

While we investigated the role of WEE1 in driving endoreplication of mammary alveolar cells during functional differentiation, other studies have evaluated the role WEE1 in ensuring proper DNA replication during proliferative phases of mammary gland development^{30,31}. In one study it was shown that loss of the KRAB zinc finger protein *Roma/Zfp157* results in increased replication stress and WEE1 downregulation³¹. Polyploidization was also observed in the *Roma* knockout. This suggests that, in the presence of genomic instability triggered by the loss of *Roma* and downregulation but not complete loss of WEE1, luminal cells undergo endoreplication to limit uncontrolled proliferation of damaged cells. In another study, mammary-specific heterozygous loss of *Wee1* in nulliparous mice resulted in increased DNA damage and dysregulation of mitotic progression, leading to the formation of luminal-type tumors after a long lag period³⁰. Interestingly, the majority of the tumor cells were polyploid, suggesting that increased DNA damage due to loss of *Wee1* during proliferation drives the endoreplication of these tumorigenic luminal cells. Homozygous knockouts, however, did not develop tumors, and it was suggested that the profound genomic instability due to complete loss of *Wee1* leads to cellular senescence, as opposed to endoreplication³⁰. Thus, a role has been established for WEE1 in inducing cellular polyploidization in response to DNA damage as a safeguard mechanism during proliferative phases of development and during tumorigenesis^{30,31}. By contrast, our study demonstrates a second role for WEE1 in harnessing DNA damage that is produced by replication stress during the proliferative phase of pregnancy to both drive functional endoreplication during the differentiative phase and facilitate the efficient production of milk during lactation.

This reviewer also raised the following 2 minor points, but they were not addressed:

1. *Figure 1a: Please indicate in the figure legends regarding how the blue and yellow colour are generated, and also the stages of the sample (P17.5?).*

We apologize for this not being indicated in the figure legend. In the revised manuscript, we detail this process of color labelling the nuclei by DNA content and indicate the stage of the sample as lactation day 5 (LD5) in both the figure legend and methods.

In Figure Legend:

Optical section from a 3D confocal image showing an alveolar unit from a MG at LD5 (left). Individual, complete nuclei color-labelled by DNA content, as calculated by the 3D sum integrated density of nuclear DNA staining using stromal cell nuclei as a reference for 2C DNA content (right; 2C in blue, >4C in yellow). E-cadherin detected in magenta. Nuclear DNA by Hoechst. White dotted line represents the border of the alveolus.

In the Methods:

Individual nuclei were segmented in 3D by filtering, thresholding and performing a 3D watershed segmentation using the channel comprising the DNA dye Hoechst. The resulting 3D binary image was used as a mask to quantify the sum integrated density of Hoechst staining within each individual, complete nucleus. Using stromal cell nuclei as a reference for diploid (2C) DNA content, we classified all complete nuclei in the image by DNA content based on their sum integrated density and labelled them by color (2C in blue, >4C in yellow). Incomplete nuclei were excluded from analysis.

2. Suggest also show a P10.5 samples as a control in this figure.

While our confocal analysis allowed for an initial visualization of DNA content *in situ*, to quantitatively evaluate DNA content in many more cells than *in situ* analysis would reasonably permit, we performed all subsequent DNA content analyses by FACS. With that said, to address the Reviewer's comment, we performed FACS DNA content analysis of CK8+ cells isolated from PD10.5 MG tissue. We find that there are no differences in the proportion of tetraploid (4C) or polyploid (>4C) cells in comparison to CK8+ cells isolated from nulliparous MG tissue (Response Figure 4).

Figure 4. a) FACS analysis of a representative MG single cell preparation used for the identification of the luminal CK8⁺ population at PD10.5. Red histogram represents the negative isotype control (IgG). Gate indicates the CK8⁺ population in relation to the negative isotype control. **b)** FACS analysis of the MG CK8⁺ cells at PD10.5 for the identification of single cells, based on propidium iodide area (PI-A) versus width (PI-W). Black box shows the gating strategy for single cell identification. **c-d)** Representative FACS histograms for DNA content analysis of the CK8⁺ population in PD10.5 (c) and nulliparous MGs (d). **e)** Quantification of FACS DNA content analysis showing the percentage of CK8⁺ cells with 2C, 4C or >4C DNA content, or in the S phase, from nulliparous and PD10.5 MGs. Data are shown as the mean \pm SD; n=3 biological replicates; two-tailed unpaired Student's t-test. p values: *** < 0.001.

Reviewer #3 (Remarks to the Author):

This is a resubmission of a paper focused on determining the role of the DNA damage response pathway in polyploidy in functionally differentiated mammary epithelial cells. The authors have addressed many of the questions from the previous review including rationalization for using the CK8 conditional mouse model and providing new data showing changes in STAT5A/STAT3; however, there are still concerns that need to be addressed.

1. Measurement of pup weights is considered the gold standard for assessing lactation function. Small changes (1.5 fold) in milk protein expression does not indicate a significant impact in lactation performance.

We are surprised the reviewer asserts that “small changes (1.5 fold) in milk protein expression does not indicate a significant impact in lactation performance.” Although we cannot find studies that examine murine lactation performance, we note that studies on bovine somatotropin (BST; also known as bovine growth hormone) in cows showed increases similar to the one we report (i.e. in the 1.5 fold range) for individual milk protein genes^{32,33}. These changes boosted milk production enough to warrant the dairy industry's adoption of recombinant BST (rBST)^{34,35}. Here, we use these changes to map the DNA damage response pathway to WEE1. Our study

culminates with the examination of our *Wee1* conditional knockout mouse model. Pups nursed by *Wee1^{fl/fl}* mothers die with no milk in their stomachs and we show consistent results across the same 3 assays used to evaluate the intraductal injection experiments: RT-qPCR (reduced milk protein gene expression), immunohistochemistry (IHC), (reduced milk) and FACS (reduced endoreplication). Thus, we successfully used our 3-assay analysis evaluating dam milk production to identify WEE1 as the CDK1 inhibitor that generates the observed heterogeneous population of polyploid cells that arises at the G2/M checkpoint during pregnancy/lactation.

*As indicated in the previous review, milk expression is highly dynamic and dependent upon the litter size and when the pups last nursed. The author commented on normalizing litter size for the *Wee1fl/fl* studies, but provided no new information on the litter size or information on the last time the pups nursed for the injection studies. For this reviewer, the lack of this information makes the data in Fig 5/6/7 not reliable.*

We apologize if we were not clear about our experimental procedures in the revised manuscript. The bottom line is that the experimental design for the intraductal injections preclude us measuring differences in pup weight.

To explain the experimental procedure more clearly, we inject the right side of a pregnant dam (4 glands: thoracic #2/3, abdominal #4 and inguinal #5 – all but the cervical #1) with the pharmacological reagent, and the left side (same 4 glands) with the vehicle control. When the dam gives birth, we allow the 8-12 pups to suckle until lactation day LD2 or LD5, depending on treatment. Then, we remove the pups, sacrifice the dam and harvest all 8 mammary glands. We use the #4 contralaterally injected mammary glands (i.e. treated versus control) for immunohistochemistry (IHC) and RT-qPCR. The other 3 contralaterally injected mammary glands are collected (i.e. 3 treated and 3 control), collated by treatment and used for FACS analysis. The key point is that by performing the intraductal injections into the same dam we can't control from which teat the pup will suckle. Therefore, there is no expectation that pups will have different weights. Instead, the analysis of contralaterally injected mammary glands (using the aforementioned 3 assays: IHC, RT-qPCR and FACS) represents a complete experiment (n=1 treated and control mammary glands harvested from one animal). This is why we perform paired statistical analysis.

By performing contralateral intraductal injections of both treatments and vehicle controls into a single animal, we are eliminating the variability we know exists between individual animals, likely due to differences in hormone expression. Thus, we are able to firmly conclude that each pharmacological treatment was responsible for the consistent and significant changes we observed across the 3 assays. We successfully used these intraductal injections to map the DNA damage response, occurring during late pregnancy/lactation, demonstrating its action via WEE1 to promote endoreplication and milk production. Moreover, we performed organoids studies with all the pharmacological reagents, assaying DNA content by FACS analysis and milk protein gene expression (*Csn2*, *Wap* and *Lalba*) by RT-qPCR; all the data are consistent across both *in vitro* and *in vivo* assays. We then generated and used a *Wee1* conditional knockout mouse model to demonstrate that pups nursed by *Wee1^{fl/fl}* mothers die with no milk in their stomachs and showed consistent results across the same 3 assays used to evaluate the intraductal injection experiments. This provides strong evidence of WEE1's role as the key CDK1 inhibitor in regulating endoreplication in the mammary gland. While previous studies have demonstrated roles for the various CDK1 inhibitors in regulating endoreplication in mammalian tissues, such a role for the CDK1 inhibitor WEE1 has only been observed in plants^{18,19}. Thus, our manipulation

of the DNA damage response, by intraductal pharmacological injections (Figs. 5/6/7) and by generation of a *Wee1* conditional knockout mouse model, allowed us to identify a novel role for WEE1 in regulating endoreplication in mammals.

Moreover, areas assessed for milk in the gland (Fig 5H-I; Fig6I-J; Fig 7g-h) is highly selective and goes to issues of rigor. Why were these areas selected? Other areas could just have easily been selected to show the opposite result.

We apologize if our methods were not made clear in the revised manuscript where we said “*Due to observed areas of high and low milk staining in a ~1cm² section of LD2 MG tissue, we imaged entire sections, quantified milk staining contained within each alveolus and calculated the average integrated density among all alveoli in the section*”. This meant that we imaged entire tissue sections comprising ~ 1/2 of each abdominal mammary gland (~1cm² in area) by stitching together a minimum of 60 fields of view to generate a tiled image. We then segmented individual alveolar lumens, using the luminal cell marker Cytokeratin 8, and quantified the integrated density of milk staining within each alveolus of the tiled image to determine the average integrated density among all alveoli in the tissue section. This process of imaging and quantification is now more clearly stated in the results and methods of the revised manuscript.

2. There are issues with the Western blots in the rebuttal. For example, in rebuttal Figure 7a and 7B, the authors show increased pATR (nullip) and pATM (PD 10.5), but there isn't a band for the ATR and ATM loading control. This suggests issues with the sample and or assay that needs to be repeated to be interpretable

In order to quantify our n=3 immunoblot data, we use the BioRad ChemiDoc MP Imaging System to collect a series of images at different intensities for each blot, being careful to ensure that we quantify the data whilst in the linear signal range. We selected the images shown in the manuscript that, in our opinion, best represented the n=3 quantified data. Below, we present overexposed blots where there is signal visible to the eye in every lane (Response Figure 5).

Figure 5. a-c) Western Blots detecting (a) pT1989-ATR/ATR, (b) pS345-CHK1/CHK1 and (c) pS1981-ATM/ATM in whole MG lysates from nulliparous (Nullip.), PD10.5, PD15.5, PD18.5 and LD2 mice. Arrows indicate alternate exposure where there is signal visible in every lane. HSP70 or ACTIN were used as loading controls.

Reviewer #4 (Remarks to the Author):

This manuscript has been significantly revised, with the results of several important new high-quality experiments. The results are novel and impactful, as they address a potentially paradigm shifting concept of the role of DNA damage in cell fate. My previous concerns have been fully addressed and I highly support the publication of this manuscript.

Thank you.

References:

- 1 Lam, F. C. The DNA damage response - from cell biology to human disease. *Transl Genet Genom* **6**, 204-222 (2022). <https://doi.org/http://dx.doi.org/10.20517/jtgg.2021.61>
- 2 Bester, A. C. *et al.* Nucleotide deficiency promotes genomic instability in early stages of cancer development. *Cell* **145**, 435-446 (2011). <https://doi.org/10.1016/j.cell.2011.03.044>
- 3 Halliwell, J. A. *et al.* Nucleosides Rescue Replication-Mediated Genome Instability of Human Pluripotent Stem Cells. *Stem cell reports* **14**, 1009-1017 (2020). <https://doi.org/10.1016/j.stemcr.2020.04.004>
- 4 Ruiz, S. *et al.* Limiting replication stress during somatic cell reprogramming reduces genomic instability in induced pluripotent stem cells. *Nat Commun* **6**, 8036 (2015). <https://doi.org/10.1038/ncomms9036>
- 5 Freije, A. *et al.* Cyclin E drives human keratinocyte growth into differentiation. *Oncogene* **31**, 5180-5192 (2012). <https://doi.org/10.1038/onc.2012.22>
- 6 Morii, M. *et al.* Imatinib inhibits inactivation of the ATM/ATR signaling pathway and recovery from adriamycin/doxorubicin-induced DNA damage checkpoint arrest. *Cell Biol. Int.* **39**, 923-932 (2015). <https://doi.org/10.1002/cbin.10460>
- 7 Molinuevo, R., Freije, A., Contreras, L., Sanz, J. R. & Gandarillas, A. The DNA damage response links human squamous proliferation with differentiation. *J Cell Biol* **219** (2020). <https://doi.org/10.1083/jcb.202001063>
- 8 Saintigny, Y. *et al.* Characterization of homologous recombination induced by replication inhibition in mammalian cells. *EMBO J* **20**, 3861-3870 (2001). <https://doi.org/10.1093/emboj/20.14.3861>
- 9 Halicka, D. *et al.* DNA Damage Response Resulting from Replication Stress Induced by Synchronization of Cells by Inhibitors of DNA Replication: Analysis by Flow Cytometry. *Methods Mol. Biol.* **1524**, 107-119 (2017). https://doi.org/10.1007/978-1-4939-6603-5_7
- 10 Dwivedi, V. K. *et al.* Replication stress promotes cell elimination by extrusion. *Nature* **593**, 591-596 (2021). <https://doi.org/10.1038/s41586-021-03526-y>
- 11 Menolfi, D. & Zha, S. ATM, DNA-PKcs and ATR: shaping development through the regulation of the DNA damage responses. *Genome Instability & Disease* **1**, 47-68 (2020). <https://doi.org/doi.org/10.1007/s42764-019-00003-9>
- 12 Stokes, M. P. *et al.* Profiling of UV-induced ATM/ATR signaling pathways. *Proc Natl Acad Sci U S A* **104**, 19855-19860 (2007). <https://doi.org/10.1073/pnas.0707579104>
- 13 Matsuoka, S. *et al.* ATM and ATR substrate analysis reveals extensive protein networks responsive to DNA damage. *Science* **316**, 1160-1166 (2007). <https://doi.org/10.1126/science.1140321>

- 14 Chanoux, R. A. *et al.* ATR and H2AX cooperate in maintaining genome stability under replication stress. *J Biol Chem* **284**, 5994-6003 (2009).
<https://doi.org/10.1074/jbc.M806739200>
- 15 Muralidharan, S. V. *et al.* BET bromodomain inhibitors synergize with ATR inhibitors to induce DNA damage, apoptosis, senescence-associated secretory pathway and ER stress in Myc-induced lymphoma cells. *Oncogene* **35**, 4689-4697 (2016).
<https://doi.org/10.1038/onc.2015.521>
- 16 Diril, M. K. *et al.* Cyclin-dependent kinase 1 (Cdk1) is essential for cell division and suppression of DNA re-replication but not for liver regeneration. *Proc Natl Acad Sci U S A* **109**, 3826-3831 (2012). <https://doi.org/10.1073/pnas.1115201109>
- 17 Neganova, I. *et al.* CDK1 plays an important role in the maintenance of pluripotency and genomic stability in human pluripotent stem cells. *Cell Death Dis* **5**, e1508 (2014).
<https://doi.org/10.1038/cddis.2014.464>
- 18 Sun, Y. *et al.* Characterization of maize (*Zea mays* L.) Wee1 and its activity in developing endosperm. *Proc Natl Acad Sci U S A* **96**, 4180-4185 (1999).
<https://doi.org/10.1073/pnas.96.7.4180>
- 19 Gonzalez, N., Gevaudant, F., Hernould, M., Chevalier, C. & Mouras, A. The cell cycle-associated protein kinase WEE1 regulates cell size in relation to endoreduplication in developing tomato fruit. *Plant J.* **51**, 642-655 (2007). <https://doi.org/10.1111/j.1365-313X.2007.03167.x>
- 20 Li, H. *et al.* CRISPR metabolic screen identifies ATM and KEAP1 as targetable genetic vulnerabilities in solid tumors. *Proc Natl Acad Sci U S A* **120**, e2212072120 (2023).
<https://doi.org/10.1073/pnas.2212072120>
- 21 Chu, S. H. *et al.* Inhibition of MEK and ATR is effective in a B-cell acute lymphoblastic leukemia model driven by Mll-Af4 and activated Ras. *Blood Adv* **2**, 2478-2490 (2018).
<https://doi.org/10.1182/bloodadvances.2018021592>
- 22 Pereira, C. *et al.* Multiple 9-1-1 complexes promote homolog synapsis, DSB repair, and ATR signaling during mammalian meiosis. *Elife* **11** (2022).
<https://doi.org/10.7554/eLife.68677>
- 23 Hanna, C. *et al.* ATM Kinase Inhibition Preferentially Sensitises PTEN-Deficient Prostate Tumour Cells to Ionising Radiation. *Cancers (Basel)* **13** (2020).
<https://doi.org/10.3390/cancers13010079>
- 24 Saldivar, J. C., Cortez, D. & Cimprich, K. A. The essential kinase ATR: ensuring faithful duplication of a challenging genome. *Nat Rev Mol Cell Biol* **18**, 622-636 (2017).
<https://doi.org/10.1038/nrm.2017.67>
- 25 Brown, E. J. & Baltimore, D. ATR disruption leads to chromosomal fragmentation and early embryonic lethality. *Genes Dev* **14**, 397-402 (2000).
- 26 de Klein, A. *et al.* Targeted disruption of the cell-cycle checkpoint gene ATR leads to early embryonic lethality in mice. *Curr Biol* **10**, 479-482 (2000).
[https://doi.org/10.1016/s0960-9822\(00\)00447-4](https://doi.org/10.1016/s0960-9822(00)00447-4)
- 27 Cortez, D., Guntuku, S., Qin, J. & Elledge, S. J. ATR and ATRIP: partners in checkpoint signaling. *Science* **294**, 1713-1716 (2001). <https://doi.org/10.1126/science.1065521>
- 28 Xu, X. *et al.* BRCA1 represses DNA replication initiation through antagonizing estrogen signaling and maintains genome stability in parallel with WEE1-MCM2 signaling during pregnancy. *Hum. Mol. Genet.* **28**, 842-857 (2019). <https://doi.org/10.1093/hmg/ddy398>

- 29 Kurose, A. *et al.* Effects of hydroxyurea and aphidicolin on phosphorylation of ataxia telangiectasia mutated on Ser 1981 and histone H2AX on Ser 139 in relation to cell cycle phase and induction of apoptosis. *Cytometry A* **69**, 212-221 (2006).
<https://doi.org:10.1002/cyto.a.20241>
- 30 Vassilopoulos, A. *et al.* WEE1 murine deficiency induces hyper-activation of APC/C and results in genomic instability and carcinogenesis. *Oncogene* **34**, 3023-3035 (2015).
<https://doi.org:10.1038/onc.2014.239>
- 31 Ho, T. L. F., Guilbaud, G., Blow, J. J., Sale, J. E. & Watson, C. J. The KRAB Zinc Finger Protein Roma/Zfp157 Is a Critical Regulator of Cell-Cycle Progression and Genomic Stability. *Cell reports* **15**, 724-734 (2016). <https://doi.org:10.1016/j.celrep.2016.03.078>
- 32 Yang, J., Zhao, B., Baracos, V. E. & Kennelly, J. J. Effects of bovine somatotropin on beta-casein mRNA levels in mammary tissue of lactating cows. *J Dairy Sci* **88**, 2806-2812 (2005). [https://doi.org:10.3168/jds.S0022-0302\(05\)72960-X](https://doi.org:10.3168/jds.S0022-0302(05)72960-X)
- 33 McCoard, S. A. *et al.* Mammary transcriptome analysis of lactating dairy cows following administration of bovine growth hormone. *Animal : an international journal of animal bioscience* **10**, 2008-2017 (2016). <https://doi.org:10.1017/S1751731116000987>
- 34 Dohoo, I. R. *et al.* A meta-analysis review of the effects of recombinant bovine somatotropin. 1. Methodology and effects on production. *Can. J. Vet. Res.* **67**, 241-251 (2003).
- 35 McBride, W. D., Short, S. & Ell-Osta, H. The Adoption and Impact of Bovine Somatotropin on U.S. Dairy Farms. *Review of Agricultural Economics* **26**, 472-488 (2004).

Reviewer #1 (Remarks to the Author):

The authors have still not addressed the physiological signals that initiate the DNA damage response. I understand that this is not easy to do but it does diminish the impact of their work. Nevertheless, this is an interesting study.

Reviewer #2 (Remarks to the Author):

The authors have addressed all the points I raised. I am satisfied with it.

Reviewer #3 (Remarks to the Author):

The authors have included a significant amount of work, improving the manuscript. The majority of my concerns have been adequately answered; however, a concern did arise in the response to reviewer 2 that needs to be addressed. As reviewer 2 brought up regarding mammary-specific deletion of WEE1 kinase, a similar study by Dyer et al (Breast Cancer Research and Treatment. 2017 Dec;166(3):725-741) showed that deletion of ATM using the WAP-cre resulted in decreased alveolar development during late pregnancy and lactation performance. This study is significant because it goes to the premise by the authors that ATR is required and not ATM for lactation and needs to be discussed.

REVIEWERS' COMMENTS

Reviewer #1 (Remarks to the Author):

The authors have still not addressed the physiological signals that initiate the DNA damage response. I understand that this is not easy to do but it does diminish the impact of their work. Nevertheless, this is an interesting study.

Thank you for acknowledging the interest of the study.

Reviewer #2 (Remarks to the Author):

The authors have addressed all the points I raised. I am satisfied with it.

Thank you.

Reviewer #3 (Remarks to the Author):

The authors have included a significant amount of work, improving the manuscript. The majority of my concerns have been adequately answered; however, a concern did arise in the response to reviewer 2 that needs to be addressed. As reviewer 2 brought up regarding mammary-specific deletion of WEE1 kinase, a similar study by Dyer et al (Breast Cancer Research and Treatment. 2017 Dec;166(3):725-741) showed that deletion of ATM using the WAP-cre resulted in decreased alveolar development during late pregnancy and lactation performance. This study is significant because it goes to the premise by the authors that ATR is required and not ATM for lactation and needs to be discussed.

Thank you for bringing up the interesting paper by Dyer et al. The findings of this study could potentially be interpreted to suggest that ATM, and not ATR, is the DNA repair enzyme responding to the DNA damage that peaks during mid-pregnancy or that, in the absence of ATM, ATR is not even serving as a back-up DNA repair enzyme, as evidenced by the observed reduction in lactation. However, the Dyer paper shows that alveologenesis and lactogenesis occur normally in *Atm* Δ/Δ mammary glands, and it is not until mid-lactation that structural and lactational defects are observed in the *Atm* Δ/Δ mice. The authors state: “Taken together, these findings support the conclusion that *Atm* Δ/Δ dams display lactation defects at mid-lactation time points and, since this phenotype is not apparent at earlier points in lactation, this defect appears to be progressive in nature.” Further, they link these defects to increased apoptosis in the lactating KO gland, likely due to reduced *Sod2* expression, with the authors concluding: “Moreover, as *Atm*-deficiency results in increased apoptosis within the mammary gland, we propose that *Atm* is not required to support milk production during lactation per se but rather to maintain mammary epithelial cell viability.” Thus, the paper does not suggest that ATM is required in the mammary gland to build a milk supply, but rather that it is required to sustain the milk supply beyond early lactation. Loss of *ATM* under *Wap-Cre* still allows for alveologenesis and lactogenesis to occur normally, likely because ATR is present to respond to DNA damage and mount the DNA damage response that results in alveolar endoreplication and efficient milk

production. Thus, the data presented by Dyer et al. are consistent with our model. And we have included a sentence in the manuscript reflecting this information.

Line 306: Moreover, previous studies have shown that loss of *Atm* under *Wap-Cre* control allows for normal alveologenesis and lactogenesis, with structural and lactational defects only appearing at mid-lactation due to increasing cell death by apoptosis⁴⁷.